# Genome-wide maps of nucleolus interactions reveal distinct layers of repressive chromatin domains

Cristiana Bersaglieri[1,2], Jelena Kresoja-Rakic[1], Shivani Gupta[1], Dominik Bär[1], Rostyslav Kuzyakiv[1,3], Martina Panatta[1,4] & Raffaella Santoro [1✉]

Eukaryotic chromosomes are folded into hierarchical domains, forming functional compartments. Nuclear periphery and nucleolus are two nuclear landmarks contributing to repressive chromosome architecture. However, while the role of nuclear lamina (NL) in genome organization has been well documented, the function of the nucleolus remains under-investigated due to the lack of methods for the identification of nucleolar associated domains (NADs). Here we have established DamID- and HiC-based methodologies to generate accurate genome-wide maps of NADs in embryonic stem cells (ESCs) and neural progenitor cells (NPCs), revealing layers of genome compartmentalization with distinct, repressive chromatin states based on the interaction with the nucleolus, NL, or both. NADs show higher H3K9me2 and lower H3K27me3 content than regions exclusively interacting with NL. Upon ESC differentiation into NPCs, chromosomes around the nucleolus acquire a more compact, rigid architecture with neural genes moving away from nucleoli and becoming unlocked for later activation. Further, histone modifications and the interaction strength within A and B compartments of NADs and LADs in ESCs set the choice to associate with NL or nucleoli upon dissociation from their respective compartments during differentiation. The methodologies here developed will make possible to include the nucleolar contribution in nuclear space and genome function in diverse biological systems.

[1] Department of Molecular Mechanisms of Disease, DMMD, University of Zurich, 8057 Zurich, Switzerland. [2] Molecular Life Science Program, Life Science Zurich Graduate School, University of Zurich, 8057 Zurich, Switzerland. [3] Service and Support for Science IT, University of Zurich, 8057 Zurich, Switzerland. [4] RNA Biology Program, Life Science Zurich Graduate School, University of Zurich, 8057 Zurich, Switzerland. ✉email: raffaella.santoro@dmmd.uzh.ch

In the nucleus of eukaryotic cells, chromosomes are arranged in a complex three-dimensional (3D) architecture that is thought to be important to ensure the correct execution of gene expression programs[1–5]. For example, it has been observed that the genome segregates into two large compartments, named A-type and B-type, that correspond to domains highly enriched for open and active chromatin and for closed and repressed chromatin, respectively[6]. One important aspect of spatial genome organization is also the local nuclear environment. Subnuclear compartments can serve as scaffold for chromatin tethering and allow the concentration of factors, thereby facilitating functions that rely on proteins found in limiting concentrations[7]. Genomic interactions with the nuclear lamina (NL), which lies on the inner surface of the inner nuclear membrane, are mainly characterized by features typical of heterochromatin[8]. Lamina associated domains (LADs) contain genes in transcriptionally silent state or with low expression levels, have a low overall gene density, correspond to late-replicating DNA, and are typically enriched for repressive histone marks[9–11]. Another compartment suggested to serve as a scaffold for the location of repressive chromatin is the nucleolus, a membraneless sub-nuclear compartment that is the site of ribosome biogenesis[12] and is made up of distinct, coexisting liquid phases[13]. It has been suggested that the nucleolus and the NL might serve as interchangeable scaffolds for the localization of repressive domains[14,15]. However, the understanding of this genomic dynamics in the nuclear space remains still elusive. While the role of NL in genome organization has been well documented due to the identification and characterization of LADs in many cell types[8], the function of the nucleolus has remained under-investigated. One of the major reasons is that the identification of nucleolar associated domains (NADs) remains still a technical challenge. Previous attempts were based on the biochemical purification of nucleoli, a method that relies on sonication of nuclei, adjusting the power so that nucleoli remain intact while the rest of the nuclei are fragmented[16–18]. Using this methodology, first insights were provided into the composition of NADs, which appear to mainly consist of inactive regions[19–21]. However, this method presents several technical limitations. First, since the heterochromatin is generally resistant to sonication[22], the identification of NADs upon sequencing of nucleoli purified through sonication can be biased toward repressive chromatin domains. Secondly, the experimental procedures to isolate nucleoli can be subjected to a certain variation, generating highly divergent NAD maps, even from the same cell types[23,24] (Supplementary Fig. 1). Third, it is difficult to achieve the purification of nucleoli in cells with open genome, such as embryonic stem cells (ESCs), unless protein-DNA crosslinking reagents are used, with a consequent extension of the sonication time. Consequently, while data of LADs have been frequently used in studies aimed to analyze genome organization in the cell's nucleus, genomic contacts with the nucleolus have so far been excluded from these analyses. However, to fully understand the relationship between 3D genome organization and function, it is necessary to integrate also information on the organization of chromosomes around the nucleolus. This calls for the establishment of methods for accurate genome-wide mapping of NADs that should be based on experimental procedures alternative to the highly variable biochemical isolation of nucleoli.

In this study, we establish Nucleolar-DamID and HiC-rDNA methods that provide accurate genome-wide maps of NADs. The data reveal layers of genome compartmentalization by showing distinct, repressive transcriptional and chromatin states based on the interaction with the nucleolus, NL, or both. NADs correspond to regions of the genome with a repressive state that are enriched in H3K9me2 and depleted in H3K27me3 relative to sequences only contacting the NL. Moreover, we find that the chromosome organization around the nucleolus changes according to the developmental stage. Upon differentiation of ESCs into neural progenitor cells (NPCs), the architecture of all chromosomes surrounding the nucleolus becomes more rigid and compact, increasing the interaction frequency with centromere-proximal regions. Our analysis also uncovers histone modifications and the interaction strength within A and B compartment of NADs and LADs in ESCs set the choice to associate with the nucleolus or NL upon detachment from their respective compartments during differentiation. Although still transcriptionally inactive, genes moving away from the nucleolus in NPCs are implicated in neuron development, indicating that the detachment from the repressive nucleolar compartment marks the first step toward activation in later stages of differentiation. The results reveal the role of the nucleolus as a repressive compartment that is implicated in the control of gene expression program during lineage commitment. Finally, the methodologies developed here for the identification of NADs at the genome-wide level will now make possible to include the contribution of the nucleolus in future studies investigating the relationship between nuclear space and genome function.

## Results

**Establishment of Nucleolar-DamID**. We thought to map NADs by establishing an alternative experimental procedure that is not based on biochemical isolation of nucleoli. We reasoned to adapt the DNA adenine methyltransferase identification (DamID) method, which was successfully used to identify LADs in many cell types[25]. In this application, Lamin B1 was fused to Dam from *Escherichia coli*. When Lamin B1-Dam is expressed in cells, DNA in molecular contact with the NL is methylated at the N6 position of adenine (m6A) within GATC sequences and can be mapped. However, since the nucleolus is a membrane-free compartment, the application of DamID for the identification of NADs (Nucleolar-DamID) requires further adaptations. A first criterion was to fuse Dam with proteins that are mainly localized within nucleolus and interact with DNA independently of the sequence content. We reasoned to exclude nucleolar proteins implicated in ribosome biogenesis (nucleolin, fibrillarin, UBF, etc.) since their use might influence the readout of Nucleolar-DamID, such as adenine methylation of sequences only within the rRNA genes, thereby excluding the detection of other genomic domains located within nucleoli. Therefore, we thought to engineer a nucleolar histone, which can bind DNA sequences without motif specificity and has the ability to localize exclusively within nucleoli (Fig. 1a). To assess this possibility, we inserted a nucleolar localization signal (NoLS, RKKRKKK)[26,27] at the C-terminus of the histone H2B (H2B-NoLS). Live cell imaging of NIH3T3 cells transfected with a plasmid expressing GFP-tagged H2B-NoLS revealed a prominent and preferential localization in nucleoli when compared to the homogeneous nuclear distribution of H2B-GFP (Fig. 1b). We obtained similar results with immunofluorescence analyses showing that GFP-tagged H2B-NoLS colocalizes with the nucleolar protein NPM1 (Fig. 1c). Chromatin fractionation analyses showed that H2B-GFP-NoLS associates with chromatin similarly to H2B-GFP (Fig. 1d). Next, we tested whether the nucleolar localization of H2B-NoLS depends on the integrity of the nucleolus by inhibiting rRNA synthesis with Actinomycin D (ActD), a potent inhibitor of rRNA gene transcription. Downregulation of rRNA synthesis is known to induce a spatial reorganization of the nucleolar structure with the migration of its components, including rRNA genes, to the nucleolar periphery, forming the so-called nucleolar caps[28]. The nucleolar caps are also bound by fibrillarin and the nucleolar transcription terminator factor I (TTF1). Treatment of NIH3T3 cells with ActD for 24 h after transfection of plasmids expressing GFP-tagged proteins

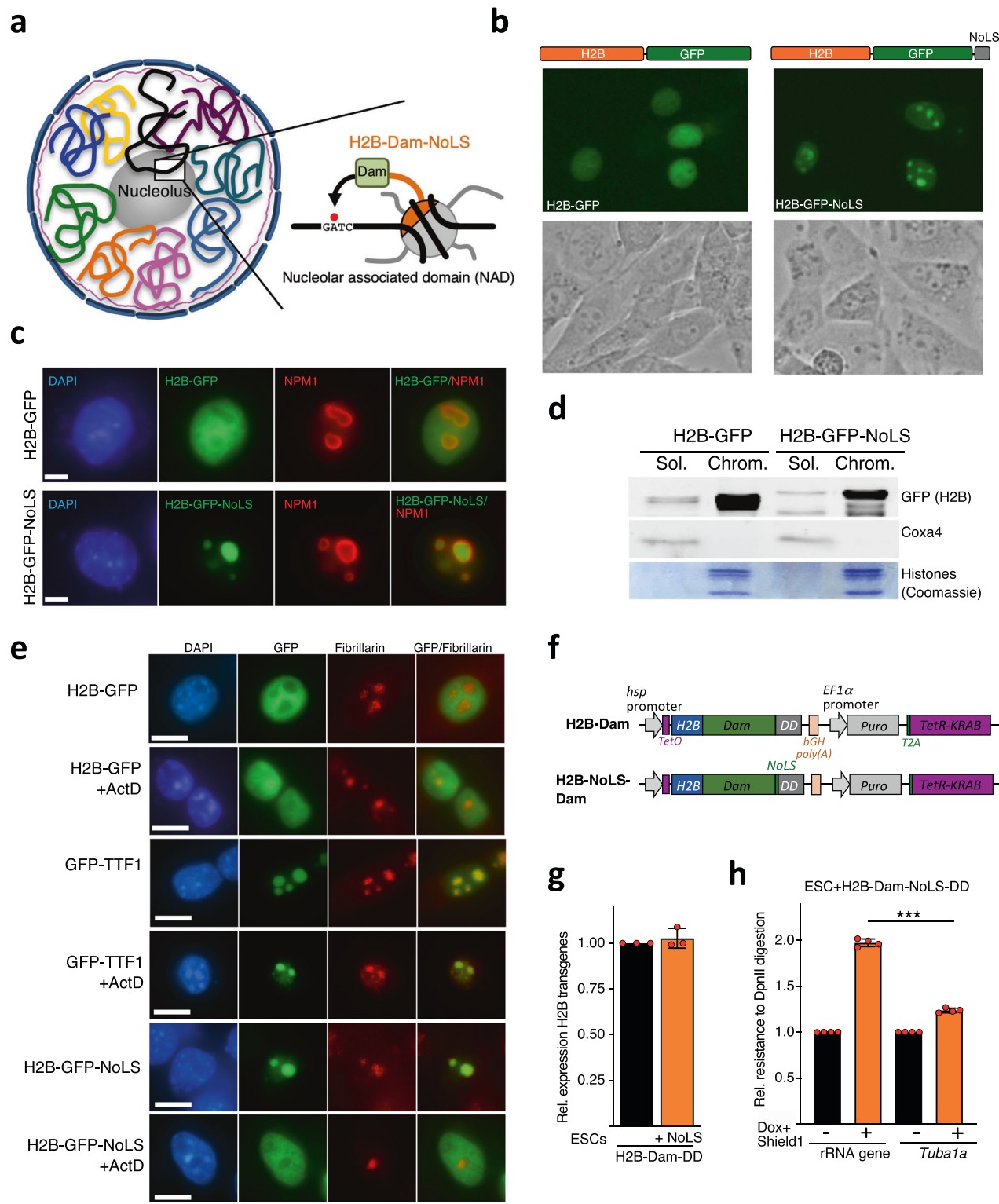

did not affect the localization of H2B-GFP whereas, as expected, Fibrillarin and ectopically expressed GFP-TTF1, which both associate with rRNA genes[29], segregated into nucleolar caps (Fig. 1e). In contrast, upon ActD treatment, H2B-GFP-NoLS signal localized outside the nucleolus. These results indicated that the nucleolar localization of H2B-NoLS depends on nucleolus integrity. Furthermore, the data suggest that genomic regions that incorporated H2B-NoLS in the nucleolus redistributed outside of it upon nucleolar segregation.

To assess the efficacy of the H2B-NoLS in DamID application, we established mouse ESC lines with heterozygotic integration into the Rosa26 locus of a transgene that allows inducible expression of H2B-Dam or H2B-Dam-NoLS chimeric proteins (Fig. 1f, Supplementary Fig. 2a). In order to not saturate Dam sites, the expression of Dam-fused proteins was kept low by placing transgene transcription under the control of the minimal hsp promoter that was further modulated using a double inducible system for the control of transcription and protein

**Fig. 1 Establishment of Nucleolar-DamID. a** Scheme representing the strategy for the establishment of Nucleolar-DamID. **b** Live cell imaging of NIH3T3 cells 24 h post-transfection with plasmids expressing H2B-GFP and H2B-GFP-NoLS under the minimal CMV promoter. Phase contrast images serve to visualize the nucleoli. **c** Representative immunofluorescence images of NIH3T3 cells transfected with plasmids expressing H2B-GFP and H2B-NoLS-GFP. Nucleoli can be visualized with the nucleolar protein NPM1. Scale bars represent 5 μm. **d** Chromatin-bound (Chrom.) and soluble (Sol.) fractions of equivalent number of NIH3T3 cells transfected with H2B-GFP and H2B-GFP-NoLS were analyzed by western blot using anti-GFP antibodies. Coxa4 and histones are shown as loading and fractionation control. **e** Representative immunofluorescence images of NIH3T3 cells transfected with plasmids expressing H2B-GFP, TTF1-GFP, and H2B-NoLS-GFP and treated for 24 h with or without ActD (50 ng/ml). ActD was added to cells 24 h post-transfection. The nucleolar protein Fibrillarin serve to visualize nucleoli in untreated cells and nucleolar caps in ActD-treated cells. Large nucleoli can be visualized by the low DAPI intensity. Scale bars represent 5 μm. **f** Scheme representing the constructs used for double inducible expression of Dam-fused H2B and nucleolar H2B (H2B-NoLS) proteins. **g** qRT-PCR showing similar expression levels of H2B-Dam and H2B-NoLS-Dam in ESCs. Mean values from data from three biologically independent experiments. Error bars represent s.d. **h** m6A levels at rRNA genes and *Tuba1a* in ESCs with and without 15 h treatment with 100 ng Doxycycline (Dox) and 1 μM Shield1. m6A levels were measured by digestion of genomic DNA with DpnII, which is blocked by m6A, followed by quantitative amplification with primers encompassing the Dam GATC element. Normalization was achieved through measurements with primers encompassing sequences lacking GATC. Mean values from data of four biologically independent experiments. Error bars represent s.d. Statistical significance (*P*-values) for the experiments was calculated using the paired two-tailed *t* test (*** <0.001). Source data are provided as a Source Data file.

stability. We used the TetR-KRAB repressor system by inserting two Tet operator (*TetO*) sequences downstream the minimal *hsp* promoter and an *EF1α* promoter-*Puro-T2A-TetR-Krab* cassette downstream the *Dam* transgenes. Furthermore, we placed at the C-terminus of both transgenes a destabilization domain (DD) that causes proteins to be rapidly targeted for proteasomal degradation unless the protein is shielded by the synthetic small molecule Shield1[30]. The lack of leakiness of this inducible system was confirmed by the transfection of HEK 293T cells with plasmids containing the sequences *hsp-TetO-GFP-Dam-DD-EF1a-puro-T2A-TetR-KRAB* (Supplementary Fig. 2b). Expression of H2B-Dam-DD and H2B-Dam-NoLS-DD transgenes in the corresponding ESC lines were induced for 15 h of treatment with doxycycline (Dox) and Shield1. The low expression of both transgenes did not allow measurements of protein levels by western blot but only quantifications by qRT-PCR, which showed that H2B-Dam-DD and H2B-Dam-NoLS-DD were expressed at similar levels (Fig. 1g). The expression of the Dam-fused histones constructs did not affect ESC morphology and 45S pre-rRNA and pluripotency gene expression levels (Supplementary Fig. 2c,d), indicating that both rRNA gene transcription and pluripotency states are not perturbed. To test the specificity of H2B-Dam-NoLS in depositing m6A at nucleolar sequences, we measured GATC methylation levels at rRNA genes, which are located in nucleoli, and *Tuba1a*, which we did not expect to be located in nucleoli (Fig. 1h). We found significantly higher m6A levels at rRNA genes compared to *Tuba1a*, indicating that the nucleolar H2B histone was preferentially incorporated in rRNA genes, the known genetic component of the nucleolus.

**Nucleolar-DamID identifies NADs**. The identification of LADs by DamID has been obtained by measuring the m6A ratio of Lamin B1-Dam over a freely diffusible Dam fused to GFP (GFP-Dam)[9]. However, in the Nucleolar-DamID, we opted to use as control the measurement of m6A levels in ESCs expressing H2B-Dam. This strategy was based on the fact that compared to GFP-Dam, which has also been reported to have some preference for open chromatin[31], the two Dam-fused histones should only differ in their nuclear localization. Furthermore, the use of H2B-Dam as control will also serve to compensate for the eventual incorporation of nucleolar histones in genomic regions outside the nucleolus.

We combined two independent Nucleolar-DamID experiments, which were highly correlated (Pearson correlation 0.89) (Supplementary Fig. 3a). We used a previously established DamIDseq pipeline[32] and constructed genome-wide maps of NADs in ESCs by taking the resulting H2B-Dam-NoLS over H2B-Dam m6A

ratio as a measure for the relative contact frequency of DNA sequences with the nucleolus (FDR < 0.01) (Fig. 2a, Supplementary Fig. 3b, Supplementary Data 1). We found that nucleolar H2B contacts in ESCs display broad domains ranging between 70 kb and 6.2 Mb (Supplementary Fig. 3c). These contacts (from here on termed as NADs) were distributed all over the chromosomes with some preferences for chromosomes containing rRNA genes (Fig. 2b, c, Supplementary Fig. 3b). The X chromosome, which is active in the male ESCs used in this study, showed the lowest enrichment in NADs. Consistent with the well-known co-localization of centromeres in the vicinity of the nucleolus[33], we found that NADs are enriched in the centromere-proximal regions of the large majority of chromosomes, which in the mouse cells are all acrocentric (Fig. 2b). Furthermore, the Nucleolar-DamID identified as NADs a sub-class of NADs that the SPRITE method identified to interact with rRNA transcripts and form inter-chromosomal contacts (i.e. nucleolar hub)[34] (Supplementary Fig. 3b). To further support the specificity of the Nucleolar-DamID, we recovered reads containing rRNA gene (rDNA) contacts from a high resolution (<750 bp) Hi-C map in ESCs, the highest to date in mammalian cells[35] (from here on named as HiC-rDNA). The adaptation of HiC analysis for the identification of rDNA contacts was based on the modification of the mouse reference genome, which does not contain rRNA gene sequences, by adding at the end of chromosome 12 one copy of rRNA gene unit (see methods). As in the case of NADs identified by SPRITE[34], rDNA contacts obtained with HiC-rDNA should represent a sub-class of NADs since not all genomic domains associating with the nucleolus must necessarily interact with the rRNA genes. The most frequent rDNA contacts were found at chromosomes containing rRNA genes (Fig. 2d, e, Supplementary Fig. 4, Supplementary Data 2), an expected result since these interactions should mainly occur in *cis*. Furthermore, rDNA contacts were enriched at centromeric-proximal regions of all chromosomes (Fig. 2e, f, Supplementary Fig. 4), indicating the presence of genomic interactions between different chromosomes. Importantly, 76% of rDNA contacts identified by HiC-rDNA correspond to NADs found with the Nucleolar-DamID (Fig. 2g, h). Thus, the Nucleolar-DamID can identify a large portion of rDNA contacts, a sub-class of NADs identified by HiC-rDNA. Finally, we measured by DNA-FISH the cellular localization of several regions identified as NAD by the Nucleolar-DamID. We found that all analyzed regions identified to contact the nucleolus but not the NL (NAD-only, see later in the text for the details of this classification) display a significant closed proximity to the nucleolus relative to regions corresponding to LAD and not mapped as NAD (LAD-only) and sequences that do not contact the nucleolus or the NL (iNAD/iLAD, Fig. 2i–l, Supplementary

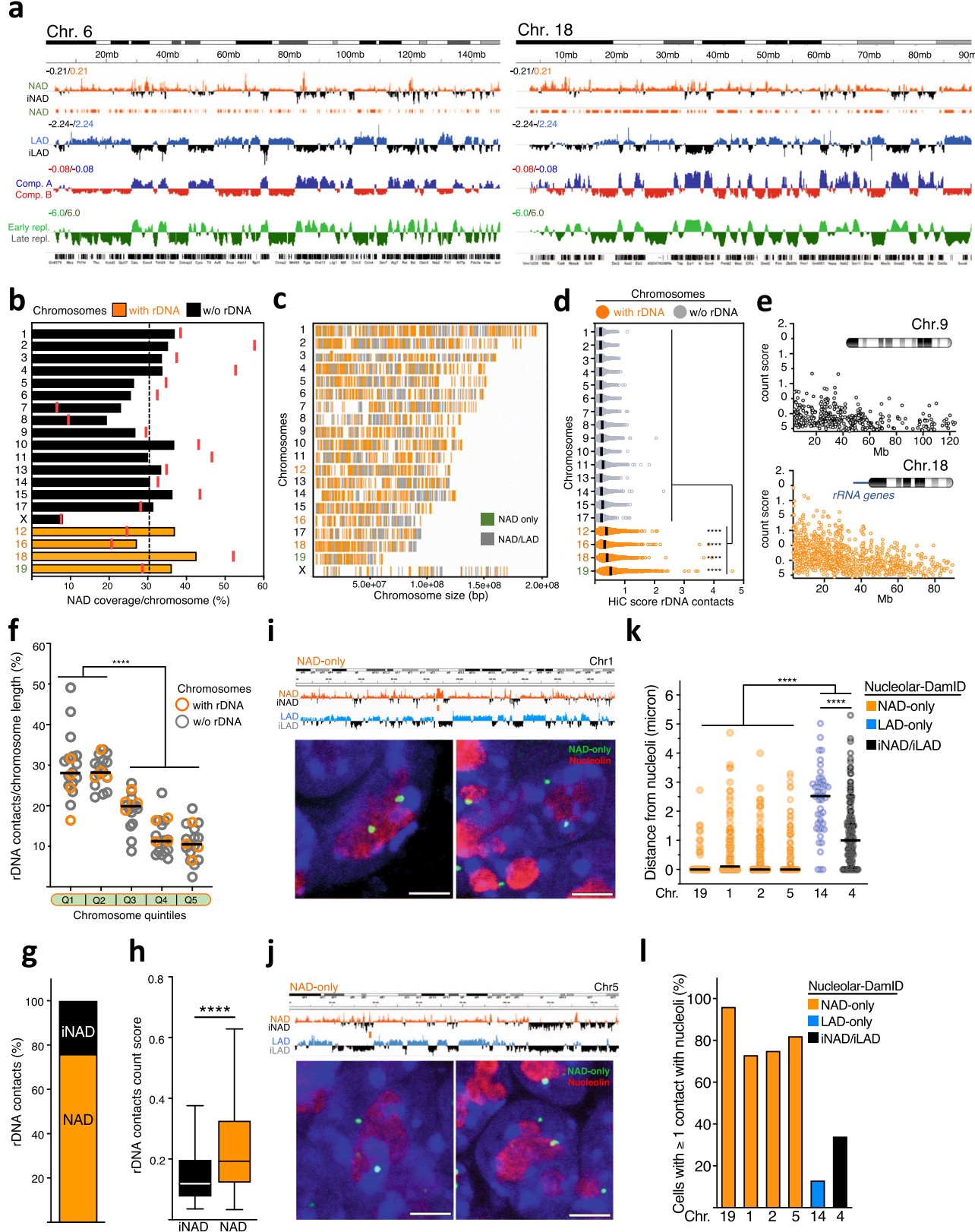

Fig. 5a–e). Importantly, the two NADs identified by the Nucleolar-DamID at chromosomes 1 and 5 were not detected as NAD by previous studies using nucleolar purification-based methods[23,24]. Together, the data showed that the Nucleolar-DamID is an accurate method for genome-wide identification of NADs.

**Distinct layers of repressive chromatin states distinguish genomic domains according to their interaction with the nucleolus, nuclear lamina, or both.** To gain insights on how genome compartmentalization at nucleolus and NL is related to each other, we analyzed the NAD composition relative to LADs,

**Fig. 2 Nucleolar-DamID identifies NADs. a** Chromosomal view of NADs, LADs[9], A and B compartments[41], and early and late replicating DNA[74] in ESCs. NADs are measured as log$_2$ ratio of m6A levels between H2B-Dam-NoLS and H2B-Dam. iNAD: domains not contacting nucleoli. iLAD: domains not contacting NL. **b** Bars represent NAD coverage values at mouse chromosomes. Red lines showed NAD coverage values in the centromere-proximal end of each chromosome. Dotted line shows NAD whole genome coverage. **c** Nucleolar H2B-Dam chromosomal interaction maps. **d** rDNA-contacts obtained from HiC maps[35]. Data represent HiC scores of unique contacts for each chromosome. Statistical significance (P-values) was calculated using the unpaired two-tailed *t* test (**** <0.0001). Lines are mean values. **e** Representative images showing normalized count score of rDNA-contacts obtained from HiC-rDNA on chromosomes 9 and 18. **f** Distribution of rDNA-contacts. Values represent the proportion of the number of unique contacts for each chromosome quintile. Statistical significance (P-values) was calculated using the unpaired two-tailed *t* test (**** <0.0001). **g** NADs identified by Nucleolar-DamID are enriched in rDNA-contacts. Data represent the proportion of unique HiC-rDNA contacts at NADs and regions non contacting the nucleolus (iNAD). **h** Hi-C normalized count score of identified unique rDNA-contacts at NADs and iNADs. Statistical significance (P-values) was calculated using the unpaired two-tailed *t* test (**** <0.0001). Box plots depict the minimum and maximum values. The horizontal line within the boxes represents the mean value. **i, j** Upper panels represent NAD and LAD profiles of chromosome 1 (**i**) and 5 (**j**) and the DNA-FISH probes (orange bar) hybridizing to regions identified by the Nucleolar-DamID as NAD-only. Lower panels. Example images from immunofluorescences for nucleolin (red) combined with the corresponding DNA-FISH probe (green) and DAPI (blue). Scale bar is 5μm. **k** Distance of the indicated DNA-FISH probes from nucleoli (micron). Statistical significance (P-values) was calculated using the unpaired two-tailed *t* test (**** <0.0001). **l** Quantification of the number of cells displaying at least one DNA-FISH probe signal contacting the nucleolus. Data are from the measurements of 70–120 cells for each condition. Source data are provided as a Source Data file.

which were previously identified in ESCs using LaminB1-DamID[9]. We found that about 53% of NADs correspond to LADs whereas 40% of LADs are also NADs (Fig. 3a, b, Supplementary Data 1). These results are consistent with early works showing that a large portion of NADs obtained from purified nucleoli correspond to LADs[19]. We termed this NAD subclass NAD/LAD whereas NADs not overlapping with LADs were named NAD-only. Accordingly, the average Lamin B1-Dam signal highly increased at NAD/LAD boundaries and to a much less degree at NAD-only boundaries (Fig. 3c). These results support previous imaging data showing that some LADs can also be found in the vicinity of the nucleolus[14,15]. NAD-only, NAD/LAD, and LADs that do not overlap with NAD (LAD-only) have low gene density (Fig. 3d) and, consistent with previous reports, NADs are particularly enriched in olfactory receptor genes and zinc finger genes[19,20] (Supplementary Data 3). Both NAD sub-classes and LAD-only showed distinct chromatin and transcriptional features. A large portion of NAD-only regions localizes in the active A compartment (66%), is early replicating (65%), and has higher gene density relative to NAD/LAD and LAD-only sequences (Fig. 3d, f). However, NAD-only has a lower gene density relative to the whole genome and low level of gene expression compared to genes located in the active A compartment, indicating that the localization at the nucleolus correlates with low gene activity (Fig. 3d, g). In contrast to NAD-only, the majority of NAD/LAD and LAD-only are located in the repressive B compartment (90% and 80%) and are late replicating (82% and 63%), a result that is consistent with previous data showing that total LADs are characterized by these repressive features[9,10,36] (Fig. 3d–f). However, compared to LAD-only, NAD/LAD appear to be more enriched in the B compartment and in late-replicating DNA regions and display lower gene density and gene expression levels (Fig. 3d–g), suggesting that sequences that can localize at both nucleolus and NL have enhanced repressive chromatin features than sequences that anchor only to NL. These results were also supported by the different levels of active and repressive histone marks at these genomic domains (Fig. 3h). As expected, NAD-only sequences have higher levels of the active histone modifications H3K4me3, H3K27ac, and H3K4me1 compared to LAD-only and NAD/LAD. However, and consistent with the low gene expression (Fig. 3g), NAD-only contains lower levels of active histone marks compared to genomic regions within the A active compartment. Furthermore, relative to A compartment, NAD-only is depleted of H3K27me3 and enriched in H3K9me2 (Fig. 3h), suggesting that the localization close to the nucleolus marks transcriptionally repressive states that might not depend on Polycomb. We also

observed that NAD/LAD and LAD-only regions show distinct chromatin features. NAD/LAD displays a more repressive chromatin state than LAD-only. They contain a lower amount of active histone marks and, in particular, higher levels of H3K9me2 whereas H3K9me3 are similar (Fig. 3h). Furthermore, the levels of the facultative heterochromatin mark H3K27me3 are lower in NAD/LAD than in LAD-only, indicating that contacts with the nucleolus, even for sequences able to interact with the NL, shape a unique repressive chromatin state that is characterized by the enrichment in H3K9me2 and low H3K27me3 content. These results indicate that genomic regions display a distinct repressive chromatin composition according to their ability to localize at the nucleolus, NL, or both.

As in the case of LADs, both NAD-only and NAD/LAD are characterized by abrupt borders, which display a sharp transition in several chromatin features (Fig. 3i). The average signal of active histone marks, CTCF and early replicating DNA sharply decrease at the border of both NAD subclasses and LAD-only. These results further indicate that contacts with the nucleolus demarcate less active chromatin domains. In contrast to LADs, which were described to be enriched in H3K27me3 and Polycomb components Ezh2 and Ring1b near LAD borders[37], NAD-only and NAD/LAD display a drastic decrease of these factors at their corresponding borders (Fig. 3i). H3K9me2 levels and late DNA replication signals sharply increase at the borders of both NAD sub-classes and LAD-only whereas H3K9me3 does not show this trend. Thus, the identification of NADs with the Nucleolar-DamID allowed us to distinguish different layers of genome compartmentalization by defining regions that are exclusively localized at nucleoli, NL, or both. Further, the results revealed that NADs correspond to regions of the genome with a repressive state that is poor in H3K27me3 and specifically enriched in H3K9me2.

**The chromatin state of rRNA genes regulates H3K9me2 levels at sequences adjacent to the nucleolus.** Previous work has shown that the chromatin state of rRNA genes could affect chromatin structures outside the nucleolus[38]. In ESCs, all rRNA genes are euchromatic due to the impairment of processing of the long non-coding IGS-rRNA into mature pRNA, which is required for the recruitment of the repressive nucleolar remodeling complex NoRC to rRNA genes[38–41]. The active state of rRNA genes in ESCs could be reversed by the addition of mature pRNA that causes NoRC recruitment and consequent formation of heterochromatin at rRNA genes, including the increase in H3K9me2 and H3K9me3 (Fig. 4a, b)[42,43]. This heterochromatinization was

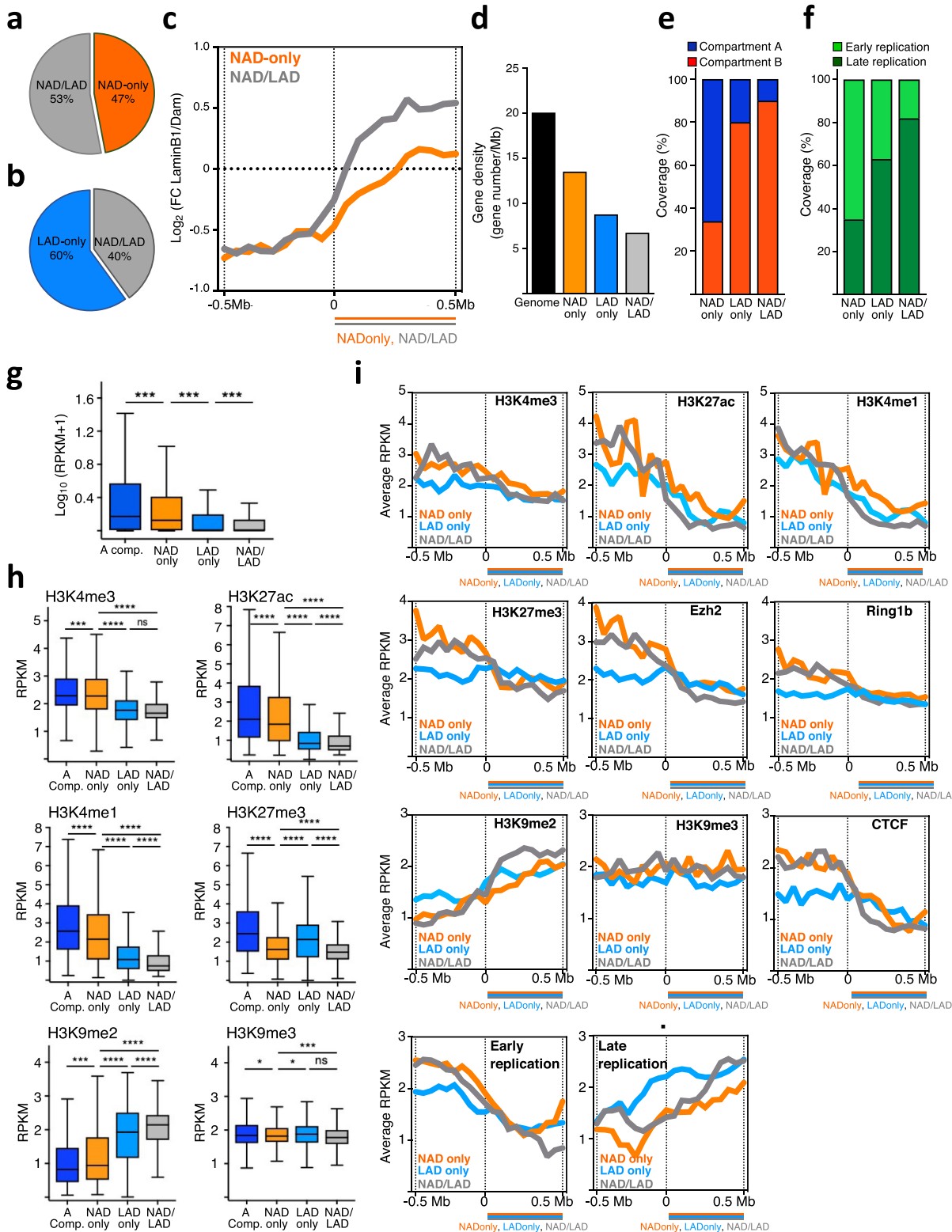

not only limited within the nucleolus but it also extended to other regions of the genome outside the nucleolus, such as minor satellites, highlighting a crosstalk between nucleolar and nuclear chromatin (Fig. 4b). The identification of NADs in ESCs prompted us to identify which other genomic regions are affected by the chromatin state of rRNA genes and their relationship with the nucleolus. We measured H3K9me2 and H3K9me3 by ChIPseq of ESCs transfected with mature pRNA (Fig. 4c,

Supplementary Fig. 6). We found that H3K9me2 levels increase at several genomic regions all over the chromosomes of ESC + pRNA compared to control cells whereas H3K9me3 levels are not affected (Fig. 4c, d). We observed a sharp increase in H3K9me2 at regions adjacent to the border of NADs but not of LADs, indicating an expansion of H3K9me2 domains specifically at regions neighboring NADs (Fig. 4e, f). These results revealed that thenucleolus is not only a scaffold where repressive chromatin

**Fig. 3 Distinct layers of repressive chromatin states distinguish genomic domains according to their interaction with the nucleolus, nuclear lamina, or both. a** Venn diagram showing the proportion of NAD-only and NAD/LAD regions in NADs identified by Nucleolar-DamID. **b** Venn diagram showing the proportion of LAD-only and NAD/LAD regions in LADs of ESCs. **c** Lamin B1-DamID scores plotted over NAD-only and NAD/LAD boundaries in ESCs. **d** Gene density of total genome, NAD subclasses, and LAD-only. **e** Amounts (%) of NAD-only, LAD-only and NAD/LAD in A and B compartments. **f** Amounts (%) of early and late replicating DNA of NAD-only, LAD-only, and NAD/LAD sequences. **g** Expression values (RPKM) of genes within A compartment (A Comp.), NAD-only, LAD-only, and NAD/LAD. Statistical significance (P-values) was calculated using the unpaired two-tailed *t* test (***<0.001). Box plots depict the minimum and maximum values. The horizontal line within the boxes represents the mean value. **h** Levels of active histone marks (H3K4me3, H3K27ac, H3K4me1) and repressive histone marks (H3K27me3, H3K9me2, H3K9me3) at genomic regions located at the A compartment (A Comp.) and NAD-only, LAD-only, and NAD/LAD regions. Dataset used in this analysis is listed in Supplementary Data 12. Values are shown as average RPKM. Statistical significance (P-values) was calculated using the unpaired two-tailed *t* test (*<0.05, ***<0.001, ****<0.0001, ns: non-significant). Box plots depict the minimum and maximum values. The horizontal line within the boxes represents the mean value. i. Occupancy (average RPKM) of histone modifications, Ezh2, Ring1b, CTCF and early and late-DNA replication plotted over the boundaries of NAD-only (orange lane), NAD/LAD (blue line), and NAD/LAD (gray line), respectively. Source data are provided as a Source Data file.

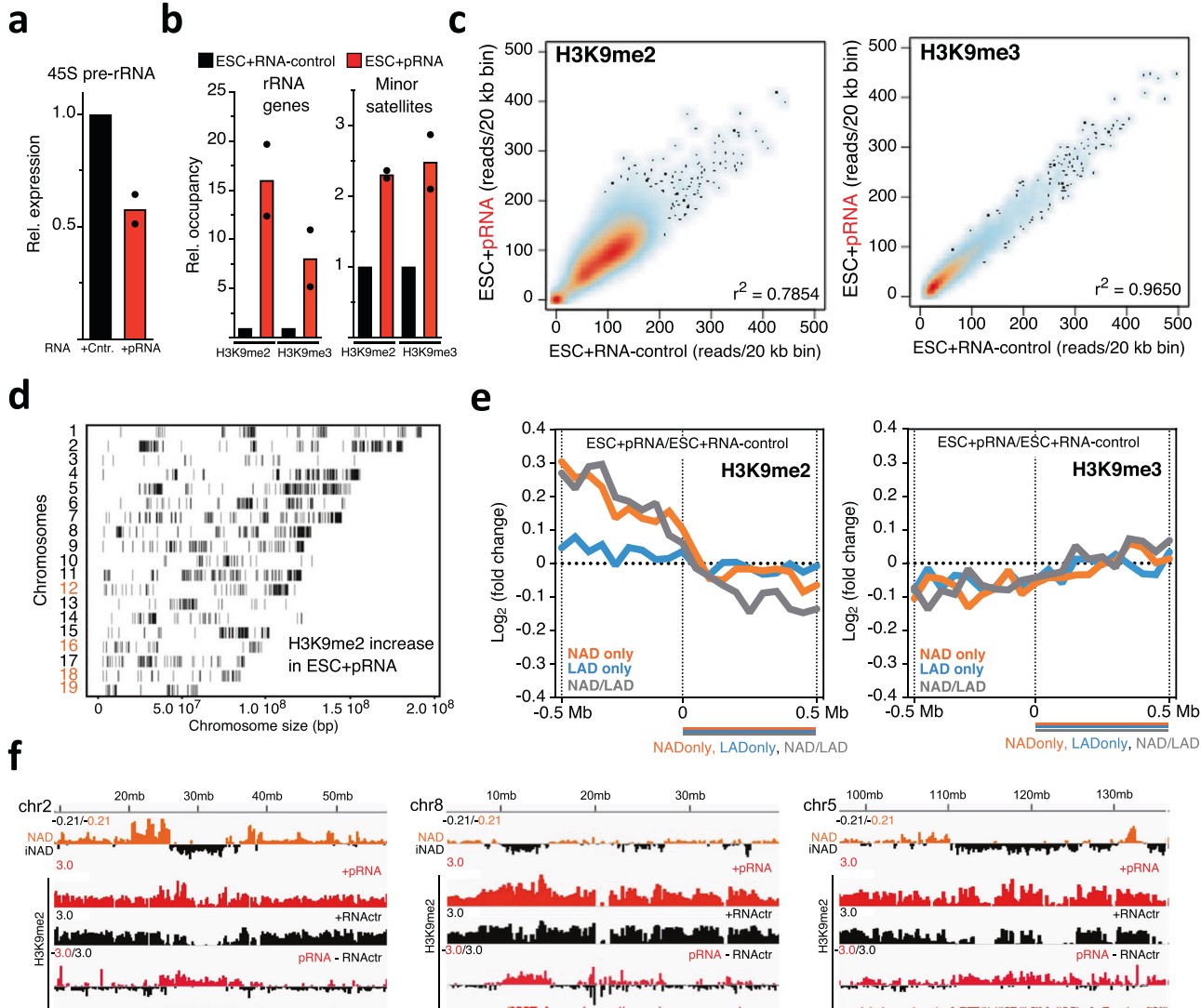

**Fig. 4 The chromatin state of rRNA genes regulates H3K9me2 levels at sequences adjacent to the nucleolus. a** qRT-PCR showing 45 S pre-rRNA levels in ESCs transfected with pRNA or RNA-control. Values were normalized to *β-actin* mRNA and to ESCs transfected with RNA-control. Data are from two independent experiments. **b** H3K9me2 and H3K9me3 ChIP in ESCs transfected with pRNA or RNA-control. Data were measured by qPCR and normalized to input and ESC + RNA-control. Data are from two independent experiments. **c** Addition of pRNA in ESCs caused an increase in H3K9me2 at several genomic regions. Scatter plot showing H3K9me2 and H3K9me3 levels (reads/20 kb bin) between ESC + pRNA and ESC + RNA-control. **d** Chromosomal interaction map showing the distribution of regions with increased H3K9me2 levels in ESC + pRNA compared to ESC + RNA-control. **e** Heterochromatinization of rRNA genes promotes H3K9me2 expansion at regions neighboring NADs. H3K9me2 fold changes in ESC + pRNA vs. ESC + RNA-control plotted over the boundaries of NAD-only, LAD-only, and NAD/LAD. **f** Representative images showing the increase of H3K9me2 at regions neighboring NADs. Source data are provided as a Source Data file.

can be positioned but also it is part of the regulatory process for the establishment or maintenance of repressive chromatin states.

**Genome organization around the nucleolus during ESC differentiation.** To determine whether NAD composition could change according to developmental stage, we performed Nucleolar-DamID and HiC-rDNA analyses in neural progenitor cells (NPCs) derived upon differentiation of ESCs using established protocols[44]. We validated NADs mapped in NPC by DNA-FISH (Supplementary Fig. 7a–g). Both Nucleolar-DamID and HiC-rDNA revealed a common, global chromosome architecture between ESCs and NPCs, with substantial overlapping interactions with the nucleolus (Fig. 5a, b, Supplementary Fig. 4, S8a, and Supplementary Data 1, 2, 4). However, there were remarkable differences in the organization of the chromosomes around the nucleolus during ESC differentiation into NPCs. Both Nucleolar-DamID and HiC-rDNA revealed that in all chromosomes contacts with the nucleolus are more frequent in NPCs than in ESCs (Fig. 5c, d, Supplementary Fig. 4, 8b). However, NAD coverage and the number of rDNA contacts were higher in ESCs than in NPCs (Fig. 5e, Supplementary Fig. 8c). Furthermore, a large fraction of rDNA contacts in NPCs were located at chromosomes bearing rRNA genes whereas in ESCs the distribution of contacts was more homogeneous among the chromosomes compared to NPCs. Finally, rDNA contacts in NPCs were more enriched at the centromere-proximal regions of all chromosomes than in ESCs (Fig. 2f, 5f). These results suggest that the architecture of chromosomes surrounding the nucleolus in NPCs is in a more compact, rigid form, making fewer but more frequent contacts with rRNA genes. In contrast, the structure of chromosomes around the nucleolus of ESCs appears more flexible, establishing more but less frequent rDNA contacts. These data are consistent with previous reports showing that ESCs harbor a more open and dynamic chromatin than differentiated cells[45,46]. We also observed that NAD/LAD and LAD-only in NPCs also show abrupt borders with a sharp transition in active and repressive histone marks whereas NAD-only in NPCs display this sharp transition only for H3K27ac and H3K9me2 and shows at the border more active chromatin features than in ESCs (Supplementary Fig. 8d).

Next, we investigated for the presence of genomic contacts with the nucleolus that are specific for either ESCs or NPCs. We identified large domains that contain rDNA contacts exclusively in ESCs or NPCs (here after referred as ESC$_{sp}$- or NPC$_{sp}$-rDNA contacts; average length 0.51 Mb and 0.23 Mb, respectively; Fig. 5g, Supplementary Data 5). We validated this cell-type-specific contacts with the nucleolus by DNA-FISH (Supplementary Fig. 7d–g). The large majority (>80%) of ESC$_{sp}$- and NPC$_{sp}$-rDNA contacts were located in the repressive B compartment (Fig. 5h). Analysis of eingenvector values revealed that ESC$_{sp}$-rDNA contacts increase the interaction strength within the active A compartment in NPCs whereas contacts in the repressive B compartment decrease (Fig. 5g–j). 37% of ESC$_{sp}$-rDNA contacts switched from B to A compartment in NPCs whereas ESC$_{sp}$-rDNA contacts that remain in B (47%) or A compartments (16%) in NPCs significantly increased their eigenvector values in NPC (Fig. 5g–j). Similar results were observed for NPC$_{sp}$-rDNA contacts in ESCs. Thus, rDNA contacts specific to ESCs or NPCs correspond to cell-type specific repressive features of the genome organization.

Since rDNA contacts represent a sub-group of NADs, we extended our analyses to all NADs defined by Nucleolar-DamID. We identified NADs specific for ESCs or NPCs (here after defined as ESC$_{sp}$-NAD and NPC$_{sp}$-NAD; Supplementary Data 6). We found that ESCs have more cell-type-specific NADs than NPCs

(16% and 8% coverage relative to the corresponding total NADs). Remarkably, the large majority of ESC$_{sp}$-NAD-only (78%) did not associate with the NL upon detachment from the nucleolus in NPCs and became iNAD/iLAD. In contrast, 66% of ESC$_{sp}$-NAD/LAD interacted with the NL in NPC (Fig. 6a). Similar changes were also observed with NPC$_{sp}$-NAD in ESCs. These results suggest that a NAD associating with the nucleolus but not with the NL in a cell type cannot associate with the NL upon detachment from the nucleolus in the other cell type. On the other hand, a NAD that can associate with both the nucleolus and NL can interact with the NL upon dissociation from the nucleolus. We found similar results for ESC$_{sp}$-LAD and NPC$_{sp}$-LAD (Fig. 6b). 91% of ESC$_{sp}$-LAD-only in NPCs and 81% of NPC$_{sp}$-LAD-only in ESCs became iNAD/iLAD. On the other hand, 71% of ESC$_{sp}$-NAD/LAD in NPCs and 83% of NPC$_{sp}$-NAD/LAD in ESCs associated with the nucleolus. These results suggest that the position of a NAD or a LAD relative to the nucleolus or NL in a cell type can influence their cellular location upon dissociation from the nucleolus or NL in the other cell type. To determine whether these properties could be linked to chromatin modifications, we compared histone modification levels between ESC$_{sp}$-NAD-only and ESC$_{sp}$-NAD/LAD that in NPCs became LAD-only or iNAD/iLAD (Fig. 6c). We found that ESC$_{sp}$-NAD becoming iNAD/iLAD in NPCs are significantly enriched in active histone modifications H3K27ac, H3K4me3, and H3K4me1 and the repressive H3K27me3 compared to ESC$_{sp}$-NAD becoming LAD-only in NPCs. In contrast, ESC$_{sp}$-NAD becoming LAD-only in NPCs were more enriched in H3K9me2 compared to ESCsp-NAD becoming iNAD/iLAD in NPC (Fig. 6c). Thus, the state of histone modifications of ESC$_{sp}$-NAD in ESCs seems to influence the cellular location of a NAD upon detachment of the nucleolus in NPCs. The analysis of LADs revealed that ESC$_{sp}$-LAD enriched in H3K27me3 in ESCs have the propensity to not interact with the nucleolus in NPCs whereas all the other examined histone modifications did not show any correlation for the cellular localization in NPCs (Fig. 6d). These results are consistent with the data showing that NADs are depleted of H3K27me3 relative to active A compartment and LAD-only (Fig. 3h, i) and further suggest that the nucleolus is refractory to H3K27me3 enriched chromatin.

Consistent with the analysis of rDNA contacts, ESC$_{sp}$-NAD in NPCs increased the interaction strength within the A compartment whereas contacts in the B compartment decreased (Fig. 6e, Supplementary Fig. 9a, b). A large fraction of ESC$_{sp}$-NAD (73%) becoming iNAD/iLAD in NPCs was located in the A compartment of ESCs whereas 89% of ESC$_{sp}$-NAD becoming LAD-only in NPCs were in the B compartment (Supplementary Fig. 9a). This correlation could also be observed in ESC$_{sp}$-LAD, although to a minor extent (Supplementary Fig. 9c). Remarkably, the eigenvector values of ESC$_{sp}$-NAD and ESC$_{sp}$-LAD respectively moving to the NL or nucleolus of NPCs were lower compared to ESC$_{sp}$-NAD and ESCsp-LAD that became iNAD/iLAD in NPCs (Fig. 6f). We found similar results for both NPC$_{sp}$-NAD and NPC$_{sp}$-LAD in ESCs. These results suggest that histone modifications and genome compartmentalization in a cell type can influence the location of a NAD or a LAD upon detachment from the nucleolus or NL in the other cell type.

**NAD detachment from the nucleolus unlocks genes for activation in later stages of differentiation.** Next, we analyzed whether the detachment of NADs from the nucleoli during the differentiation into NPCs correlates with changes in gene expression. First, we asked whether changes in rDNA contacts between ESCs and NPCs correspond to gene expression changes. We performed RNAseq of ESCs and NPCs and found that the

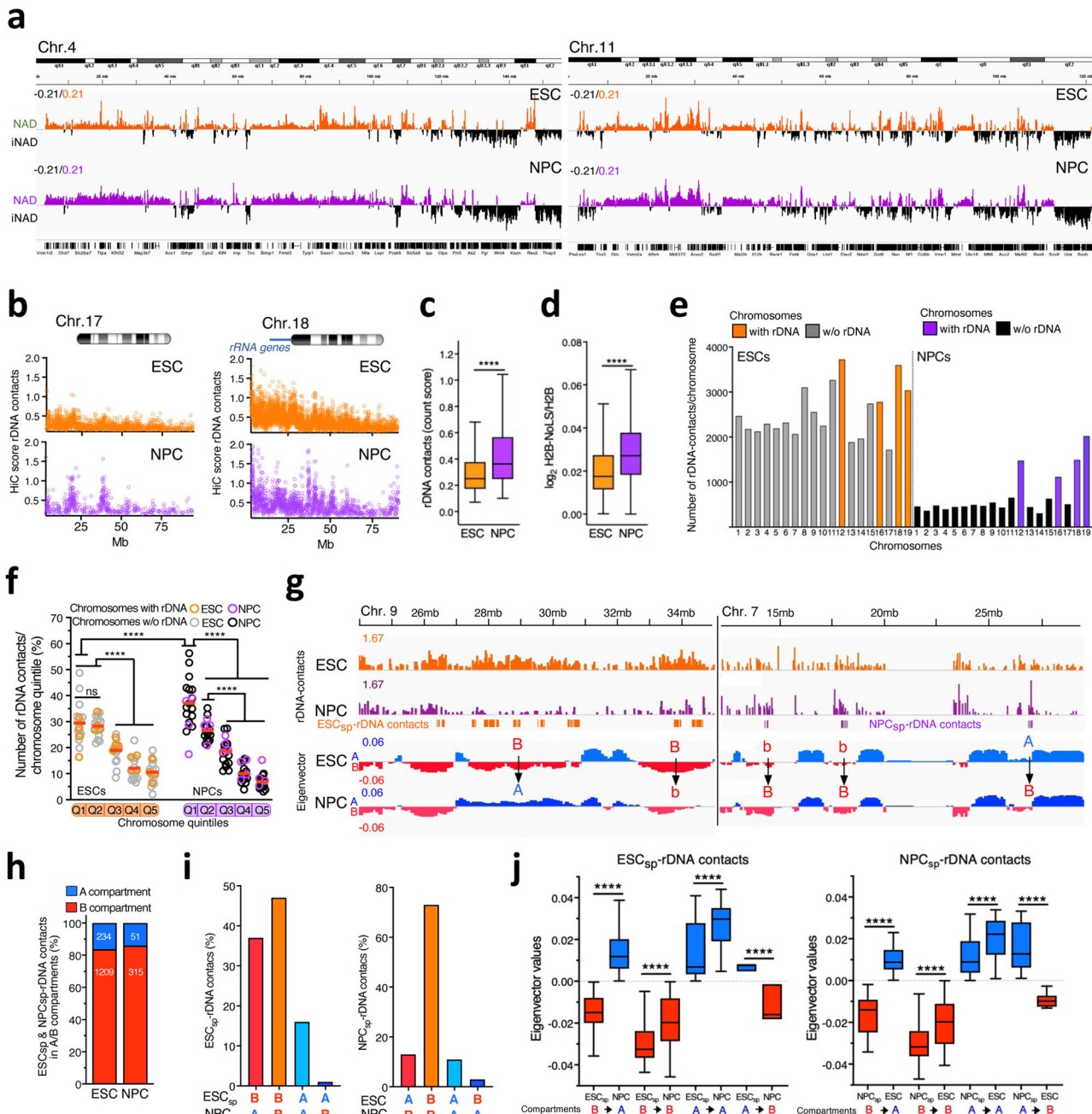

**Fig. 5 ESC and NPC differ in their chromosome organization around the nucleolus. a** Nucleolar-DamID. Chromosomal view of NADs in ESCs and NPCs. NADs are measured as $\log_2$ ratio of m6A levels between H2B-Dam-NoLS and H2B-Dam. iNAD: regions not contacting the nucleolus. iLAD: regions not contacting the NL. Chromosomes 4 and 11 are shown. **b** Representative images showing HiC-score of rDNA contacts at chromosome 18, which contains rRNA genes and chromosome 7. **c, d** Genomic contacts with the nucleolus are more frequent in NPCs than in ESCs. HiC-score of rDNA contacts (**c**) and Nucleolar-DamID values of NADs (**d**) in ESCs and NPCs. Statistical significance (P-values) was calculated using the unpaired two-tailed t test (****<0.0001). Box plots depict the minimum and maximum values. The horizontal line within the boxes represents the mean value. **e** ESCs have more rDNA contacts than NPC. Number of unique rDNA contacts at chromosomes in ESCs and NPCs. **f** rDNA contacts are enriched in the centromeric-proximal regions of chromosomes of NPCs relative to ESCs. To allow a better comparison, data of ESCs of Fig. 2f were plotted together with data of NPCs. Values represent the proportion of rDNA-contacts for each chromosome quintile. Statistical significance (P-values) was calculated using the unpaired two-tailed t test (****<0.0001). **g** Representative images of $ESC_{sp}$- and $NPC_{sp}$-rDNA contacts and their eigenvector values for A and B compartment. B to A and A to B represent switch of compartments. B to b and b to B indicate a decrease or increase of eigenvector values between ESCs and NPCs. **h** $ESC_{sp}$- and $NPC_{sp}$-rDNA contacts are in the repressive B compartment. Amounts (%) of $ESC_{sp}$- and $NPC_{sp}$-rDNA contacts in A and B compartments. **i** Values represent the number of $ESC_{sp}$- and $NPC_{sp}$-rDNA contacts and their corresponding location in A and B compartments of ESCs and NPCs. **j** Box plots showing eigenvector values of $ESC_{sp}$- and $NPC_{sp}$-rDNA contacts in the active A and repressive B compartments. Statistical significance (P-values) was calculated using the paired two-tailed t test (****<0.0001). Box plots depict the minimum and maximum values. The horizontal line within the boxes represents the mean value. Source data are provided as a Source Data file.

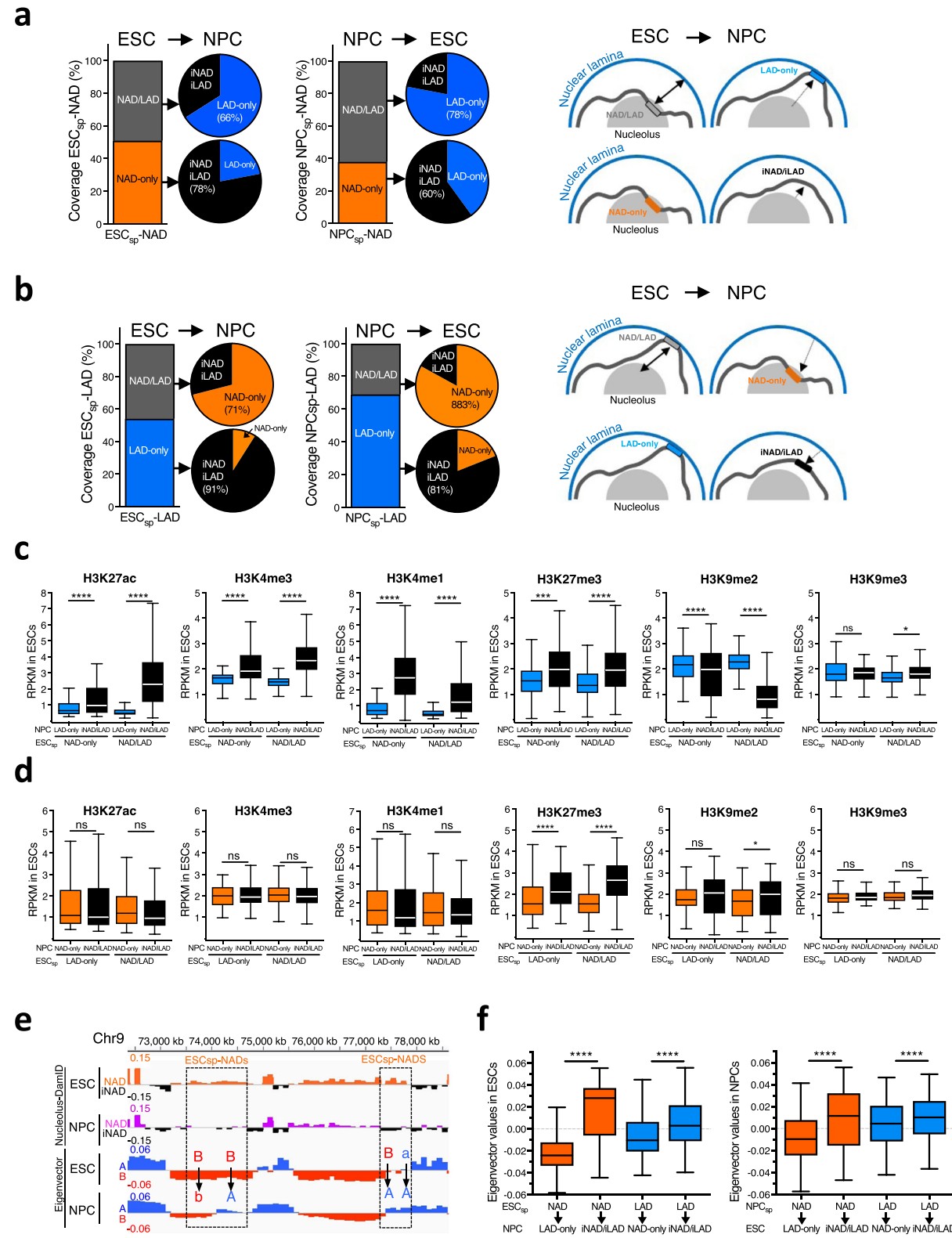

majority of genes located at ESC$_{sp}$- and NPC$_{sp}$-rDNA contacts are lowly expressed (<1 RPKM) (Fig. 7a, Supplementary Data 7, 8). We did not observe significant changes in their gene expression between ESCs and NPCs, indicating that the detachment from the nucleolus is not sufficient to reactivate gene expression. However, we found that the top 10 gene ontology (GO) terms for

genes located in ESC$_{sp}$-rDNA contacts were linked to pathways implicated in neuron development and differentiation (Fig. 7b, Supplementary Data 9), suggesting that the relocation of these genes away from the nucleolus might be a first step toward their activation in later stages of differentiation. On the other hand, genes located at NPC$_{sp}$-rDNA contacts were linked to pathways

**Fig. 6 Chromatin features of cell type specific NADs. a** Coverage of $ESC_{sp}$- and $NPC_{sp}$-NAD types in ESCs and NPCs. Venn diagrams represent the proportion of $ESC_{sp}$- and $NPC_{sp}$-NAD types as iNAD/iLAD and LAD-only in ESCs and NPCs. Right panel. Model showing the location of $ESC_{sp}$-NAD-only and -NAD/LAD in ESCs and in NPCs. Dotted arrows represent the relocation of $ESC_{sp}$-NAD that lost the interaction with the nucleolus in NPC. **b** Coverage of $ESC_{sp}$- and $NPC_{sp}$-LAD types in ESCs and NPCs. Venn diagrams represent the proportion of $ESC_{sp}$- and $NPC_{sp}$-LAD types as iNAD/iLAD and NAD-only in NPCs and ESCs. Right panel. Model showing the location of $ESC_{sp}$-LAD-only and -NAD/LAD in ESCs and in NPCs. Dotted arrows represent the relocation of a $ESC_{sp}$-LAD that lost the interaction with NL in NPC. **c** Levels of histone modifications at $ESC_{sp}$-NAD-only and -NAD/LAD in ESCs according to their location in NPCs. Values are average RPKM. Statistical significance (*P*-values) was calculated using the unpaired two-tailed *t* test (*<0.05, ***<0.001, ****<0.0001, ns: non-significant). Box plots depict the minimum and maximum values. The horizontal line within the boxes represents the mean value. **d** Levels of histone modifications at $ESC_{sp}$-LAD-only and -NAD/LAD in ESCs according to their location in NPCs. Values are average RPKM. Statistical significance (*P*-values) was calculated using the unpaired two-tailed *t* test (*<0.05, ****<0.0001, ns: non-significant). Box plots depict the minimum and maximum values. The horizontal line within the boxes represents the mean value. **e** Representative image of $ESC_{sp}$-NAD and their eigenvector values for A and B compartment in ESCs and NPCs. Arrows highlight changes in eigenvector values between ESCs and NPCs. B to A represents regions switching from B (ESCs) to A (NPCs) compartment. B to b and a to A indicate higher eigenvector values in ESCs compared to NPCs. **f** Box plots showing eigenvectors values of $ESC_{sp}$-NADs, $ESC_{sp}$-LADs, $NPC_{sp}$-NADs, and $NPC_{sp}$-LADs in ESCs and NPCs, respectively, that became LAD-only, NAD-only, and iNAD/iLAD in NPCs and ESCs. Statistical significance (*P*-values) was calculated using the unpaired two-tailed *t* test (****<0.0001). Box plots depict the minimum and maximum values. The horizontal line within the boxes represents the mean value. Source data are provided as a Source Data file.

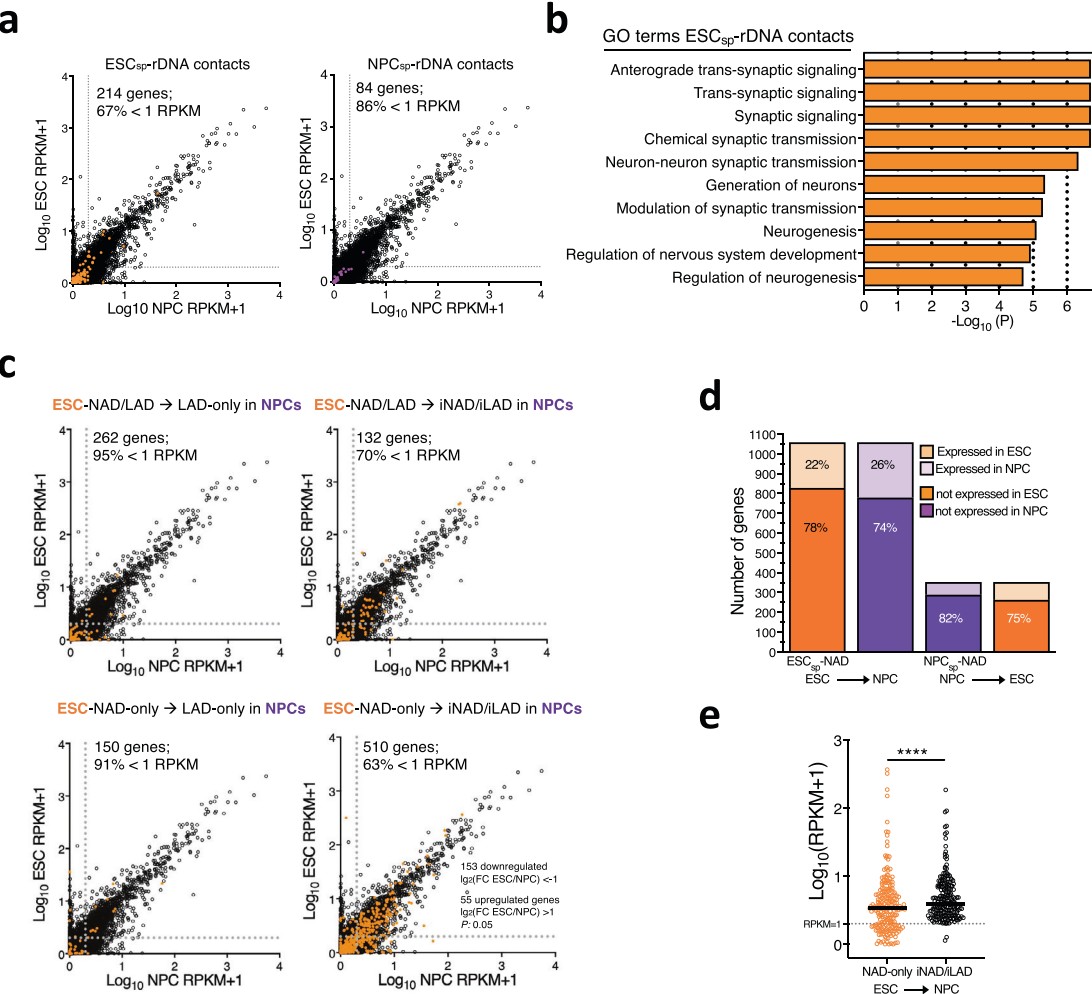

**Fig. 7 NAD detachment from the nucleolus unlock genes for activation in later stages of differentiation. a** Scatter plot showing expression levels between ESC and NPCs. Expression of genes located at $ESC_{sp}$- and $NPC_{sp}$-rDNA contacts are highlighted in orange and magenta, respectively, whereas total genes are represented in black. Dotted lines indicate RPKM value as 1. **b** Top 10 gene ontology terms of genes located at $ESC_{sp}$-rDNA contacts. **c** Scatter plot showing gene expression levels between ESC and NPCs. Expression of genes located at $ESC_{sp}$-NAD-only and $ESC_{sp}$-NAD/LAD that become LAD-only or iNAD/iLAD in NPCs. Genes located at $ESC_{sp}$-NAD are highlighted in orange, whereas total genes are represented in black. Dotted lines indicate RPKM value as 1. **d** Number of genes located at $ESC_{sp}$-NAD and $NPC_{sp}$-NAD. The proportion of low or not expressing (<1 RPKM) and expressing genes in ESCs and NPCs is indicated. **e** Expression levels of genes located at $ESC_{sp}$-NAD-only that become iNAD/iLAD in NPCs. Values are from genes that were expressed (> RPKM 1) in ESCs or NPCs. Statistical significance (*P*-values) was calculated using the paired two-tailed *t* test (****<0.0001). Source data are provided as a Source Data file.

of sensory perception of smell (mainly due to the presence of several olfactory receptor genes) and metabolic processes (Supplementary Data 9).

We performed a similar analysis for genes located at $ESC_{sp}$- and $NPC_{sp}$-NADs (1201 and 377 genes, respectively) and found that also in this case they were either not or low expressing in both cell types (74-82% of genes with RPKM < 1), showing in general no significant changes in gene expression between ESCs and NPCs (Fig. 7c, d, Supplementary Fig. 10a, Supplementary Data 7, 10). Consistent with previous results[9], genes located at $ESC_{sp}$-LAD were also low expressing and the majority of them did not show significant changes between ESCs and NPCs, indicating that loss of contacts with the NL, irrespective of the new location relative to the nucleolus, does not necessarily cause transcriptional activation (Supplementary Fig. 10b).

The only case where we found some changes in gene expression was for genes located in $ESC_{sp}$-NAD-only that became iNAD/iLAD in NPCs. A large fraction of genes expressed in ESCs or in NPCs displayed a modest but significantly lower expression in ESCs relative to NPCs (RPMK > 1, 221 out of 500 genes, Fig. 7c, e). Remarkably, independently of gene expression levels, these genes showed significant enrichment in many pathways linked to neuron development and differentiation (Supplementary Fig. 10c, Supplementary Data 11), suggesting that their detachment from both repressive nucleolar and NL compartments in NPCs might unlock them for activation in a next step of the differentiation process. Accordingly, genes in $ESC_{sp}$-NAD that repositioned exclusively at NL (LAD-only) in NPCs were mainly enriched in pathways linked to keratinocyte and epidermal differentiation (Supplementary Fig. 10d, Supplementary Data 11), which should not be activated later in neurogenesis and thus remained in contact with a repressive compartment such as the NL. These results further support a role of the nucleolus as repressive compartment that is implicated in the control of gene expression program during lineage commitment.

## Discussion

The nucleolus is the largest compartment of the eukaryotic cell's nucleus, known to act as a ribosome factory, thereby sustaining the translation machinery for protein synthesis[47]. Increasing evidence indicates that the role of the nucleolus and rRNA genes might go beyond the control of ribosome biogenesis. One such role is linked to the organization of the genome since repressive chromatin domains can often be found in the vicinity of the nucleolus. However, this aspect of genome compartmentalization in the cell's nucleus has so far remained under-investigated due to technical challenges in the identification of NADs.

In this work, we have established methods that allow accurate genome-wide identification of NADs. While Nucleolar-DamID has identified NADs using as readout DNA interactions with the engineered nucleolar histone H2B, HiC-rDNA has detected a subclass of NADs that interact with the nucleolar rRNA genes. Although these methods were based on completely different methodologies, they have both revealed similar features of chromosome organization around the nucleolus in both ESCs and NPCs, such as the low gene density and low expression state, the enrichment on NADs at the proximal centromeric regions and at chromosomes containing rRNA genes, and the increase in contact frequency with the nucleolus in NPCs relative to ESCs. Previous methods for NAD identification have used sonication-based biochemical purification of nucleoli, a methodology that is subjected to a certain variation in nucleoli preparation between different cell types and relatively biased toward sonication-resistant heterochromatin[19–21,23,24,48]. Relative to this methodology, the

application of Nucleolar-DamID and HiC-rDNA has no bias for any kind of chromatin state and does not vary between cell types, thereby allowing direct comparisons of chromosome architecture around the nucleolus between distinct cell states. The accuracy of our method in mapping NADs is further supported by DNA-FISH analyses that validated the high-frequency contacts with the nucleolus of NADs identified by the Nucleolar-DamID but that were not detected by previous studies using nucleolar purification-based methods in ESCs[23,24]. Thus, NAD maps provided in this work can be integrated into future studies of 3D genome organization that so far did not take into account the nucleolus because of the lack of accurate genome-wide NAD profiles.

We have found some similarities between NAD maps obtained with previous nucleoli purifications and our methodologies, such as repressive chromatin features, the enrichment in olfactory receptor and zinc finger genes, and a certain overlap with LADs[19,20,24,48]. Similarly, a previous analysis using HiC data from human cells has showed that contacts with rRNA genes correspond to repressive domains[49]. Further, nucleolar purifications from ESCs have also found that NAD-only has higher gene density and gene expression levels than NAD-LAD[24]. It is important here to note that major and minor satellites, which compose centric and pericentric domains and are known to be in contact with the nucleolus, could not be identified by Nucleolar-DamID due to the lack of GATC motif. However, Nucleolar-DamID and HiC-rDNA have revealed additional details of chromosome architecture around the nucleolus that have not emerged from previous studies. The analysis of NADs in ESCs has revealed that 50% of NADs coincide with LADs, a result that is consistent with previous imaging data showing that chromosomal regions that were LADs in the mother cell could relocate close to the nucleolus after completion of cell division[14]. Our work has not only identified these LADs able to contact the nucleolus but has also determined that this LAD sub-type differs in the chromatin composition compared to LADs that do not contact the nucleolus (LAD-only). Thus, the identification of NADs by Nucleolar-DamID has revealed additional layers of repressive compartments, showing distinct chromatin features between genomic domains associating with the nucleolus, NL, or both. Indeed, although NADs share some similarities with known features of LADs, such as low overall gene density and low expression levels[8], they also display some distinct characteristic. In particular, NAD/LAD demarcate regions of the genome that are enriched in H3K9me2 and depleted in H3K27me3 relative to LAD-only. This was also evident by the depletion of Polycomb components Ezh2 and Ring1b at the borders of nucleolar contacts whereas LAD-only is enriched as previously reported[37]. Thus, contacts with the nucleolus, even for sequences able to interact with the NL, shape a distinct repressive chromatin state.

The enrichment of NADs for H3K9me2 was of particular interest since previous work has linked this modification with the chromatin state of rRNA genes[38]. Previous studies showed that the induction of repressive chromatin at rRNA genes, which in ESC are all active, causes global remodeling of ESC genome toward a more heterochromatic state, including the formation of highly condensed heterochromatic structures[38]. In this work, we have shown that the induction of repressive chromatin at rRNA genes increase H3K9me2 levels also at genomic regions adjacent to NADs whereas regions neighboring LAD-only are not affected. This link between nucleolar and nuclear chromatin states indicate that the nucleolus is not only a compartment able to act a scaffold for the location of repressive chromatin domains but it can also participate in establishing repressive chromatin features at regions neighboring the nucleolus.

The analysis of NAD composition during differentiation of ESCs into NPCs has revealed a common, global chromosome architecture around the nucleolus of both cell types. This result is consistent with the similar composition of LADs in ESCs, NPCs, and terminally differentiated astrocytes[9], indicating that the chromosome architecture at the nuclear lamina and the nucleolus is globally conserved between different cell types. This similarity could not be found in previous NAD analyses using nucleoli purified from ESCs and MEFs that showed only 40% common NAD sequences (Supplementary Fig. 1)[24,48]. We have also detected some remarkable differences in NAD composition between ESCs and NPCs. Compared to ESCs, the structures of all chromosomes around the nucleolus of NPCs appears more compact and rigid, displaying fewer unique contacts but with higher interaction frequency, in particular at centromere-proximal regions. These results are consistent with previous reports showing that ESCs harbor a more open and dynamic chromatin than differentiated cells[45,46]. Further, we have identified cell-type-specific NADs, unique to ESCs or NPCs, and showed that the position of a NAD or a LAD relative to the nucleolus or NL in ESCs correlates with their cellular location upon dissociation from the nucleolus or NL during differentiation. In other words, a NAD that interacts with the nucleolus but not with the NL in ESCs, after losing contacts with the nucleolus upon differentiation, still maintains features that makes it refractory to associate with the NL. We have obtained similar results for ESC$_{sp}$-LADs. We have showed that these properties can be attributed to chromatin modifications and the interaction strength within the A or B compartments. For example, a ESC$_{sp}$-NAD with high H3K9me2 content in ESCs will preferentially contact the NL in NPCs. In contrast, a LAD that is enriched in H3K27me3 in ESCs will rarely contact the nucleolus, a further indication that the nucleolus is refractory to H3K27me3 enriched chromatin. Thus, chromatin feature of NADs and LADs in one cell type might set the choice to associate with the NL or the nucleolus upon dissociation from their respective compartments in the other cell type.

Our results have indicated that loss of contacts with the nucleolus does not generally correspond to gene activation. Considering that genes located at NADs in ESCs and that become iNAD/iLAD in NPCs were highly related to neurogenesis, it appears that the release from the repressive nucleolus compartment might unlock these genes for activation in a later stage of differentiation. A similar observation was also reported for LADs[9]. Consistent with these results, genes contacting the nucleolus of ESCs and moving in NPCs to the NL, the other repressive compartment, were related to pathways such as keratinocyte differentiation that should not be activated during neurogenesis.

In summary, our work has established methodologies to identify NADs in every cell type and allowed us to finally distinguish distinct layers of chromatin organization that depend on the interaction with the nucleolus, NL, or both. We predict that the application of Nucleolar-DamID and HiC-rDNA will be relevant for future works, including the understanding of the determinants for the specific targeting to nucleolus or NL or the application of single-cell analyses that were prohibitive with the previous biochemical based-methods. Further, considering that structural changes in the nucleolus are often observed in cancer and premature ageing[50–52], the identification and analysis of NAD composition will be important to understand the alterations of 3D genome organization in disease linked to nucleolus alterations. Finally, the identification of NADs by Nucleolar-DamID will feed the study of genome organization and provide insights into basic principles of genome organization and its role in gene expression and cell function.

## Methods

**Cell culture.** One hundred and twenty-nine mouse embryonic stem cells (E14 line) were cultured in either 2i media composed of DMEM-F12 and Neurobasal medium (1:1, Life Technologies), supplemented with 1× N2/B27 (Life Technologies), 1× penicillin/streptomycin/l-glutamine (Life Technologies), 50 μM β-mercaptoethanol (Life Technologies), recombinant leukemia inhibitory factor, LIF (Polygene, 1,000 U/ml) and MEK and GSK3β inhibitors, 2i (Sigma CHIR99021 and PD0325901, 3 and 1 μM, respectively). ESCs were seeded at a density of 50,000 cells/cm² in culture dishes (Corning® CellBIND® surface) coated with 0.1% gelatin without a feeder layer. Propagation of cells was carried out every 2 days using enzymatic cell dissociation.

NIH3T3 and HEK 293T cells were cultured in Dulbecco's modified Eagle's medium (DMEM, Life Technologies) supplemented with 10% fetal calf serum (FCS, Biowaste) and 1% penicillin/streptomycin (Life Technologies).

Neural progenitor cells were generated from ESCs, according to a previously established protocol[44]. In brief, differentiation used a suspension-based embryoid bodies formation (Bacteriological Petri Dishes, Bio-one with vents, Greiner). The neural differentiation media (DMEM, 10% fetal calf serum, 1× MEM NEAA, 2 mM Pen/Strep, β-mercaptoethanol, and sodium pyruvate) was filtered through 0.22 μm filters and stored at 4 °C. During the 8-day differentiation procedure, media was exchanged every 2 days. In the last 4 days of differentiation, the media was supplemented with 2 μM retinoic acid to generate neural precursors that are Pax-6-positive radial glial cells.

**Establishment of ESC lines for Nucleolar-DamID.** CRISPR/Cas9 cloning and targeting strategy were performed as previously described[53]. ESCs were co-transfected with a plasmid expressing the Cas9 proteins and the sgRNA guide sequence targeting the *Rosa26* locus (Genome CRISPR™ mouse ROSA26 safe harbor gene knock-in kit, SH054, GeneCopoeia) and the HDR repair template plasmid containing either H2B-Dam or H2B-Dam-NoLS constructs flanked by the homology arms with a molar ratio of 1:3. Two days after transfection, ESCs were selected using 2 μg of Puromycin (Life Technologies) overnight. After recover, ESCs were further treated with 1 μg of Puromycin (Life Technologies) for three days. After additional three days of recover, cells were seeded for single cell clone isolation. Resistant ESC clones were genotyped by PCR using primers able to distinguish between insertions of the construct in one or both alleles (Supplementary Table 1).

**Transfections.** $1.5 \times 10^5$ NIH3T3 cells were plated in 6-well plates and transfected, respectively, with 200 ng of plasmid expressing H2B-GFP or H2B-GFP-NoLS under the control of minimal CMV promoter, or with 1 μg of plasmid expressing TTFI-GFP under the full CMV promoter using Transit-X2 transfection reagent (Mirus) in Opti-MEM GlutaMAX reduced-serum medium (Life Technologies). When indicated, 24 h post-transfection, cells were treated with 50 ng of Actinomycin D (Sigma) and the other half were refreshed with new media. 24 h later, cells were fixed and mounted with DAPI-mounting media (Vector) for imaging.

$2 \times 10^5$ HEK 293T cells were plated in 6-well plates. The day after, cells were transfected with 500 ng of plasmid containing the sequence *hsp-TetO-GFP-Dam-DD-EF1a-puro-T2A-TetR-KRAB* using calcium phosphate protocol. 24 h post-transfection, part of the cells were induced with 1 μg/mL doxycycline and 1 μM Shield1, and the remaining cells were refreshed with new media. The day after the treatment, cells were imaged under the microscope.

$9 \times 10^5$ ESC were plated in gelatin-coated 10 cm culture dish and transfected with 42 μg pRNA, or RNA-control using Lipofectamine MessengerMAX reagent (Invitrogen) in Opti-MEM GlutaMAX (Life Technologies) reduced-serum medium. 48 h post-transfection, ESCs were collected for downstream analyses.

**Immunofluorescence.** Live cell images of NIH3T3 cells (Fig. 1b) and HEK 293T cells (Supplementary Fig. 1b) were taken and digitally recorded using FLoid Cell Imaging Station (ThermoFisher). In the experiments shown in Fig. 1c, e, cells were grown on glass coverslips, fixed with 4% paraformaldehyde, and stained with nucleolar marker NPM1 (Sigma, B0556) or Fibrillarin (CellSignaling, 2639) and DAPI (Vectashield Dapi mounting media, Vector, H-1200). Images were taken and digitally recorded using Leica DMI6000 B microscope. Antibodies used for IF are listed in Supplementary Table 2.

**Chromatin fractionation.** 48 h post-transfection, NIH3T3 cells were collected by trypsinization, washed once with PBS and counted. ES cell pellets were resuspended at a concentration of 10mio cells/ml in chromatin fractionation buffer (10 mM Hepes pH 7.6, 150 mM NaCl, 3 mM MgCl₂, 0.5% Triton X-100, 1 mM DTT freshly supplemented with cOmplete™ Protease Inhibitor Cocktail (Roche)) and incubated for 30 min at room temperature rotating. Precipitated chromatin was fractionated by centrifugation. Total and chromatin fractionated samples were further processed by MNase (S7 Micrococcal nuclease, Roche) digest for ensuring sufficient genomic DNA fragmentation. All samples were incubated in 1x Laemmli buffer (10% glycerol, 10 mM Tris pH 6.8, 2% SDS, 0.1 mg/ml bromophenolblue, 2% β-mercaptoethanol) at 95 °C for 5 min and were further analyzed by Western Blotting.

**DNA-FISH**. DNA fluorescence in situ hybridization (DNA-FISH) for chromosomes 19 and 14 were performed with Agilent SureFISH DNA-FISH probes that were designed by Agilent technologies using their standard procedures against genomic regions defined in Supplementary Table 3. ESCs were cultured on matrigel coated coverslips. The coverslips were fixed using 3.7% Formaldehyde for 10 min at room temperature. Cells were washed twice with 1X PBS and then dehydrated with graded ethanol concentrations up to 100% ethanol and air dried. The coverslips were incubated with 10 µL mixture of a custom probe set targeting a selected DNA locus (Agilent) and SureFISH Hybridization Buffer (Agilent, G9400A) with turned cell-side down. The coverslips and probe mixture were denatured for 8 min at 83 °C, then incubated at 37 °C overnight in a dark humidified chamber. Next day, coverslips were washed with FISH Wash Buffer 1 (Agilent, G9401A) at 73 °C for 2 min on a shaking incubator at 300 rpm, and FISH Wash Buffer 2 (Agilent, G9402A) at room temperature for 1 min. Following DNA-FISH probe hybridization, immunofluorescence was performed. Coverslips were prepared for immunofluorescence by rehydration and suspension in 1X PBS. Coverslips were permeabilized with 0.5% Triton-X in 1X PBS on ice for 5 min followed by incubation at room temperature for 10 min and kept in blocking buffer (1%BSA in 1XPBS-Tween 20 (0.1%) at room temperature for 1 hr. Coverslips were then incubated with primary anti-Nucleolin antibodies (Abcam; ab22758; 1:100) in a humidified chamber overnight at 4°C. After washing with 0.1% Triton-X in 1XPBS at room temperature, they were incubated with secondary antibodies in a dark humidified chamber at room temperature for 1.5 h. After washing with 1X PBS buffer, they were stained with Hoechst 33342 followed by mounting. The secondary antibodies used for IF were goat anti-rabbit IgG (H + L) highly cross-adsorbed secondary antibody, Alexa Fluor 488 (Thermo Fisher Scientific; A11034; 1:500) and goat anti-rabbit IgG (H + L) highly cross-adsorbed secondary antibody Alexa Fluor 546 (Thermo Fischer Scientific; A11035; 1:500).

DNA-FISH probes for chromosomes 1, 2, 4, 5, and 19 were generated with oligopaint libraries that were constructed the PaintSHOP interface created by the Beliveau lab (https://oligo.shinyapps.io/paintshop/_w_33571817/#tab-1201-8)[54] and were ordered from CustomArray/Genscript in the 12 K Oligopool format. Each library contains a universal primer pair used to amplify all the probes in the library, followed by a specific primer pair hooked to the 40–46-mer genomic sequences, for a total probe of around 124-130-mers. Oligopaint libraries were produced by emulsion PCR from the pool, followed by a "two-step PCR" and lambda exonuclease as described before[55]. Specifically, the emulsion PCR with the universal primers allowed amplifying all pool of probes in the library. The "two-step PCR" led to the addition of a tail to the specific probes bound to the Alexa Fluor 488 fluorochrome. All oligonucleotides were purchased from Integrated DNA Technologies (IDT, Leuven, Belgium). All oligonucleotide sequences and the mm10 coordinates for the probe libraries are listed in Supplementary Table 3. DNA-FISH with these probes was adapted from the protocol of Cavalli's lab[56]. ESCs were grown on matrigel coated chamber slide (Lab-Tek) for 2 days. In the case of NPCs, after 7 days of differentiation, embryoid bodies were trypsinized and seeded as single cells overnight on matrigel-coated chamber slides. Cells fixed with 4% PFA for 10 min at room temperature. Cells were washed three times with 1X PBS and then permeabilized with 0.5%Triton-X100/PBS for 10 min at room temperature. After three additional washes with 1X PBS, cells were incubated with 0.1 M HCl for 10 min at room temperature, washed twice with 2X SCCT and twice with 50% Formamide-2X SCCT. Probe mixture contains Xpmol of Oligopaint probe, 0,8uL of ribonuclease A (ThermoScientific, 10 mg/mL), ad FISH hybridization buffer for a total mixture volume of 20uL, added directly on the chamber slide. Cell DNA was denaturated at 83 °C for 8 min, and hybridization was performed in a humid dark chamber overnight at 42 °C. Cells were washed twice with 2X SCCT for 15 min, once with 0.2X SCC for 10 min, twice with 1X PBS for 2 min, and three times with 1X PBT for 2 min. Slides were then incubated for 1 h at room temperature with blocking solution in a dark humid chamber, and overnight at 4 °C with primary antibody against Nucleolin. Cells were washed four times with 1X PBT at increasing incubation times (1 × 2 min, 1 × 3 min, 2 × 5 min) and then incubated for 2 h at room temperature with goat anti-mouse IgG (H + L) Alexa Fluor 546 in a dark humid chamber. Slides were washed four times with 1X PBT at increasing incubation times (1 × 2 min, 1 × 3 min, 2 × 5 min), three times with 1X PBS for 2 min. Then, slides were mounted with Vectashield DAPI mounting media (Vector, H-1200) and, after drying, stored at 4 °C.

DNA FISH/IF samples were imaged using a Leica SP8 upright Microscope, with a z-stack collected for each channel (step size, 0.15 or 0.3 um, frame interval 1 s), using the oil objective HC PL APO CS2 63x/1.40. Images were processed by ImageJ (version 2.0.0/1.53c). The individual cells were identified by Hoechst/DAPI staining and cells containing signal for DNA-FISH channel were identified manually on the corresponding fluorescent channel. Distance between the DNA-FISH signal and the nucleolar marker immunofluorescence signal was calculated using ImageJ (version 2.0.0/1.53c) and used to count the number of foci contacting nucleolus and the number of cells with at least one contact with the nucleolus.

**RNA extraction, reverse transcription, and quantitative PCR (RT-qPCR)**. RNA was purified with TRIzol reagent (Life Technologies). 1 µg total RNA was primed with random hexamers and reverse-transcribed into cDNA using MultiScribe™

Reverse Transcriptase (Life Technologies). Amplification of samples without reverse transcriptase assured the absence of genomic or plasmid DNA. The relative transcription levels were determined by normalization to beta-Actin mRNA levels, as indicated. qRT-PCR was performed with KAPA SYBR® FAST (Sigma) on a Rotor-Gene Q (Qiagen). Primer sequences are listed in Supplementary Table 1.

**In vitro synthesis of pRNA**. Topo2.1 plasmids with insertion of pRNA and RNA-control sequences were previously described[38]. BamHI-linearized plasmids were in vitro transcribed with T7 RNA polymerase (Thermo Fisher EP0111). Synthesized RNA transcripts were verified by agarose gel electrophoresis and purified using NucleoSpin RNA II column (Machere-Nagel, cat. no. 740955) according to the manufacturer's protocol.

**DamIDseq**. $7 \times 10^4$ H2B-Dam and H2B-Dam-NoLS ESC lines were seeded in 6-well plates (Corning® CellBIND® surface) coated with 0.1% gelatin without a feeder layer. To induce the expression of Dam-fused proteins, two days after seeding, half of the cells were treated with 100 ng/ml Doxycycline and 1 µM Shield1 (Clontech Takara) for 15 h. Representative pictures of the cell colonies were taken from the Olympus CKX31F2 bright field microscope using a Canon EOS880D camera. For the analysis of NPCs, 8 days after differentiation, H2B-Dam and H2B-Dam-NoLS cells were harvested with 10X trypsin to get single cells from the embryoid bodies. $1 \times 10^6$ NPCs were seeded in 6-well plates coated with 0.1% gelatin and treated with 100 ng/ml Doxycycline and 1 µM Shield1 for 15 h. The day after, cells were harvested by trypsinization and DNA was extracted using Quick-DNA Miniprep Plus kit (Zymo Research). To test the efficiency and the specificity of the treatment, quantitative measurements of the m6A levels at GATC of rRNA genes and $Tuba1a$ were assessed by DpnII digestion followed by qPCR[36] (Supplementary Table 1).

DamID-seq was performed using previously described protocols (Vogel et al., 2007). Briefly, 500 ng of genomic DNA was digested for 4 h at 37 °C with DpnI (New England Biolabs) to cut methylated GATC sites. After heat inactivation for 20 min at 80 °C, DamID adaptors (Table S12) were blunt-ended and ligated overnight at 16 °C, followed by heat inactivated at 65 °C for 10 min. In order to cut unmethylated GATC sequences, DNA was digested for 3 h at 37 °C with Dpn II (New England Biolabs) and heat inactivated at 65 °C for 20 min. Adaptor-ligated fragments were amplified using the Advantage® GC 2Polymerase mix (Clontech Takara) (primer described in Table MM, Vogel et al., 2007). DamID libraries were purified using Agencourt AMPure XP beads (Beckam Coulter). Since the size of fragments produced exceeded the one fitting the sequencing machine, the libraries were fragmented using the ds Fragmentase (New England Biolabs) to enrich the concentration of the libraries below 500 bp and, afterwards, the libraries were purified again using the Agencourt AMPure XP beads (Beckam Coulter). The quantity and quality of the isolated DNA were determined with a Qubit® (1.0) Fluorometer (Life Technologies, California, USA). The Nugen Ovation Ultra Low Library Systems (Nugen, Inc, California, USA) was used to prepare the libraries for Illumina sequencing. Briefly, Nucleolar-DamID samples (1 ng) were end-repaired and polyadenylated. Then, Illumina compatible adapters, containing the index for multiplexing, were ligated. The quality and quantity of the enriched libraries were validated using Qubit® (1.0) Fluorometer and the Bioanalyzer 2100 (Agilent, Waldbronn, Germany). The libraries were buffered tin 10 nM in Tris-Cl 10 mM, pH8.5 with 0.1% Tween 20. The TruSeq SR Cluster Kit v4-cBot-HS (Illumina, Inc, California, USA) was used for cluster generation using 8 pM of pooled normalized libraries on the cBOT. Using the TruSeq SBS Kit v4-HS (Illumina, Inc, California, USA) the sequencing was performed as pair-end 150 bp reads using the Illumina NovaSeq 6000.

Sequences are aligned to the mouse reference genome mm10 using Bowtie2 (version 2.3.4.3)[57]. The resulting sam files were converted into bam files, sorted and indexed using samtools (version 1.9)[58]. The bam files were analyzed using the damidseq pipeline script from the Brand group (http://owenjm.github.io/damidseq_pipeline) using a 100kbp resolution[32]. The pipeline bins the mapped reads into GATC-fragments according to GATC-sites indicated by a gff file for the GRCm38 mouse genome (already provided by the authors of the pipeline on the website above) and normalizes reads against the Dam control, in our case the H2B-Dam sample. The pipeline gives a bedgraph file with the log2 ratio of the m6A between H2B-Dam-NoLS and the H2B-Dam only. The bedgraphs were visualized using the tool Integrative Genome Viewer (IGV, version 2.5.2)[59] to extract representative Nucleolar-DamID tracks. The bedgraph files were converted into bigwig files using the bedGraphToBigWig UCSC package (version 4) and the Pearson correlation was assessed using "multiBigwigSummary" and "plotPCA" from deepTools (version 3.2.1)[60]. The bedgraph files were processed with the find_peaks software associated with the pipeline (https://github.com/owenjm/find_peaks) adjusting the values of the FDR and the minimum quantile (FDR < 0.01 and min_quant 0.70). Only the significative peaks common to both replicates were considered as NADs for further analysis.

The identification of NADs overlapping with LADs, genomic contacts with rRNA genes identified by HiC-rDNA, A and B compartment, early/late replicating regions, and ESCsp- and NPCsp-NADs, ESCsp- and NPCsp-LADs were performed using "Intersect intervals" from bedtools (version 2.28.0)[60]. NAD-only and NAD/LAD distribution over the chromosomes, was generated with ChIPseeker[61] in R studio (version 1.0.44). GO term analysis was performed using DAVID 6.8[62].

**ChIPseq**. ChIP analysis was performed as previously described[42]. Briefly, 1% formaldehyde was added to cultured cells to cross-link proteins to DNA. Isolated nuclei were then lysed with lysis buffer (50 mM Tris-HCl, pH 8.1, 10 mM EDTA, pH 8, 1% SDS, 1X protease inhibitor cOmplete EDTA-free cocktail, Roche). Nuclei were sonicated using a Bioruptor ultrasonic cell disruptor (Diagenode) to shear genomic DNA to an average fragment size of 200 bp. 20 µg of chromatin was diluted to a total volume of 500 µl with ChIP buffer (16.7 mM Tris-HCl, pH 8.1, 167 mM NaCl, 1.2 mM EDTA, 0.01% SDS, 1.1% Triton X-100) and incubated overnight with the ChIP-grade antibodies against H3K9me2 and H3K9me3. After washing, bound chromatin was eluted with the elution buffer (1% SDS, 100 mM NaHCO₃). Upon proteinase K digestion (50 °C for 3 h) and reversion of cross-linking (65 °C, overnight), DNA was purified with phenol/chloroform, ethanol precipitated and quantified by qPCR using the primers listed in Supplementary Table 1.

For ChIPseq analyses, the quantity and quality of the isolated DNA were determined with a Qubit® (1.0) Fluorometer (Life Technologies, California, USA) and a Bioanalyzer 2100 (Agilent, Waldbronn, Germany). The Nugen Ovation Ultra Low Library Systems (Nugen, Inc, California, USA) was used in the following steps. Briefly, ChIP samples (1 ng) were end-repaired and polyadenylated before the ligation of Illumina compatible adapters. The adapters contain the index for multiplexing. The quality and quantity of the enriched libraries were validated using Qubit® (1.0) Fluorometer and the Bioanalyzer 2100 (Agilent, Waldbronn, Germany). The libraries were normalized to 10 nM in Tris-Cl 10 mM, pH8.5 with 0.1% Tween 20. The TruSeq SR Cluster Kit v4-cBot-HS (Illumina, Inc, California, USA) was used for cluster generation using 8 pM of pooled normalized libraries on the cBOT. Sequencing was performed on the Illumina HiSeq 2500 single end 126 bp using the TruSeq SBS Kit v4-HS (Illumina, Inc, California, USA).

**ChIPseq data analysis**. Own and published ChIPseq reads were aligned to the mouse mm10 reference genome using Bowtie2 (version 2.3.4.3)[57]. Read counts were computed and normalized using "bamCoverage" from deepTools (version 3.2.1)[63] using a bin size of 50 bp. To calculate read coverage for 20 kb bin region of H3K9me2 and H3K9me3 ChIPseq, "multiBamSummary" from deepTools was used. The border profiles and the read coverage box plots were generated using deepTools. H3K9me2 increase in ESC + pRNA regions distribution over the chromosomes was generated with the ChIPseeker package[61]. Integrative Genome Viewer (IGV, version 2.5.2)[59] was used to visualize and extract representative ChIPseq tracks.

**Identification of rDNA contacts by Hi-C**. The identification of genomic contacts with rRNA genes was performed by recovering reads containing rRNA gene contacts from three published Hi-C data of ESCs and NPCs[35]. The obtained Hi-C data sets have been analyzed with Juicer[64] all in one computational pipeline for generating Hi-C maps from raw fastq data files and command line tools for feature annotation on the Hi-C maps. During Juicer analysis, raw fastq data sets have been aligned to the customized mm10 genome with Burrows-Wheeler Aligner[65] under default parameters. The modified mm10 genome contained one rRNA gene unit attached to the end of chromosome 12. The chromosomal interaction has been extracted from interaction matrices (hic files) with Juicebox tools command *dump* under the following parameters (contacts: observed, normalization applied: Knight-Ruiz matrix balancing[66] under base-pair delimited resolution with bin size 5000). ENCODE Data Analysis Consortium Blacklisted Regions[67] were excluded from the analysis with bedtools[60]. Only Hi-C reads contacting rRNA gene sequences and other genomic sequences have been selected for further analysis through the computation with bedtools pairtoBED function[60] and Python Pandas Library (https://pandas.pydata.org/). Common contacts between the three Hi-C replicates were identified using HiCcompare[68], running the tool under default parameters.

**RNAseq**. Total RNA was purified with TRIzol reagent (Life Technologies). The quality of the isolated RNA was determined with a Qubit® (1.0) Fluorometer (Life Technologies, California, USA) and a Fragment Analyzer (Agilent, Santa Clara, California, USA). Only those samples with a 260 nm/280 nm ratio between 1.8–2.1 and a 28S/18S ratio within 1.5-2 were further processed. The TruSeq Stranded mRNA (Illumina, Inc, California, USA) was used in the succeeding steps. Briefly, total RNA samples (100–1000 ng) were polyA selected and then reverse-transcribed into double-stranded cDNA. The cDNA samples were fragmented, end-repaired and adenylated before ligation of TruSeq adapters containing unique dual indices (UDI) for multiplexing. Fragments containing TruSeq adapters on both ends were selectively enriched with PCR. The quality and quantity of the enriched libraries were validated using Qubit® (1.0). The product is a smear with an average fragment size of approximately 260 bp. Libraries were normalized to 10 nM in Tris-Cl 10 mM, pH8.5 with 0.1% Tween 20. The HiSeq 4000 (Illumina, Inc, California, USA) was used for cluster generation and sequencing according to standard protocol. Sequencing was paired end at 2 × 150 bp or single end 100 bp. The quality of the 120 bp single-end reads generated by the machine was checked by FastQC, a quality control tool for high throughput sequence data[69]. The quality of the reads was increased by applying: a) SortMeRNA[70] (version 2.1) tool to filter

ribosomal RNA; b) Trimmomatic[71] (version 0.36) software package to trim the sorted (a) reads. The sorted (a), trimmed (b) reads were mapped against the mouse genome (mm10) using the default parameters of the STAR (Spliced Transcripts Alignment to a Reference, version 2.4.0.1)[72]. For each gene, exon coverage was calculated using a custom pipeline and then normalized in reads per kilobase per million (RPKM)[73], the method of quantifying gene expression from RNA sequencing data by normalizing for total read length and the number of sequencing reads.

**Reporting summary**. Further information on research design is available in the Nature Research Reporting Summary linked to this article.

## Data availability

The data that support this study are available from the corresponding author upon reasonable request. The Nucleolar-DamID, RNAseq and ChIPseq data generated in this study have been deposited in the NCBI's GEO database under accession code "GSE150822". HiC data in ESCs and NPCs were obtained from GEO "GSE96107"[35] and "GSE112222"[41]. NADs from ESCs were obtained from GEO "GSE103610"[23] and from the 4D Nucleome data portal "4DNESXE9K9DB"[24]. NADs from MEFs were obtained from the 4D Nucleome data portal "4DNES15QV1OO"[48]. LADs from ESCs and NPCs were obtained from GEO "GSE17051"[9]. Replication timing of ESCs were obtained from GEO "GSE95091"[74]. From GEO we obtained the following ChIPseq data: H3K4me3, H3K27me3, H3K9me3, and EZH2 in ESCs "GSE23943"[75], H3K4me1, H3K27ac, and Ring1b in ESCs "GSE72164"[76], H3K9me2 in ESCs "GSE77420"[77], H3K4me3, H3K27ac, and H3K27me3 in NPCs "GSE96107"[35], H3K9me2 in NPCs "GSE122263"[78]. Source data are provided with this paper.

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

## Acknowledgements

This work was supported by the Swiss National Science Foundation (31003A_173056 and 201268 to RS), and ERC grant (ERC-AdG-787074-NucleolusChromatin to R.S.). We thank Peter Meister for the assistance in DamID analysis, Catherine Aquino and the Functional Genomic Center Zurich for the assistance in sequencing, and the Center for Microscopy and Image Analysis of the University of Zurich. We thank Q. Szabo for advices in the generation of oligopaint DNA probes.

## Author contributions

C.B. and D.B. cloned the constructs for the nucleolar DamID. C.B. established ESC lines, set the conditions for the Nucleolar-DamID, performed experiments and data analysis of NADs. J.J.-R. performed and analyzed H3K9me2 ChIPseq experiments. C.B., M.P., and S.G. performed and analyzed DNA-FISH experiments. R.K. analyzed Hi-C data. All authors contributed to experimental design and data interpretation. R.S conceived and supervised the project.

## Competing interests

The authors declare no competing interests.
