## [Peer review file · Nature Communications]

REVIEWER COMMENTS

Reviewer #1 (Remarks to the Author):

The manuscript entitled "Genome-wide maps of nucleolus interactions reveal distinct layers of repressive chromatin domains" by Bersaglieri et al. describes the study of nucleolar-associated domains (NADs) in mouse embryonic stem cells (ESCs) and neural progenitor cells (NPCs) derived from ESCs by retinoic acid-induced differentiation. The basis for most of the results presented is the novel nucleolar DamID assay (Figures 1/S1). Genome-wide NAD/iNAD mapping (iNADs are defined as domains that are not or less frequently associated with nucleoli) of ESCs by nucleolar-DamID and their characterisation by integrated analyses and visualisation of selected chromatin and genome features are shown. This includes mapping long-range contacts of multicopy ribosomal DNA (rDNA) genes with other genomic regions by analysing publicly available HiC data (Figures 2/S2-3). A detailed comparison of NADs with previously published LADs (nuclear lamina associated domains) mapped by LaminB1-DamID is presented next (Figure 3). Further experiments address the effect of induced ribosomal DNA heterochromatin formation (by transfection of 'pRNA') on histone H3K9me2 and H3K9me3 patterns on NADs in ESCs (Figures 4/S4). Finally, the authors map and analyse NADs in NPCs as in ESCs, comparing the differentiated and undifferentiated state (Figures 5-6/S5-6), and focus on changes in chromatin properties, nucleolus/lamina association and, to some extent, expression of gene sets associated with neuron development and differentiation (Figures 7/S7).

The differentiation-dependent dynamics of genome organisation around nucleoli is a relevant and interesting question in 3D genome biology. This question has previously been addressed indirectly in mouse cells by comparing genome-wide maps of NADs from ESCs and mouse embryonic fibroblasts (MEFs). The compared maps were generated by biochemical purification of nucleoli in two independent studies (refs 23, 45). In addition, rDNA-HiC maps of human ESCs and differentiated ESCs are reported (Yu and Lemos, 2018, PLoS Genet). The mouse ESC/NPC in vitro experimental system is well established and functional genomics data such as LAD, HiC, histone modification maps and transcriptome data are available. Therefore, it is a good choice for analysing the dynamics of genome organisation at the nucleolus. This could add an additional layer for integrative analyses and support the more complete understanding of genome organisation and function in this model system. However, the choice of the Nucleolar-DamID assay and the lack of appropriate FISH validation raise major concerns. These and other comments and suggestions are discussed in more detail below.

1. Nucleolar-DamID and proposed controls.

Successful NAD determination depends on the selective and sensitive enrichment of genomic sequences associated with the nucleolus. In the Nucleolar-DamID, differential 6mA modification of GATC sites of nucleolar-associated genomic regions by H2B-Dam-NoLS-DD vs H2B-Dam-DD should serve this purpose. Several elements of this novel assay are critical for success, but they are not systematically analysed or insufficiently presented in the manuscript. In the H2B-Dam-NoLS-DD and H2B-Dam-DD polypeptides, histone H2B, Dam and the NoLS have a differential impact on nuclear and genomic localisation, which will ultimately determine the results of Nucleolar-DamID.

H2B

H2B-Dam is expressed in cells that also express endogenous H2B from multiple gene copies. It is unclear whether the large Dam-NoLS-DD/Dam-DD tags affect nucleosome deposition, stability or remodelling (e.g. by altering accessibility to chromatin-modifying

and remodelling enzymes)? To answer this question, H2B vs. Dam ChIP-seq should be performed in H2B-Dam-DD expressing cells and it should be shown that the tag does not affect genomic distribution.

The deposition of tagged H2B could possibly also lead to altered cell viability if the tag interferes with normal genome function at the site of deposition. It is also unclear whether nucleolar DamID preferentially captures S-phase NADs. It should be demonstrated (e.g. by using cell cycle markers and immunofluorescence (IF) detection of Dam) that H2B-Dam-NoLS-DD is present in the cell and is assembled into chromatin in comparable amounts in the different cell cycle phases G1, S and G2 (M is not relevant for Nucleolar-DamID as nucleoli are disassembled during mitosis).

Dam

Dam alone has a preference to localise to nucleoli, as shown in Supplementary Fig. 2e in Ref 10. This property increases the sensitivity of specific LAD detection (it greatly reduces the NAD background in LaminB1-Dam vs Dam-based LAD mapping), but unfortunately reduces the detectability of NADs.

Compared to other sequencing-based analyses (NGS of isolated nucleoli, rDNA-HiC or -4C, SPRITE), the accessible sequence space in DamID is limited and biased because Dam requires GATC sites. A correlation analysis of GATC density and NAD score should be shown to evaluate potential detectability vs. actual detection of chromosomal regions. This would be particularly important due to the low S/N ratio of Nucleolar-DamID compared to LaminB1-DamID. Another disadvantage of the DamID technique is that it cannot distinguish between stable nucleolar association and short-lived transient contacts, since 6mA is stable and does not disappear from regions transiently associated with the nucleolus. To assess the impact of transient contacts on the nucleolar DamID maps, the H2B-Dam ChIP-seq control mentioned above should be extended with H2B-Dam-NoLS ChIP-seq to obtain a snapshot of the H2B-Dam and H2B-Dam-NoLS genomic profiles, which should then be compared with the 6mA profiles.

In case no good Dam antibody is available, a small peptide tag, e.g. V5, could be added as antigen for ChIP and IF. For example, V5 was also used for IF in Ref 10.

NoLS

NoLS is the key to determining nucleolar association, but it does not guarantee exclusive nucleolar localisation. This is also evident from the H2B-GFP NoLS signals in Figure 1b, where a large proportion of the signal is nuclear. It is noteworthy that the H2B-GFP experiments cannot be considered as suitable controls for the Nucleolar-DamID. GFP and Dam have completely different properties and the H2B-GFP(-NoLS) distribution does not provide information about the H2B-Dam(-NoLS) distribution in the nucleus. The relaxed NoLS-based targeting of H2B-Dam-NoLS-DD to the nucleolus in combination with the partial nucleolar targeting of H2B-Dam-DD by Dam could eventually lead to high background noise in NAD determination by nucleolar DamID.

These theoretical considerations are indeed in agreement with the authors' observations: Log₂ ratios of m6A values shown in Figure 2a (for NADs also in Figure S2b) indicate that DamID is >10-fold more sensitive for LAD detection than for NAD detection:

-2.24/+2.24, i.e. >20-fold maximum difference in signal intensities between LADs and iLADs.

-0.21/+0.21, i.e. <1.5-fold maximum difference in signal intensities between NADs and iNADs.

On the scale of the LaminB1 DamID map, all nucleolar DamID signals would appear as noise.

The high S/N ratio and clear LAD/NAD contrast in LaminB1-DamID is likely due in part to

the preferential nucleolar localisation of Dam alone. The low S/N ratio in nucleolar DamID may be partly due to the same reason: Dam may cause preferential nucleolar localization of H2B-Dam. The results shown in Figure 2g (in the figure, "G" should read "g") also indicate that the method is quite noisy: the H2B-Dam-NoLS-DD signal (DpnII resistance) is only 2-fold higher for rDNA, the core sequence of the nucleolar genome, than for the Tuba1a gene. Overall, DamID is well suited for LAD analyses, but seems to be less suitable for NAD analyses in its current form.

2. Benchmarking

In addition to the essential controls suggested above, benchmarking is required to assess the quality of the Nucleolar-DamID. This is a standard procedure for any novel assay that aims to provide more accurate answers to questions previously addressed by other methods. As is also evident from the literature cited, nucleolus isolation has become the standard in the field, combined with FISH for independent validation of results. The question is how the new method compares to these standards. To this end, at least Nucleolar-DamID should be performed in parallel with nucleolus isolation-based NAD mapping in at least one cell type for which NAD maps are reported, and the results compared side by side. Moreover, an integrated visualisation and correlation analysis of the maps reported in references 22 and 23 and in this study should be presented in the Supplement to provide a first impression of the similarities and differences between NAD detections using different methods in different laboratories in different mESCs. Similarly, rDNA-HiC comparative analyses would also be required (see below under point 4). Such benchmarking could make the study a more valuable extension of the NAD analyses reported in the last decade.

3. FISH

Undoubtedly, FISH is far below the standards in the field. It is insufficient, especially for independent validation of a new methodology. Far more (e.g. 5-10 onlyNADs/NAD-LADs/onlyLADs/iNADs each for ESCs and NPCs - correlated with Dam signal intensities - would be required to get a clearer picture of Dam signal vs. FISH-based nucleolus association frequencies and of the accuracy of onlyNAD vs. onlyLAD specification by Nucleolar-DamID.

Important technical points are that dehydration with graded ethanol concentrations up to 100% ethanol and air drying, further denaturation for 8 min at 83 °C were used in the FISH protocol. It must be shown that the structure of the nucleus is not damaged under these harsh conditions. Perhaps milder conditions should be chosen under which the 3D structures are better preserved.

In Figure S2d "The signal of the LAD probe was located close to NL in the majority of analyzed ESCs (27/33)". How is "close" defined? For LAD FISH quality control, IF staining of nuclear lamina would be required and either distance measurements or counting of overlapping FISH/IF signals should be presented.

4. rDNA-HiC

Discussion of the initial rDNA-HiC datasets from two human cell lines (erythroleukemia K562 and lymphoblastoid cells, Yu and Lemos, 2016, Genome Biol Evol) and genome-wide 4C maps of mouse rDNA-genome interactions in undifferentiated and differentiated

lymphoma cells (Diesch et al, 2019, Comms Bio) is missing from the manuscript. More importantly, an extended study (Yu and Lemos, 2018, PLoS Genet) describing comprehensive rDNA-HiC analyses including long-range rDNA interaction maps of H1 embryonic stem (ES) cells and four differentiated cell types derived from H1: Mesendoderm (ME) cells, mesenchymal stem (MS) cells, neural progenitor (NP) cells and trophoblast-like (TB) cells, also remains unaddressed. Given that Diesch et al, 2019, Comms Bio and Yu and Lemos, 2018, PLoS Genet are cited in the authors' recent review paper (in manuscript ref 11), it is not understandable that these highly relevant studies are not mentioned at all in the manuscript.

Comparative analyses of mouse rDNA-4C (Diesch et al) and rDNA-HiC (this manuscript), as well as human (Yu and Lemos 2018) vs. mouse (this manuscript) ESC/NPC rDNA-HiC maps would further improve the study, but the cited studies and their main findings should at least be discussed.

5. H3K9me2 and H3K9me3 mapping in ESCs with induced rDNA heterochromatin formation.

The experiments shown in Fig. 4 and S4 (ESC pRNA transfection) interrupt the flow of the manuscript and are only weakly connected to the other experiments in the manuscript. However, they are highly interesting and open up a new direction that could be elaborated in an independent study. Fascinating questions that arise here: What factors mediate the heterochromatin formation of NADs, which is first induced by pRNA on rDNA? Is physical proximity, direct contact necessary for spreading? What determines that H3K9me2 and H3K9me3 occupancies change at rDNA and satellite repeats, while only H3K9me2 changes in NADs? The possible role of the recently published MiCEE complex (Singh et al, 2018, Nat Genet) could also be assessed in the extended study. Addressing these questions could be an interesting stand-alone study, building on the experiments shown in the manuscript. They could be then removed from here.

6. NAD coverage

Nucleolar-DamID identifies >70-80% of some chromosomes as NADs according to the colour code in Figure S2 (e.g. Chr. 1, Chr. 3, Chr. 7). These chromosomes do not have nucleolar organiser regions containing ribosomal DNA. Are they nevertheless almost completely associated with nucleoli? It is not clear how the much lower NAD coverage values (in Figure S5) are calculated and how they relate to the colour codes shown on the maps in Figure S2? How should these Nucleolar-DamID maps be interpreted?

7. Comments on the discussion

The discussion would need to be thoroughly refined. It provides a biased view of published studies in several places and it also proposes some purely speculative scenarios that are not or only marginally supported by the data presented.

Page 15-16 lines 18-28, 1-4

“Previous methods for NAD identification used sonication-based biochemical purification of nucleoli, a methodology that is subjected to a certain variation in nucleoli preparation between different cell types and relatively biased toward sonication-resistant heterochromatin 18-20,22,23,45. Relative to this methodology, the application of Nucleolar-DamID and HiC-rDNA has no bias for any kind of chromatin state and does not vary between cell types, thereby allowing direct comparisons of chromosome architecture around the nucleolus between distinct cell states.”

Nucleolar-DamID has not yet been shown to be free of bias and not to vary between (which?) cell types and cell states.

In addition to the critical aspects of Nucleolar-DamID described in point 1, this assay has the other disadvantages:

- i. Knock-in cells and cell clones must be generated.**
 - ii. H2B-Dam-DD cannot be used simultaneously with H2B-Dam-NoLS-DD in the same cells as an internal control.**
 - iii. It must be ensured that the concentrations of H2B-Dam-NoLS-DD are comparable in each cell of the clonal population. Despite integration of coding sequences at the same locus and clonal selection, this is not trivial and the level of variation could also be different in different cell lines.**
 - iv. As in iii. for H2B-Dam-DD expressing cells.**
 - v. If application in different cell types and cell states is desired, it must be ensured that the steady-state levels of H2B-Dam-NoLS-DD and H2B-Dam-DD are comparable in each setting (and as mentioned above, also homogeneous in the cell population).**
- To overcome these drawbacks, a set of quality controls needs to be established. At this premature stage, before the controls are presented, the validity of the assay is greatly overestimated.**

The disadvantages of nucleolus isolation:

- i. The protocol needs to be adapted for different cell types.**
 - ii. Co-sedimentation of chromatin can lead to non-specific background.**
- To overcome these disadvantages, the purity of the nucleolus fraction must be assessed microscopically and nucleolar protein and rDNA enrichment analyses must be performed by western blot and qPCR, respectively. They allow quantitative comparative analyses between different preparations. Co-sedimented chromatin can be separated by FACS sorting, which has been successfully used for nucleolus isolation in Arabidopsis (Pontvianne et al, 2016, Cell Rep; Pontvianne et al, 2016; Carpentier et al, 2018, Methods Mol Biol).**

At the current stage of method development, the quality controls of nucleolus isolation techniques are better established and the assay appears to be less bias and error prone than Nucleolar-DamID.

rDNA-HiC, a method introduced by the Lemos lab, has the main disadvantage that it cannot provide a complete picture of NADs. Long-range interactions of chromosomal regions other than rDNA, such as NOR-adjacent sequences and centromeres, may also be involved in nucleolar-associated genome organisation and are not detected by rDNA-HiC. This disadvantage is an inherent property of the method and cannot be overcome.

“We found some similarities between NAD maps obtained with previous nucleoli purifications and our methodologies, such as repressive chromatin features and enrichment in olfactory receptor and zinc finger genes 18,19. It is important here to note that major and minor satellites, which composed centric and pericentric domains and known to be in contact with the nucleolus, could not be identified by Nucleolar-DamID due to the lack of GATC motif. However, Nucleolar-DamID and HiC-rDNA revealed further and novel details in chromosome architecture around the nucleolus that did not emerge from previous studies.”

It cannot yet be ruled out that some of these novel details are due to DamID capturing of chromosomal regions that have a low frequency of contact with the nucleolus. As suggested above, extensive FISH validation would be required, which should then include analysis of multiple NADs identified only with Nucleolar-DamID.

“Further, NAD maps provided in this work can be integrated in future studies of 3D genome organization that have so far did not take into account the nucleolus because of the lack of genome-wide NAD profiles”

???

There is no lack of genome-wide NAD profiles. They are described in the cited literature. “... future studies of 3D genome organization that so far did not take into account the nucleolus...” - this should be rephrased.

Page 16 lines 17-19

“Thus, contacts with the nucleolus, even for sequences able to interact with the NL, shape a distinct repressive chromatin state.”

The specific features of this shaped distinct chromatin state should then highlight the nucleoli and nuclear lamina. This could be demonstrated by using multiplex IF, where the main determinants of the specific chromatin state should be visualised simultaneously.

Page 17 lines 9-14

“Further, we identified cell type specific NADs, unique to ESC or NPCs, and showed that the position of a NAD or a LAD relative to the nucleolus or NL in ESCs correlates with their cellular location upon dissociation from the nucleolus or NL during differentiation. In other words, a NAD that interacts with the nucleolus but not with the NL in ESCs, after losing contacts with the nucleolus upon differentiation, still maintains features that makes it refractory to associate with the NL.”

Positions and cellular location were only shown for one NAD and one iNAD region by FISH. DamID does not provide information on position. The association frequencies of several selected example regions should be measured by FISH, as mentioned above.

If a NAD loses contact with the nucleolus during differentiation due to the change in its chromatin state, then the lack of NL association could be due to the acquisition of new chromatin properties rather than the retention of some of the old ones. Both speculative scenarios and their combination are conceivable.

Page 18 line 3

“our work established novel methodologies to identify NADs in every cell type”

Without controls and testing, this is an overstatement for Nucleolar-DamID. Importantly, Nucleolar-DamID cannot be adapted for in vivo settings, the ultimate objective for biological studies. What is (are) the other novel methodology (methodologies)?

Given the inherent properties of the methods briefly summarised above, the method of choice should rather be biochemical purification of nucleoli in combination with FACS, the method called FANoS (fluorescence-activated nucleolar sorting) established in Arabidopsis (Pontvianne et al, 2016, Cell Rep; Pontvianne et al, 2016; Carpentier et al, 2018, Methods Mol Biol) and/or highly multiplexed FISH. FaNOS would also work in animal tissue samples.

Page 18 lines 6-8

“We predict that the application of Nucleolar-DamID and HiC-rDNA will be relevant for future works, including the understanding of the determinants for the specific targeting to nucleolus or NL or the application of single cell analyses that were prohibitive with the previous biochemical based-methods.”

Single cell/single nucleolus analyses would be possible with biochemical based FANoS.

In summary, while NAD mapping and rDNA-HiC analyses are not new, the authors present a novel method called Nucleolar-DamID assay. However, this method is not yet well

established, controls and benchmarking have yet to be presented. The topic of differentiation-dependent dynamics of the nucleolar-associated genome is still relevant with many open questions, although it has been partially investigated by rDNA-HiC in human cells and indirectly by NAD analyses in mouse cells. Due to the insufficient technical quality, which is more due to the choice of method than to the quality of the overall performance, I can only recommend the manuscript for publication if the above-mentioned concerns are satisfactorily addressed.

Reviewer #2 (Remarks to the Author):

The genomic loci associated with the nucleolar periphery (known as nucleolar-associated domains, or "NADs" for short) are of high interest to the study of genome organization. Previous techniques for NAD analyses include biochemical isolation of nucleoli, DamID using a nucleolar targeting peptide, or TSA-Seq, an antibody-based proximity labeling method (PMID: 30154186, 33446254). Here, the authors have devised a novel DamID-based method for genome-scale detection of NADs, fusing the bacterial Dam adenine methyltransferase to histone H2B plus a nucleolar targeting sequence. DamID is an appealing idea, since it can in principle be applied to all cell types, can be performed on fewer cells than required for biochemical purification (and possibly be extended to single cell approaches as has Lamin-DamID), and is independent of antibodies that can vary in quality. Additionally, the experimental manipulations via differentiation and pRNA treatment make this a valuable study. However, there are features of this paper that need more consideration, as discussed below.

1. As an alternative method for NAD mapping, the authors mine Hi-C datasets for rDNA-containing junction fragments. Since the field would benefit from comparison of different methods, it is a missed opportunity that there is no side-by-side comparison of NADs mapped by the DamID method and the results obtained via the Hi-C analyses. Furthermore, nothing is done here to test whether these methods are providing better maps of NAD associations than achieved in other studies. For example, how do the DamID and Hi-C-based peaks obtained in this study compare to those in previous biochemical analyses of mouse ES cell NADs (PMID: 32160538, 32219510)

2. Those DamID/Hi-C comparisons could be useful in dealing with this question: There are clear increases in the number of Hi-C-rDNA contacts and H2B-NoLS-DamID signals in the differentiated NPC cells compared to the undifferentiated ES cells (Fig. 5c-d). This is the expected result since it is well established that differentiation leads to increased levels of heterochromatin, and previous analyses showed many fewer rDNA junction fragments in human ESCs than in human somatic cells (PMID: 29570716). Therefore, it is counter-intuitive that genomic NAD coverage would be higher in the ES cells than in NPCs (Fig. S5b). Additionally, previous comparisons of mouse ES and somatic fibroblast NADs detected via biochemical isolation indeed detected a greater genomic NAD coverage in the somatic cells (PMID: 32219510). Does this mean that NPCs have fewer NADs than do fibroblasts? An alternative possibility would be that NAD peaks are too liberally assigned from the DamID data in the current study. That is, in ES cells with a more mobile genome (e.g. PMID: 16399082, 24831699), broadly distributed but transient interactions (e.g. Fig. 5e) could be recorded as contacts by the Dam methyltransferase. Therefore, independent validation of NAD peaks in the mES cells would be valuable. This would be best tested via DNA-FISH studies as an orthogonal method. I note that only a single NAD locus is tested here by this method, and comes from the highly nucleolar-associated, rDNA-containing chr19 (Fig. 2i), and therefore analysis of a more diverse set of loci is warranted. Alternatively, would the Hi-C analyses provide an alternative metric by which to validate the DamID data?

3. What are the spots of H2B-NoLS-GFP in Fig. 1d that do not appear to localize with the main nucleolar mass? There needs to be co-staining with a fiduciary nucleolar marker protein. If these spots are not nucleolar, what are they? The concern is that these could influence the designation of NAD regions in unforeseen ways.

4. The analysis of histone modifications at NAD boundaries is interesting. The changes in the H3K9me2 levels flanking NADs is clearly elevated in the ES cells treated with pRNA. What happens to H3K27me3 at the boundaries under the same conditions? And what happens to both these marks at boundaries upon differentiation into NPC cells?

5. The ability to view and analyze the genomic data at GEO was not sufficient. There are xlsx files of called peaks, but no availability was made of the raw DamID replicate datasets. The DamID score files (GSE150822_ESC_NADs_Nucleolar-DamID_score.csv.gz and GSE150822_NPC_NADs_Nucleolar-DamID_score.csv.gz) don't have any inserted files when opened.

6. Finally, more needs to be done to acknowledge previous contributions in this area. For example, use of rDNA crosslinks from Hi-C studies was previously pioneered by Bernardo Lemos (PMID: 27797956, 29570716), and a different nucleolar-targeted DamID was reported by the van Steensel laboratory (PMID: 33446254).

Reviewer #3 (Remarks to the Author):

To the authors:

Remarks to the authors:

Bersaglieri et al. in their manuscript "Genome-wide maps of nucleolus interaction reveal distinct layers of repressive chromatin domains" adapted the previously described DamID method to be able to map for the first time the chromatin domains (NADs) which interact with the nucleolus in ESC and after their differentiation into NPC. In addition, they provide new HiC-rDNA data about the frequency of chromatin interactions with the nucleolus. These findings are very interesting and address an important missing aspect of epigenetic regulation: the contribution of the nucleolus to the 3D organization of the chromatin in the eukaryotic nucleus.

The data presented are of high quality and the paper is well-written.

Therefore, I recommend the paper for publication in Nature Communications after addressing my minor comments

General Comments:

The authors have chosen to normalize the H2B-Dam_NoLS to H2B-Dam to compensate for an eventual incorporation of the tagged H2B into chromatin regions outside of the nucleolus. They have shown many convincing controls to exclude this, however they did not consider the possibility that incorporation of H2B-Dam-NoLS due to its rather big size could interfere with rRNA expression and integrity of the nucleolar chromatin. In particular

1. in Fig. 1D they could show that localization of TTF1 in the presence of H2B-Dam_NoLS without ActD is not affected

2. It would be important to show that H2B-Dam_NoLS does not affect rRNA expression by RT-PCR

Minor Points:

1. I did not find anywhere (neither in the text or in the figure legend) explained what compartment A and B refer to. I suggest to shortly define them for the non `chromatin organization` experts

2. In Fig S2A, panel b and c have been swapped and do not correspond to the description in

the text

3. In Fig S4 the correlation coefficient like in Fig.4c is missing

Genome-wide maps of nucleolus interactions reveal distinct layers of repressive chromatin domains.

Bersaglieri *et al.*,

Response to the reviewers

We thank the reviewers for the positive comments. In this revised version, we modified the text to clarify points that were not sufficiently clear and included new data according to the reviewers' suggestions. All the new data support the conclusions of our work.

List of changes

Modifications in the text were highlighted in red.

The new data are listed here below.

Description of new data	Reviewer	Figure Number in the revised manuscript
IF images with nucleolar markers	Rev. 2	Fig. 1c
IF images of cells treated with ActD with nucleolar markers	Rev. 2&3	Fig. 1e
DNA-FISH images and corresponding quantifications in ESCs	Rev1&2	Fig. 2i-l; Fig. S4a-e
IGV track comparing NAD profiles in ESCs from different publication	Rev. 1	Fig. S1
Representative bright field images of ESC lines expressing H2B-Dam-DD and H2B-Dam-NoLS-DD transgenes and parental ESCs (WT)	Rev. 1	Fig. S2c
Measurement of 45S pre-rRNA and pluripotency factors in the H2B-Dam-DD and H2B-Dam-NoLS-DD ESC lines.	Rev. 1&3	Fig. S2d
Inclusion of correlation coefficient	Rev. 3	Fig. S6
DNA-FISH images and corresponding quantifications in NPCs	Rev. 1&2	Fig. S7a-g
Quantification of NADs common to ESCs and NPCs	Rev. 1	Fig. S8a
NAD and LAD border analysis in NPCs	Rev. 2	Fig. S8d

REVIEWER COMMENTS

Reviewer #1 (Remarks to the Author):

The manuscript entitled "Genome-wide maps of nucleolus interactions reveal distinct layers of repressive chromatin domains" by Bersaglieri et al. describes the study of nucleolar-associated domains (NADs) in mouse embryonic stem cells (ESCs) and neural progenitor cells (NPCs) derived from ESCs by retinoic acid-induced differentiation. The basis for most of the results presented is the novel nucleolar DamID assay (Figures 1/S1). Genome-wide NAD/iNAD mapping (iNADs are defined as domains that are not or less frequently associated with nucleoli) of ESCs by nucleolar-DamID and their characterisation by integrated analyses and visualisation of selected chromatin and genome features are shown. This includes mapping long-range contacts of multicopy ribosomal DNA (rDNA) genes with other genomic regions by analysing publicly available HiC data (Figures 2/S2-3). A detailed comparison of NADs with previously published LADs (nuclear lamina associated domains) mapped by LaminB1-DamID is presented next (Figure 3). Further experiments address the effect of induced ribosomal DNA heterochromatin formation (by transfection of 'pRNA') on histone H3K9me2 and H3K9me3 patterns on NADs in ESCs (Figures 4/S4). Finally, the authors map and analyse NADs in NPCs as in ESCs, comparing the differentiated and undifferentiated state (Figures 5-6/S5-6), and focus on changes in chromatin properties, nucleolus/lamina association and, to some extent, expression of gene sets associated with neuron development and differentiation (Figures 7/S7).

The differentiation-dependent dynamics of genome organisation around nucleoli is a relevant and interesting question in 3D genome biology. This question has previously been addressed indirectly in mouse cells by comparing genome-wide maps of NADs from ESCs and mouse embryonic fibroblasts (MEFs). The compared maps were generated by biochemical purification of nucleoli in two independent studies (refs 23 (Bizhanova et al), 45 Vertii et al). In addition, rDNA-HiC maps of human ESCs and differentiated ESCs are reported (Yu and Lemos, 2018, PLoS Genet).

Author: We would like to clarify that the works referred by this Reviewer are very different from our study that established a novel method to measure NADs in mouse ESCs and derived NPCs. The reviewer refers to published NAD studies using nucleolar purification from mouse ESCs and MEFs (two independent studies from the same lab, Bizhanova et al, Vertii et al) and HiC data from human cells (Yu and Lemos). We would like to clarify that, nowadays, comparisons between ESCs and MEFs are not anymore considered the proper system to study "the differentiation-dependent dynamics of genome organisation" since these cell types are too distant from each other. In our study, we analysed ESCs and the derived NPCs, a system that is well accepted to determine how the genome undergoes reorganization during the early steps of development. Further, we would like to start to clarify that the "rDNA-HiC maps of human ESCs and differentiated ESCs reported by Yu and Lemos" could not be further analysed, as stated by the authors, due to the low sequencing depth "...greater coverage for these cells is necessary to draw sufficiently dense contact maps for a more fine-grained and meaningful biological contrast." Further, Yu and Lemos did not measure or discuss the differences between ESC and differentiated cells, but only general features of rDNA-contacts measured with the human LCL and K562 cell lines such as the enrichment in the repressive B compartment. Further details of the comparison of our and these studies can be found below.

The mouse ESC/NPC in vitro experimental system is well established and functional genomics data such as LAD, HiC, histone modification maps and transcriptome data are available. Therefore, it is a good choice for analysing the dynamics of genome organisation at the nucleolus. This could add an additional layer for integrative analyses and support the more complete understanding of genome organisation and function in this model system. However, the choice of the Nucleolar-DamID assay and the lack of appropriate FISH validation raise major concerns. These and other comments and suggestions are discussed in more detail below.

Author: Here below is our detailed response to the concerns raised by the Reviewer. This revised manuscript included additional DNA-FISH analyses that together with the previous HiC-rDNA results proved the validity of the Nucleolar-DamID to accurately identify NADs.

1. Nucleolar-DamID and proposed controls.

Successful NAD determination depends on the selective and sensitive enrichment of genomic sequences associated with the nucleolus. In the Nucleolar-DamID, differential 6mA modification of GATC sites of nucleolar-associated genomic regions by H2B-Dam-NoLS-DD vs H2B-Dam-DD should serve this purpose. Several elements of this novel assay are critical for success, but they are not systematically analysed or insufficiently presented in the manuscript. In the H2B-Dam-NoLS-DD and H2B-Dam-DD polypeptides, histone H2B, Dam and the NoLS have a differential impact on nuclear and genomic localisation, which will ultimately determine the results of Nucleolar-DamID.

H2B

H2B-Dam is expressed in cells that also express endogenous H2B from multiple gene copies. It is unclear whether the large Dam-NoLS-DD/Dam-DD tags affect nucleosome deposition, stability or remodelling (e.g. by altering accessibility to chromatin-modifying and remodelling enzymes)? To answer this question, H2B vs. Dam ChIP-seq should be performed in H2B-Dam-DD expressing cells and it should be shown that the tag does not affect genomic distribution.

Author. It is not clear to us what impact has in our work the fact that endogenous H2B is expressed by multiple gene copies.

We would like to clarify that the inducible H2B-Dam-DD ESC lines were only used to mark genomic contacts with the nucleolus. All the other analyses such as transcription, gene expression, ChIPseq, and HiC were performed with wild type ESCs using our own and published dataset, as clearly indicated. Therefore, any hypothetical perturbation on nucleosome deposition, stability, or remodelling cannot affect the readout of the methodology here used since the Dam-ID has been only applied to identify genomic contacts with the nucleolus. Further, the new data from both transfected NIH3T3 cells (high transgene expression) and ESC lines (very low transgene expression) showed the absence of alterations in nucleolus structure, 45S pre-rRNA levels, expression of pluripotency genes, and ESC morphology (**Figs. 1c,d, Figs. S2c,d**). Thus, the method we used does not alter nucleolus morphology and function and pluripotency state.

We would like to clarify that the expression of the transgenes in ESCs has been induced for only 15 hours and been kept very low, using a minimal promoter as done in previous Lamin B1-DamID analyses. As stated in the manuscript, the expression is undetectable at protein levels with antibodies for IF and western blot analyses. This is a requirement for DamID analyses in order to not saturate the methylation of GATC sequences. Therefore, the suggested ChIPseq of the DamID constructs cannot be performed since their expression is extremely low and cannot be detected with antibodies. For the reasons described above, we think that it is not necessary to validate the Nucleolar-DamID with Dam-ChIPseq experiments.

The deposition of tagged H2B could possibly also lead to altered cell viability if the tag interferes with normal genome function at the site of deposition.

Author. If there had been defects in cell viability, we would not have continued with this strategy. The expression of the DamID constructs did not affect the ESC lines, which display compact and bright colonies

similarly to parental ESCs (new data in **Fig. S2c**). Further, we measured the expression of rRNA genes and pluripotency markers and found no alterations (**Fig. S2d**). Thus, we can exclude with high confidence that “the deposition of tagged H2B could possibly lead to altered cell viability”. This is also an expected result considering the relatively short time (15 h) of extremely low levels of transgene expression. We would also like to clarify that in the past many cell lines expressing H2B tagged with a large tag were published and no defects in cell viability were reported.

It is also unclear whether nucleolar DamID preferentially captures S-phase NADs. It should be demonstrated (e.g. by using cell cycle markers and immunofluorescence (IF) detection of Dam) that H2B-Dam-NoLS-DD is present in the cell and is assembled into chromatin in comparable amounts in the different cell cycle phases G1, S and G2 (M is not relevant for Nucleolar-DamID as nucleoli are disassembled during mitosis).

Author. The investigation of when the Dam-fused histones are assembled into DNA is out of the scope of this work. To our knowledge, it has never been reported a cell cycle specific LADs. By analogy, we think that the existence of “S-phase NADs” is very unlikely. For the reasons clarified above we cannot detect with antibodies the Dam-fused histones in ESCs. Further, since the transgenes are expressed under the control of a TET-ON inducible system and Shield, it is expected that upon induction all cells express the transgenes. Finally, we do not think that IF experiments using cell cycle markers can detect when the Dam-fused histones are incorporated into chromatin.

Dam

Dam alone has a preference to localise to nucleoli, as shown in Supplementary Fig. 2e in Ref 10 (Guelen *et al*). This property increases the sensitivity of specific LAD detection (it greatly reduces the NAD background in LaminB1-Dam vs Dam-based LAD mapping), but unfortunately reduces the detectability of NADs.

Author. All DamID experiments published so far have never used as control Dam alone but always as fusion with another protein, usually GFP. This is done to decrease the elevated Dam expression/activity levels that otherwise will unavoidably saturate the GATC sites of the genome. Also in our case, we used as control a Dam-fused construct (i.e. H2B-Dam).

The criticism of the reviewer is based on the Supplementary Fig. S1 of Guelen *et al* (2008) that reported the nucleolar localization of the unfused Dam in “one cell”. We have deeply searched in the literature and the nucleolar localization of “Dam alone” has never been reported elsewhere. Further, to our knowledge, no other papers using DamID for the identification of LADs or TFs reported a bias for the nucleolus.

As shown in the image below (live-cell image of transfected HEK293T), Dam fused to GFP displays a homogeneous nuclear localization with no preferential nucleolar localization. This information was also present in **Figure S2** showing GFP-Dam inducible expression in HEK293T cells. These results are consistent with the literature. Therefore, the reported nucleolar localization of Dam alone is not relevant to our work since we used Dam-fusion proteins. Finally, the interpretation of this Reviewer concerning the sensitivity of specific LAD detection that “unfortunately reduces the detectability of NADs” is very unclear to us and definitively not relevant for this work.

Compared to other sequencing-based analyses (NGS of isolated nucleoli, rDNA-HiC or -4C, SPRITE), the accessible sequence space in DamID is limited and biased because Dam requires GATC sites. A correlation analysis of GATC density and NAD score should be shown to evaluate potential detectability vs. actual detection of chromosomal regions.

Author. We would like to clarify that HiC and derivative technologies are also based on sequence recognition since they use enzymatic digestion with 4- or 6-cutter enzymes. In the case of Nucleolar DamID, we do not think that the GATC sequence represents a limitation for the genome wide identification of NADs. In the mouse genome the frequency of GATC sequences is ~260 bp on average. Only few portions of the genome are free of GATC such as minor and major satellites (repeat unit length 120bp and 234 bp, respectively) and this point was already clarified in the discussion. Further, minor and major satellite sequences are not present in the reference genome and, consequentially, cannot be simply detected by next generation sequencing even by using nucleolus purification-based methods. Thus, we exclude that the GATC sequence represents a limitation of the Nucleolar-DamID.

The GATC density score between NADs and iNADs is shown here below and, as expected, does not show any enrichment in NADs. We decided to not include this measurement in the manuscript since it does not add any relevant information.

Track

This would be particularly important due to the low S/N ratio of Nucleolar-DamID compared to LaminB1-DamID.

Author. The experimental setups of the Nucleolar-DamID and Lamin B1-DamID are very different. Consequentially, no conclusion can be drawn by comparing the m6A ratio values between H2B-NoLS-Dam vs H2B-Dam and Lamin B1-DamID vs GFP-Dam.

1. In LaminB1-DamID, the control GFP-Dam is completely different from Lamin B1 and does not interact with LADs. In our case, we are comparing two similar proteins (H2B and H2B-NoLS) which only differ in the nuclear localization. Thus, the control H2B-Dam can also interact with NADs since it is a histone. This is why the m6A ratio Lamin B1 vs Dam is higher than H2B-Dam vs H2B-NoLS-Dam.
2. The LAD profile was obtained by microarray and the analysis was performed with a R program developed by van Steensel lab. Our Nucleolar-DamID was performed with NGS and analysed with the established DamIDseq pipeline published by Marshall & Brand.

Another disadvantage of the DamID technique is that it cannot distinguish between stable nucleolar association and short-lived transient contacts, since 6mA is stable and does not disappear from regions transiently associated with the nucleolus. To assess the impact of transient contacts on the nucleolar DamID maps, the H2B-Dam ChIP-seq control mentioned above should be extended with H2B-Dam-NoLS ChIP-seq to obtain a snapshot of the H2B-Dam and H2B-Dam-NoLS genomic profiles, which should then be

compared with the 6mA profiles. In case no good Dam antibody is available, a small peptide tag, e.g. V5, could be added as antigen for ChIP and IF. For example, V5 was also used for IF in Ref 10.

Author. To our knowledge, no genomic methods, including the biochemical purification of nucleoli, can distinguish between stable and transient interactions. What is measured with these methods is the contact frequency and this is what we have done in this study, in contrast to previous works using nucleolar purification that showed only NAD coverage. Further, we would like to clarify that the suggested ChIP analysis will not help since the ChIP itself cannot distinguish between stable and transient interactions. Therefore, we think that the analysis of the nature of the nucleolar interactions (stable or transient) goes beyond the scope of this study and the lack of this information is not a disadvantage compared to other NAD methods that also failed to provide this information. Further, as already stated above, the low transgene expression does not allow the use of methods based on antibodies. Finally, the new DNA-FISH data (see details in “3. FISH” section) validated the NADs identified by the Nucleolar-DamID in both ESCs and NPCs (**Fig. 2** and **Fig. S4** and **S7**).

NoLS

NoLS is the key to determining nucleolar association, but it does not guarantee exclusive nucleolar localisation. This is also evident from the H2B-GFP NoLS signals in Figure 1b, where a large proportion of the signal is nuclear.

Author. We kindly disagree. The images of **Figure 1b** (transient transfection of NIH3T3 cells) showed a clear enrichment of H2B-GFP-NoLS. This can also be observed in the IFs of the old Figure 1d and the new **Figures 1c,e**. Further, we stated in the manuscript that the use of the H2B-Dam as control was also thought to compensate eventual incorporation of the nucleolar-H2B in regions outside the nucleolus. Finally, to detect GFP-signal, the images were taken from transiently transfected cells that express much higher level of GFP-fusion proteins compared to the stable ESC lines where transgene expression is very low and cannot be detected by antibodies. This revised version includes new IF analyses using NPM1 and Fibrillarin as nucleolar markers (**Figs. 1c,e**). The data are consistent with the previous results showing the nucleolar localization of H2B-NoLS.

It is noteworthy that the H2B-GFP experiments cannot be considered as suitable controls for the Nucleolar-DamID. GFP and Dam have completely different properties and the H2B-GFP(-NoLS) distribution does not provide information about the H2B-Dam(-NoLS) distribution in the nucleus. The relaxed NoLS-based targeting of H2B-Dam-NoLS-DD to the nucleolus in combination with the partial nucleolar targeting of H2B-Dam-DD by Dam could eventually lead to high background noise in NAD determination by nucleolar DamID.

Author. The use of GFP to visualize the cellular localization was done since we wanted to obtain life-cell images and avoid fixation procedures that can sometimes alter the signal, in particular nucleolar signal. We think that this control was correctly done and there is no reason to think that H2B tagged with GFP or Dam tag should behave differently. Considering that the data of the Nucleolar Dam-ID were validated with HiC-rDNA, SPRITE, and the new additional DNA-FISH analyses, we do not think that we need further validations to demonstrate the nucleolar localization of the nucleolar H2B.

DamID: microarray, lentivirus, pipeline

These theoretical considerations are indeed in agreement with the authors' observations:

Log₂ ratios of m6A values shown in Figure 2a (for NADs also in Figure S2b) indicate that DamID is >10-fold more sensitive for LAD detection than for NAD detection: $-2.24/+2.24$, i.e. >20-fold maximum difference in signal intensities between LADs and iLADs. $-0.21/+0.21$, i.e. <1.5-fold maximum difference in signal intensities between NADs and iNADs. On the scale of the LaminB1 DamID map, all nucleolar DamID signals would appear as noise.

Author. The theoretical considerations of Reviewer 1 are not in agreement with our data. As explained above, the calculations of the Reviewer 1 are not valid since he is comparing values obtained with different controls, sequencing technologies, and bioinformatic pipelines.

The high S/N ratio and clear LAD/NAD contrast in LaminB1-DamID is likely due in part to the preferential nucleolar localisation of Dam alone. The low S/N ratio in nucleolar DamID may be partly due to the same reason: Dam may cause preferential nucleolar localization of H2B-Dam.

Author. This point was discussed above. DamID experiments, including ours, do not use Dam alone and Dam-GFP does not localize in the nucleolus (see above IF image above). Consequentially, the “one cell” showing nucleolar localization of Dam alone reported in the Supplementary Figure of Guelen et al is not a relevant information for this work.

The results shown in Figure 2g (in the figure, "G" should read "g") also indicate that the method is quite noisy: the H2B-Dam-NoLS-DD signal (DpnII resistance) is only 2-fold higher for rDNA, the core sequence of the nucleolar genome, than for the Tuba1a gene.

Author. The signal is specific and significant, and this is what it counts in this enzymatic assay.

Overall, DamID is well suited for LAD analyses, but seems to be less suitable for NAD analyses in its current form.

Author. We kindly disagree. If only one single technical issue raised by this reviewer was true, we would not obtain NAD profiles that are consistent with the current literature, including the enrichment in LADs. In contrast, we would have obtained a flat profile with no NAD enrichment. Further, the new FISH data (see details in “3. FISH” section) strongly indicated that the Nucleolar-DamID is superior to previous methodologies since it is able to identify NADs that were not detected with the nucleolar purification.

2. Benchmarking

In addition to the essential controls suggested above, benchmarking is required to assess the quality of the Nucleolar-DamID. This is a standard procedure for any novel assay that aims to provide more accurate answers to questions previously addressed by other methods. As is also evident from the literature cited, nucleolus isolation has become the standard in the field, combined with FISH for independent validation of results.

Author. The nucleolus isolation to identify NAD was so far the only possible method. Define it as standard is an overstatement. Further, the Nucleolar-DamID was validated with HiC-rDNA and showed a 100% overlapping with SPRITE data. The additional validation of NADs by FISH (see details in “3. FISH” section) supported the robustness and accuracy of the Nucleolar-DamID.

The question is how the new method compares to these standards. To this end, at least Nucleolar-DamID should be performed in parallel with nucleolus isolation-based NAD mapping in at least one cell type for which NAD maps are reported, and the results compared side by side. Moreover, an integrated visualisation and correlation analysis of the maps reported in references 22 and 23 (Lu et al, Bizhanova et al) and in this study should be presented in the Supplement to provide a first impression of the similarities and differences between NAD detections using different methods in different laboratories in different mESCs. Similarly,

rDNA-HiC comparative analyses would also be required (see below under point 4). Such benchmarking could make the study a more valuable extension of the NAD analyses reported in the last decade.

Author. We think it is not necessary to perform in parallel nucleolus isolation-based NAD and Nucleolar-DamID since the two methodologies are completely different from each other. Further, since it is almost impossible to obtain nucleoli from ESCs unless they are crosslinked, we decided to not perform nucleoli purification. Instead, as indicated below, we used NAD profiles published by others.

As suggested, we included a new **Figure S1** showing IGV tracks of our NAD profiles and NADs profiles published by Lu *et al* and Bizhanova *et al.* in ESCs (see also image below). As stated in the manuscript, the two previously published NAD profiles (Lu *et al*, Bizhanova *et al*) that used biochemical purification of the nucleoli extensively differ from each other (i.e. NAD genome coverage 7.5% in Lu *et al.*, and Bizhanova *et al.* 30%), highlighting the difficulties to standardize nucleoli purification even from the same cell type. The nucleolar-DamID will not encounter this problem as also evident by the similar, as expected, NAD profiles between ESCs and NPCs.

In the previous version of this work, we stated that “*We found some similarities between NAD maps obtained with previous nucleoli purifications and our methodologies, such as repressive chromatin features and enrichment in olfactory receptor and zinc finger genes*”. In the discussion of this revised manuscript, we further clarified this point. The comparison with the NADs detected by Bizhanova *et al* showed that 40% of NAD sequences identified by the Nucleolar-DamID in ESCs are also detected as NADs by Bizhanova *et al.* We think this is a good value considering not only the different methodologies but also the different bioinformatic analyses to call for NADs. However, we think that the NAD profile of Bizhanova *et al* failed to detect all NADs in ESCs. Indeed, the new DNA-FISH data validated the contacts with the nucleolus of sequences identified as NAD by the Nucleolar-DamID and that were not detected as NADs by Bizhanova *et al* (see details in “3. FISH” section). Another substantial difference between the two methodologies is that the NAD profiles of ESCs and MEFs generated by the same group (Bizhanova *et al* and Verti *et al*) have only 40% of common NADs, suggesting that the NAD profiles between these two cell types is remarkably distinct. This is in contrast with the NAD profiles of ESCs and NPCs generated with the Nucleolar-DamID and HiC-rDNA, showing that NAD composition between ESCs and NPCs is very similar (84% of NAD sequences are in common between ESCs and NPCs, new data included in **Fig. S8a**), indicating a basal genome composition around the nucleolus between different cell states. This result is also in agreement with studies on LADs between different cell types such as ESCs, NPCs, and terminally differentiated astrocytes (Peric-Hupkes *et al.*, 2010). We reason that the major reason of this difference between the Nucleolar-DamID and the biochemical purification of nucleoli is the sonication step since it is very well known that crosslinked cells need more sonication power to fragment the genome and the open ESC genome might be particularly sensitive to this treatment. This further highlights the need of NAD methods that can be performed in all cell types. The nucleolar-DamID has clearly shown this property. We included some part of this discussion in the manuscript.

3. FISH.

Undoubtedly, FISH is far below the standards in the field. It is insufficient, especially for independent validation of a new methodology. Far more (e.g. 5-10 onlyNADs/NAD-LADs/onlyLADs/iNADs each for ESCs and NPCs - correlated with Dam signal intensities - would be required to get a clearer picture of Dam signal vs. FISH-based nucleolus association frequencies and of the accuracy of onlyNAD vs. onlyLAD specification by Nucleolar-DamID.

Important technical points are that dehydration with graded ethanol concentrations up to 100% ethanol and air drying, further denaturation for 8 min at 83 °C were used in the FISH protocol. It must be shown that the structure of the nucleus is not damaged under these harsh conditions. Perhaps milder conditions should be chosen under which the 3D structures are better preserved. In Figure S2d “The signal of the LAD probe was located close to NL in the majority of analyzed ESCs (27/33)”. How is “close” defined? For LAD FISH

quality control, IF staining of nuclear lamina would be required and either distance measurements or counting of overlapping FISH/IF signals should be presented.

Author. This revised version included additional FISH data, all supporting the robustness and accuracy of the Nucleolar-DamID. We would like also to clarify that the data of the Nucleolar-DamID have already been validated by the HiC-rDNA. Moreover, the Nucleolar-DamID identified as NADs all the sequences found to interact with rRNA transcripts with the SPRITE method, a subclass of NADs.

In this revised version (**Fig. 2** and **Fig. S5 and S7**), we included the analysis of genomic regions classified as NAD-only, LAD-only, and iNAD/iLAD by the Nucleolar-DamID (6 probes). Importantly, two validated NAD regions (Chr. 1 and 5) were not detected as NAD by the nucleolar purification-based methods (Bizhanova et al). The DNA-FISH analyses validated all the identified NADs and showed that 70-90% of cells have these NADs contacting the nucleolus. The DNA-FISH analysis of Bizhanova et al. further highlights the accuracy of our Nucleolar-DamID. In Bizhanova et al., the FISH probe pPK1006 corresponding to Chr7: 12834113-13005735 showed high frequency contacts with the nucleolus by DNA-FISH. However, as stated by the authors, this was not identified as NAD by the biochemical purification of the nucleoli. In contrast, these sequences were detected as NAD by the Nucleolar-DamID. We also validated ESC- and NPC-specific NADs (**Fig. S7**). Thus, the DNA-FISH analyses included in this revised version further supported the accuracy of the Nucleolar-DamID in identifying NADs at much higher accuracy than previous methods. We clarified this point in the Discussion.

4. rDNA-HiC

Discussion of the initial rDNA-HiC datasets from two human cell lines (erythroleukemia K562 and lymphoblastoid cells, Yu and Lemos, 2016, *Genome Biol Evol*) and genome-wide 4C maps of mouse rDNA-genome interactions in undifferentiated and differentiated lymphoma cells (Diesch et al, 2019, *Comms Bio*) is missing from the manuscript. More importantly, an extended study (Yu and Lemos, 2018, *PLoS Genet*) describing comprehensive rDNA-HiC analyses including long-range rDNA interaction maps of H1 embryonic stem (ES) cells and four differentiated cell types derived from H1: Mesendoderm (ME) cells, mesenchymal stem (MS) cells, neural progenitor (NP) cells and trophoblast-like (TB) cells, also remains unaddressed. Given that Diesch et al, 2019, *Comms Bio* and Yu and Lemos, 2018, *PLoS Genet* are cited in the authors' recent review paper (in manuscript ref 11), it is not understandable that these highly relevant studies are not mentioned at all in the manuscript. Comparative analyses of mouse rDNA-4C (Diesch et al) and rDNA-HiC (this manuscript), as well as human (Yu and Lemos 2018) vs. mouse (this manuscript) ESC/NPC rDNA-HiC maps would further improve the study, but the cited studies and their main findings should at least be discussed.

Author. In the manuscript, we cited and discussed all the publications that used biochemical purification of nucleoli to map NADs and data from SPRITE. In this revised version we cited the work of Yu and Lemos but not the 4C analyses described in Dietsch et al. The reviewer should consider that a direct comparison with the data of Yu and Lemos and Dietsch *et al* is technically impossible for the reasons listed below.

1. The 4C analysis of Dietsch *et al* identified rDNA interactions in a mouse MYC-driven lymphoma model. Our work analysed mouse ESCs and NPCs. The work of Diesch *et al* mainly focused on the role of UBF and the regulation of gene expression in this cancer model whereas our data are mainly dedicated to describe the structure of chromosomes around the nucleolus. Consequentially, the data cannot be directly compared. Further, Dietsch *et al* stated that the rDNA contacts “*did not show a bias towards the rDNA-containing chromosomes 12, 15, 18, and 19 in the Eμ-Myc C57BL/6 genetic background but were uniformly distributed across all chromosomes*”. There was no further discussion on this result since this was not the focus of their work. However, this is an unexpected result since rDNA contacts on chromosomes containing rRNA genes are *cis* interactions and the 4C analysis should have revealed a high frequency contact at these chromosomes that was very evident in our HiC-rDNA analyses and published SPRITE data. We have also reanalysed their data and found that the distribution and the number of rDNA contacts obtained by 4C did not show the expected enrichment of rDNA contacts at the centromeric proximal regions. This

enrichment was found in our HiC-rDNA analysis and is consistent with SPRITE data and the well-known co-localization of centromeres in the vicinity of the nucleolus. Thus, for the reasons described above, we decided to not discuss the work of Diesch *et al* in this manuscript.

2. Lemos work pioneered the use of HiC to detect genomic contacts with rRNA genes in human cell lines. However, our HiC-rDNA is a completely different method (this point is described later in more details). Consistent with our results, Lemos' work revealed the enrichment of rDNA contacts in the repressive B compartment. We have now reported this information in this revised manuscript. However, we cannot provide and discuss additional information between the two HiC analyses for the following reasons. First, our HiC-rDNA is based on a completely different bioinformatic pipeline. While Lemos lab selected contigs by blasting rDNA sequences and clustering the data at 1 Mb resolution, our HiC-rDNA method is based on the alignment of the contigs to a modified reference genome that contains one copy of rDNA at the end of chromosome 12 and clustering of the contact at 5 kb resolution. Second, the data of Yu and Lemos mainly described the human LCL and K562 cell lines. The HiC data of the H1 embryonic stem (ES) cells and the four differentiated cell types derived from H1: Mesendoderm (ME) cells, mesenchymal stem (MS) cells, neural progenitor (NP) cells and trophoblast-like (TB) cells had low reads and, as clearly stated by the authors, no statistical analyses could be performed. The only information that was provided for ESC/MS/NP/TB is the number of reads, without any further discussion. Notably, Lemos' work did not contain information about the distribution of rDNA contacts along the chromosomes and whether chromosomes containing rRNA genes had higher rDNA-contacts, which are a key result of our analyses. Further, they did not provide data on gene density, gene expression, and histone modifications. Finally, to our surprise, this work did not provide the coordinates of the identified rDNA contacts making impossible the analysis of these data by others. Thus, for the reason listed above, we did not provide a detailed discussion of this work in our manuscript since we do not know what and how to compare these data with ours.

5. H3K9me2 and H3K9me3 mapping in ESCs with induced rDNA heterochromatin formation.

The experiments shown in Fig. 4 and S4 (ESC pRNA transfection) interrupt the flow of the manuscript and are only weakly connected to the other experiments in the manuscript. However, they are highly interesting and open up a new direction that could be elaborated in an independent study. Fascinating questions that arise here: What factors mediate the heterochromatin formation of NADs, which is first induced by pRNA on rDNA? Is physical proximity, direct contact necessary for spreading? What determines that H3K9me2 and H3K9me3 occupancies change at rDNA and satellite repeats, while only H3K9me2 changes in NADs? The possible role of the recently published MiCEE complex (Singh et al, 2018, Nat Genet) could also be assessed in the extended study. Addressing these questions could be an interesting stand-alone study, building on the experiments shown in the manuscript. They could be then removed from here.

Author. We think that these data well integrate in the manuscript and provide functional information about the nucleolus in genome organization. Further, they provide additional strong evidence that the NAD profile we obtained is highly specific as evident by the increase of H3K9me2 at regions adjacent to NADs whereas LAD-only were not affected. This is a key result of the manuscript.

6. NAD coverage

Nucleolar-DamID identifies >70-80% of some chromosomes as NADs according to the colour code in Figure S2 (e.g. Chr. 1, Chr. 3, Chr. 7). These chromosomes do not have nucleolar organiser regions containing ribosomal DNA. Are they nevertheless almost completely associated with nucleoli? It is not clear how the much lower NAD coverage values (in Figure S5) are calculated and how they relate to the colour codes shown on the maps in Figure S2? How should these Nucleolar-DamID maps be interpreted?

Author. The values of NAD coverage for each single chromosome were shown in **Figure 2b** and do not correspond to the values reported by the Reviewer (70-80%). Figure S2b (now **Fig. S3b**) displayed only NAD tracks of ESCs obtained with Nucleolar-DamID and sequences corresponding to nucleolar hubs identified by the SPRITE method. For clarity, we modified **Figure S3b** by adding the peak profiles calculated

with the established DamIDseq pipeline to visualize the statistically significant NADs as done for **Figure 2a**. The same pipeline was used to calculate NAD coverage in NPC shown in **Figure S8b**. Coordinates of NADs were listed in the Supplementary tables.

7. Comments on the discussion

The discussion would need to be thoroughly refined. It provides a biased view of published studies in several places and it also proposes some purely speculative scenarios that are not or only marginally supported by the data presented.

Author. We kindly disagree with this criticism. Our detailed response can be found here below.

Page 15-16 lines 18-28, 1-4 “Previous methods for NAD identification used sonication-based biochemical purification of nucleoli, a methodology that is subjected to a certain variation in nucleoli preparation between different cell types and relatively biased toward sonication-resistant heterochromatin 18-20,22,23,45. Relative to this methodology, the application of Nucleolar-DamID and HiC-rDNA has no bias for any kind of chromatin state and does not vary between cell types, thereby allowing direct comparisons of chromosome architecture around the nucleolus between distinct cell states.”

Nucleolar-DamID has not yet been shown to be free of bias and not to vary between (which?) cell types and cell states.

Author. The NAD profiles included in the new **Figure S1** unambiguously showed that nucleoli purifications vary between labs (ESC-NAD: Bizhanova et al, vs Lu et al.) and cell types (ESC-NAD Bizhanova et al. vs MEF-NAD Verti et al., same lab). We think that our statement is correct since the sonication-free Nucleolar-DamID has no bias toward heterochromatin as in the case of NADs from purified nucleoli. Further, the Nucleolar-DamID can be applied to every cell type as evident by the similarity of ESC- and NPC-NAD profiles. This was an expected result since a similar composition of LADs in ESCs, NPCs, and terminally differentiated astrocytes was previously reported (Peric-Hupkes et al., 2010), indicating that the chromosome architecture at the nuclear lamina and the nucleolus is globally conserved between different cell types. In contrast, this similarity in NAD composition between cell types could not be found in previous NAD analyses using nucleoli purified from ESCs and MEFs that showed only 40% common NAD sequences (Bizhanova et al. and Verti et al). We included this information in the Discussion.

In addition to the critical aspects of Nucleolar-DamID described in point 1, this assay has the other disadvantages:

i. Knock-in cells and cell clones must be generated.

Author. This is not a limitation. We established a system where the generation of stable clones can be obtained in 2-3 weeks.

ii. H2B-Dam-DD cannot be used simultaneously with H2B-Dam-NoLS-DD in the same cells as an internal control.

Author. To our knowledge, no method for LAD or NAD identification used simultaneous identification of control and the protein of interest. Thus, we do not understand the reason to list this point as specific disadvantage of Nucleolar-DamID.

iii. It must be ensured that the concentrations of H2B-Dam-NoLS-DD are comparable in each cell of the clonal population. Despite integration of coding sequences at the same locus and clonal selection, this is not trivial and the level of variation could also be different in different cell lines.

Author. We did not encounter this problem. Using our system, all clones that we generated had similar transgene expression.

iv. As in iii. for H2B-Dam-DD expressing cells.

Author. As in iii.

v. If application in different cell types and cell states is desired, it must be ensured that the steady-state levels of H2B-Dam-NoLS-DD and H2B-Dam-DD are comparable in each setting (and as mentioned above,

also homogeneous in the cell population). To overcome these drawbacks, a set of quality controls needs to be established. At this premature stage, before the controls are presented, the validity of the assay is greatly overestimated.

Author. All these controls were provided in the previous version of this manuscript. We stated that the ESC lines derived from a single cell and not from a cell population (**Suppl. Fig. S2**). The transgenes were inserted in the *Rosa26* locus and we provided data (**Fig. 1g**) showing that H2B-Dam and nucleolar H2B-Dam are expressed at similar levels.

The disadvantages of nucleolus isolation:

i. The protocol needs to be adapted for different cell types.

Author. Indeed, this is a big disadvantage that makes very difficult the comparison between different cell lines as evident by the profiles showed in the new **Figure S1**. The advantage of the Nucleolar-DamID is that does not require extensive adaptation for the application in different cell lines as in the case of biochemical purification of nucleoli. Indeed, we used the same protocol for ESCs and NPCs.

ii. Co-sedimentation of chromatin can lead to non-specific background. To overcome these disadvantages, the purity of the nucleolus fraction must be assessed microscopically and nucleolar protein and rDNA enrichment analyses must be performed by western blot and qPCR, respectively. They allow quantitative comparative analyses between different preparations. Co-sedimented chromatin can be separated by FACS sorting, which has been successfully used for nucleolus isolation in Arabidopsis (Pontvianne et al, 2016, Cell Rep; Pontvianne et al, 2016; Carpentier et al, 2018, Methods Mol Biol). At the current stage of method development, the quality controls of nucleolus isolation techniques are better established and the assay appears to be less bias and error prone than Nucleolar-DamID.

Author. The co-sedimentation does not solve the problem of the sonication that is key to identify all NADs with nucleoli purification. The microscopy and nucleolar enrichment analysis are a qualitative check for the nucleolus but not for the isolated genomic regions. The works cited by the Reviewer were from Arabidopsis. To our knowledge this methodology has never been applied in mammalian cells. Finally, the new data in **Figure S1** undoubtedly prove that it is the nucleolar purification that is *bias and error prone* not the Nucleolar-DamID!

rDNA-HiC, a method introduced by the Lemos lab, has the main disadvantage that it cannot provide a complete picture of NADs. Long-range interactions of chromosomal regions other than rDNA, such as NOR-adjacent sequences and centromeres, may also be involved in nucleolar-associated genome organisation and are not detected by rDNA-HiC. This disadvantage is an inherent property of the method and cannot be overcome.

Author. This is exactly what we stated in the Results and Discussion. Our HiC-rDNA or the SPRITE method can only detect a subclass of NADs that contact rRNA genes or rRNA transcripts, respectively. We used the HiC-rDNA to further validate our Nucleolar-DamID that identified all NADs, including the rDNA-contacts. Further, we would like to clarify the HiC-rDNA provided important information, including interactions between different chromosomes that both Nucleolar-DamID and nucleoli purifications cannot detect.

“We found some similarities between NAD maps obtained with previous nucleoli purifications and our methodologies, such as repressive chromatin features and enrichment in olfactory receptor and zinc finger genes 18,19. It is important here to note that major and minor satellites, which composed centric and pericentric domains and known to be in contact with the nucleolus, could not be identified by Nucleolar-DamID due to the lack of GATC motif. However, Nucleolar-DamID and HiC-rDNA revealed further and novel details in chromosome architecture around the nucleolus that did not emerge from previous studies.”

It cannot yet be ruled out that some of these novel details are due to DamID capturing of chromosomal regions that have a low frequency of contact with the nucleolus. As suggested above, extensive FISH

validation would be required, which should then include analysis of multiple NADs identified only with Nucleolar-DamID.

Author. The Nucleolar-DamID was analysed with an established bioinformatic pipeline for DamIDseq and we applied high stringency parameters to identify statistically significant NADs (high frequency contacts). The DNA-FISH analyses further demonstrated that the identified NADs are indeed contacting the nucleolus at very high frequency. Further, we demonstrated that the Nucleolar-DamID could identify NADs that were not mapped by the nucleolar purification-based method. Finally, all the downstream analyses supported the high quality of the data and are consistent with our HiC-rDNA and published data including SPRITE.

“Further, NAD maps provided in this work can be integrated in future studies of 3D genome organization that have so far did not take into account the nucleolus because of the lack of genome-wide NAD profiles” ??? There is no lack of genome-wide NAD profiles. They are described in the cited literature.

Author. To our knowledge, the works with published NAD profile using nucleolus purification are only cited in Reviews (including ours) or in the Introduction of original papers to describe the repressive chromatin states of NADs. To our knowledge, these published NAD profiles have not yet been used in data analyses by other studies. We modified the sentence to clarify this point.

“... future studies of 3D genome organization that so far did not take into account the nucleolus...” - this should be rephrased.

Author. We corrected the sentence.

Page 16 lines 17-19. “Thus, contacts with the nucleolus, even for sequences able to interact with the NL, shape a distinct repressive chromatin state.” The specific features of this shaped distinct chromatin state should then highlight the nucleoli and nuclear lamina. This could be demonstrated by using multiplex IF, where the main determinants of the specific chromatin state should be visualised simultaneously.

Author. Multiplex IF cannot provide quantitative information as done with our analyses. These are complex image experiments and go beyond the scope of our work

Page 17 lines 9-14 “Further, we identified cell type specific NADs, unique to ESC or NPCs, and showed that the position of a NAD or a LAD relative to the nucleolus or NL in ESCs correlates with their cellular location upon dissociation from the nucleolus or NL during differentiation. In other words, a NAD that interacts with the nucleolus but not with the NL in ESCs, after losing contacts with the nucleolus upon differentiation, still maintains features that makes it refractory to associate with the NL.”

Positions and cellular location were only shown for one NAD and one iNAD region by FISH. DamID does not provide information on position. The association frequencies of several selected example regions should be measured by FISH, as mentioned above. If a NAD loses contact with the nucleolus during differentiation due to the change in its chromatin state, then the lack of NL association could be due to the acquisition of new chromatin properties rather than the retention of some of the old ones. Both speculative scenarios and their combination are conceivable.

Author. We validated by DNA-FISH ESC-specific NADs in ESCs and NPCs (**Fig. S7**).

The reviewer misinterpreted our statement in the Discussion and the data described in the Results section. We suggested that there are some features that let remember genomic domains with which compartment they like to interact with upon detachment from the nucleolus or NL. (i.e. “*In other words, a NAD that interacts with the nucleolus but not with the NL in ESCs, after losing contacts with the nucleolus upon differentiation, still maintains features that makes it refractory to associate with the NL.*”). We only stated that the results are consistent with our analysis of chromatin modifications of NAD-only, NAD/LAD, and LAD-only.

Page 18 line 3 “our work established novel methodologies to identify NADs in every cell type” Without controls and testing, this is an overstatement for Nucleolar-DamID. Importantly, Nucleolar-DamID cannot be adapted for in vivo settings, the ultimate objective for biological studies.

Author. Our data clearly showed that the Nucleolar-DamID protocol does not require adaptation for each cell types whereas the nucleolus purification does it. We do not need to prove this using thousands of cell types since the data from ESC and NPC clearly proved this. The Nucleolar-DamID can be adapted to in vivo setting as done for the Lamin B1-DamID. However, this point was not all discussed in our manuscript, and we do not understand why this criticism was raised by the Reviewer.

What is (are) the other novel methodology (methodologies)?

Author. Nucleolar-DamID and HiC-rDNA. We would like to clarify that our HiC-rDNA method completely differs from the previously HiC studies since we used a completely different pipeline as described above. This makes it a new method.

Given the inherent properties of the methods briefly summarised above, the method of choice should rather be biochemical purification of nucleoli in combination with FACS, the method called FaNoS (fluorescence-activated nucleolar sorting) established in Arabidopsis (Pontvianne et al, 2016, Cell Rep; Pontvianne et al, 2016; Carpentier et al, 2018, Methods Mol Biol) and/or highly multiplexed FISH. FaNoS would also work in animal tissue samples.

Author. To our knowledge, FaNoS method has only been applied in Arabidopsis. Therefore, the statement of the reviewer clearly remains a speculation and cannot affect the robustness of our Nucleolar-DamID.

Page 18 lines 6-8 “We predict that the application of Nucleolar-DamID and HiC-rDNA will be relevant for future works, including the understanding of the determinants for the specific targeting to nucleolus or NL or the application of single cell analyses that were prohibitive with the previous biochemical based-methods.” Single cell/single nucleolus analyses would be possible with biochemical based FaNoS.

Author. We understand that the Reviewer likes FaNoS technique, but this is not the subject of this work. Further, we would like to correct the Reviewer. Since the majority of mammalian cells contains more than one nucleolus, FaNoS can only be applied for single nucleoli analysis, but not for single cell.

In summary, while NAD mapping and rDNA-HiC analyses are not new, the authors present a novel method called Nucleolar-DamID assay. However, this method is not yet well established, controls and benchmarking have yet to be presented. The topic of differentiation-dependent dynamics of the nucleolar-associated genome is still relevant with many open questions, although it has been partially investigated by rDNA-HiC in human cells and indirectly by NAD analyses in mouse cells. Due to the insufficient technical quality, which is more due to the choice of method than to the quality of the overall performance, I can only recommend the manuscript for publication if the above-mentioned concerns are satisfactorily addressed.

Author. All the points were discussed in detail in the previous sections. We would like to clarify again that the *topic of differentiation-dependent dynamics of the nucleolar-associated genome* was not analysed in such details by previous studies. All the results contained in Figures 2 to 7 supported the quality of the generated NAD profiles using two novel methods that complement to each other and are consistent with the published literature. The additional FISH analyses supported the accuracy of our data and proved that the Nucleolar-DamID is superior to the previous nucleolus purification-based methods since it identified as NADs genomic domains that were not detected as NADs by previous studies.

Reviewer #2 (Remarks to the Author):

The genomic loci associated with the nucleolar periphery (known as nucleolar-associated domains, or “NADs” for short) are of high interest to the study of genome organization. Previous techniques for NAD analyses include biochemical isolation of nucleoli, DamID using a nucleolar targeting peptide, or TSA-Seq, an antibody-based proximity labeling method (PMID: 30154186, 33446254). Here, the authors have devised a novel DamID-based method for genome-scale detection of NADs, fusing the bacterial Dam adenine methyltransferase to histone H2B plus a nucleolar targeting sequence. DamID is an appealing idea, since it can in principle be applied to all cell types, can be performed on fewer cells than required for biochemical purification (and possibly be extended to single cell approaches as has Lamin-DamID), and is independent of antibodies that can vary in quality. Additionally, the experimental manipulations via differentiation and pRNA treatment make this a valuable study. However, there are features of this paper that need more consideration, as discussed below.

1. As an alternative method for NAD mapping, the authors mine Hi-C datasets for rDNA-containing junction fragments. Since the field would benefit from comparison of different methods, it is a missed opportunity that there is no side-by-side comparison of NADs mapped by the DamID method and the results obtained via the Hi-C analyses.

Author. We are sorry to not have sufficiently clarified this point. The main purpose to establish HiC-rDNA analyses was to find a way to validate the Nucleolar-DamID results with another dataset. The HiC-rDNA identifies only a subclass of NADs that interact with rRNA genes whereas the Nucleolar-DamID represent all NADs. Accordingly, a large fraction (>80%) of rDNA-contacts identified with HiC-rDNA were defined as NAD by the nucleolar DamID and NADs identified by the Nucleolar-DamID have high contact score defined by HiC-rDNA (**Fig. 2g,h**). The results of these two methodologies showed similar results such as (1) enrichment of NADs at the proximal centromeric regions (**Figs. 2b,e,f**), (2) the enrichment of contacts at chromosomes containing rRNA genes (**Figs. 2b,e,d**), (3) the low gene density and low expression state (**Fig. 3 and 5**), the increase in contact frequency with the nucleolus in NPCs relative to ESCs (**Fig. 5 and S8b**), the lack of gene expression changes upon detachment from the nucleolus (**Fig. 7, Fig. S10a,b**), and the corresponding enrichment in GO terms linked to neurogenesis (**Fig. 7b, Fig. S10c,d**). We have further clarified these points in the discussion.

Furthermore, nothing is done here to test whether these methods are providing better maps of NAD associations than achieved in other studies. For example, how do the DamID and Hi-C-based peaks obtained in this study compare to those in previous biochemical analyses of mouse ES cell NADs (PMID: 32160538 → Lu et al Cell Report; 32219510→ Bizhanova.)

Author. To better clarify the difference in NAD profiles obtained by previous studies and ours work, we include a new **Figure S1** showing IGV tracks of NADs from purified nucleoli (Bizhanova *et al*; Lu *et al.*) and NADs from the Nucleolar-DamID (see also image below).

In this revised version, we included new DNA-FISH analyses (**Fig. 2, Fig. S4 and S7**) that not only validated the NADs identified by the Nucleolar-DamID but also demonstrated that the method is superior to the previous nucleolus purification-based methods since it identified as NADs genomic domains that were not detected as NADs by the previous studies. We clarified this point in the revised manuscript.

As stated in the Introduction, the two previously published NAD profiles from Bizhanova *et al* and Lu *et al* extensively differ from each other (i.e. NAD genome coverage 7.5% in Lu *et al.*, and Bizhanova *et al.* 30%), highlighting the difficulties to standardize nucleoli purification even from the same cell type. The nucleolar-DamID will not encounter this problem. This is also evident by the similar, as expected, NAD profiles between ESCs and NPCs (ca. 85% common NADs, see also new **Fig. S8a**). This result is also in agreement with the reported similar LAD composition between different cell types such as ESCs, NPCs, and terminally differentiated astrocytes (Peric-Hupkes *et al.*, 2010), indicating a basal genome composition around the nucleolus and at the nuclear lamina between different cell types. In contrast, previous NAD profiles

generated with nucleolus purifications (Bizhanova et al., Verti et al.) showed that only 40% of NADs sequences are in common between ESCs and MEFs, indicating that the NAD profiles between these two cell types is remarkably distinct. We have further clarified this point in the discussion.

2. Those DamID/Hi-C comparisons could be useful in dealing with this question: There are clear increases in the number of Hi-C-rDNA contacts and H2B-NoLS-DamID signals in the differentiated NPC cells compared to the undifferentiated ES cells (Fig. 5c-d). This is the expected result since it is well established that differentiation leads to increased levels of heterochromatin, and previous analyses showed many fewer rDNA junction fragments in human ESCs than in human somatic cells (PMID: 29570716→ Yu & Lemos PLoSGen). Therefore, it is counter-intuitive that genomic NAD coverage would be higher in the ES cells than in NPCs (Fig. S5b).

Author. We would like to clarify that we reported that the contact frequency measured by HiC-rDNA and Nucleolar-DamID is higher in NPCs than in ESCs (Fig. 5). However, we also found that NAD coverage and

the number of unique rDNA contacts are higher in ESCs than in NPCs. As stated in the manuscript, we suggested that these results indicate that *the architecture of chromosomes surrounding the nucleolus in NPCs is in a more compact, rigid form, making fewer but more frequent contacts with rRNA genes. In contrast, the structure of chromosomes around the nucleolus of ESCs appears more flexible, establishing more but less frequent contacts.* These results are consistent with previous reports showing that ESCs harbour a more open and dynamic chromatin than differentiated cells. In this revised manuscript, we further clarified this point in the results section.

We would also like to clarify that the work referred by the Reviewer (Yu and Lemos, PLoS Gene) mainly described the human LCL and K562 cell lines. The HiC data of the human H1 ESC line had low reads and no statistical analyses could be performed as stated by the authors “...greater coverage for these cells is necessary to draw sufficiently dense contact maps for a more fine-grained and meaningful biological contrast.” Further, Yu and Lemos did not measure or discuss the differences between ESC and differentiated cells but only general features of rDNA-contacts measured with the LCL and K562 cell lines such as the enrichment in B compartment. We have now cited this work in the discussion section.

Additionally, previous comparisons of mouse ES and somatic fibroblast NADs detected via biochemical isolation indeed detected a greater genomic NAD coverage in the somatic cells (PMID: 32219510 → Bizhanova). Does this mean that NPCs have fewer NADs than do fibroblasts?

Author. Bizhanova *et al.* described that the NAD coverage in ESC is smaller than in MEFs (31% vs 40%). However, Bizhanova *et al.* did not provide any information on contact frequency. Further ESC and MEFs showed only 40% of common NADs, an unexpected result since LAD (Peric-Hupkes *et al*) and NAD (our work) composition showed a common, global chromosome architecture around the nuclear lamina and the nucleolus between different cell types (ESCs, NPCs, and terminally differentiated astrocytes for LADs). Since the methods to detect NADs are different, we cannot conclude whether NPCs have fewer NADs than fibroblasts.

An alternative possibility would be that NAD peaks are too liberally assigned from the DamID data in the current study. That is, in ES cells with a more mobile genome (e.g. PMID: 16399082, → Meshorer *et al* 2006; PMID: 24831699 → Boskovic *et al*_G&D), broadly distributed but transient interactions (e.g. Fig. 5e) could be recorded as contacts by the Dam methyltransferase.

Author. We assigned NAD by performing damidseq pipeline using high stringency parameter (FDR <0.01 and min_quant 0.70; only the significant peaks common to both replicates were considered as NADs for further analysis). Further, HiC-rDNA and Nucleolar-DamID revealed similar results, showing that the structure of chromosomes around the nucleolus of ESCs appears more flexible, establishing more but less frequent nucleolar contacts. These results are consistent with the notion that the genome of ESC is dynamic. The novelty here is that we describe it from the nucleolus' point of view. Finally, the new DNA-FISH data validated NADs identified by the Nucleolar-DamID in both ESCs and NPCs.

Therefore, independent validation of NAD peaks in the mES cells would be valuable. This would be best tested via DNA-FISH studies as an orthogonal method. I note that only a single NAD locus is tested here by this method, and comes from the highly nucleolar-associated, rDNA-containing chr19 (Fig. 2i), and therefore analysis of a more diverse set of loci is warranted. Alternatively, would the Hi-C analyses provide an alternative metric by which to validate the DamID data?

Author. As described above, the results from the HiC-rDNA are consistent with the nucleolar-DamID. We have further clarified these points in the discussion. As suggested, this revised version included additional DNA-FISH analyses in both ESCs and NPCs (**Fig. 2, Fig. S4 and S7**). These data not only validated the NADs identified by the Nucleolar-DamID but also demonstrated that the method is superior to the previous nucleolus purification-based methods since it identified as NADs genomic domains that were not listed as NADs by previous studies.

3. What are the spots of H2B-NoLS-GFP in Fig. 1d that do not appear to localize with the main nucleolar mass? There needs to be co-staining with a fiduciary nucleolar marker protein. If these spots are not nucleolar, what are they? The concern is that these could influence the designation of NAD regions in unforeseen ways.

Author. We apologize for the inaccuracy in not having highlighted all the nucleoli present in the image of H2B-NoLS-GFP of the old **Fig. 1d**. Although DAPI helps in visualizing nucleoli, some small nucleoli cannot be clearly identified by DAPI. Thus, we followed the suggestion of the Reviewer and replaced these images using NPM1 and Fibrillarin as nucleolar markers (new **Figs. 1c,e**). These results supported the nucleolar localization of H2B-NoLS-GFP as shown in the life cell images of **Fig. 1b**.

4. The analysis of histone modifications at NAD boundaries is interesting. The changes in the H3K9me2 levels flanking NADs is clearly elevated in the ES cells treated with pRNA. What happens to H3K27me3 at the boundaries under the same conditions? And what happens to both these marks at boundaries upon differentiation into NPC cells?

Author. We did not perform H3K27me3 ChIPseq in ESC+pRNA since the results showed that H3K9me2 is a major feature of NADs and NADs are depleted of H3K27me3. Further, it is well described in the literature that pRNA-mediated heterochromatinization acts through NoRC complex that is associated with H3K9me2/3 activities. In this new revised manuscript, we provided boundaries analyses of NADs in NPCs using published datasets that showed a sharp transition between active and repressed chromatin domains (**Fig. S8d**).

5. The ability to view and analyze the genomic data at GEO was not sufficient. There are xlsx files of called peaks, but no availability was made of the raw DamID replicate datasets. The DamID score files (GSE150822_ESC_NADs_Nucleolar-DamID_score.csv.gz and GSE150822_NPC_NADs_Nucleolar-DamID_score.csv.gz) don't have any inserted files when opened.

Author. We are sorry for this inconvenient. However, the files do not seem to have any problem as we were able to open them directly from the GEO and contained data. Further, these data were also included in the Suppl. Tables S1 and S4.

6. Finally, more needs to be done to acknowledge previous contributions in this area. For example, use of rDNA crosslinks from Hi-C studies was previously pioneered by Bernardo Lemos (PMID: 27797956 → Yu and Lemos GBE, 29570716 → Yu & Lemos PLoS Gen), and a different nucleolar-targeted DamID was reported by the van Steensel laboratory (PMID: 33446254 → SPIN paper).

Author. Lemos work pioneered the use of HiC to detect genomic contacts with rRNA genes. However, the methodologies that Yu and Lemos and our work applied to HiC data to identify rDNA contacts are very different. While Lemos lab selected contigs by blasting rDNA sequences and clustering the data at 1 Mb resolution, our HiC-rDNA method is based on the alignment of the contigs to a modified reference genome that contains one copy of rDNA at the end of chromosome 12 and clustering of the contact at 5 kb resolution. Lemos' work mainly analysed human cell lines LCL and K562 while the information on the human H1 ESC line and derive lineages were limited due to low sequencing depth that did not allow further downstream analyses. The HiC-rDNA of our work analysed mouse ESCs and NPCs using HiC of high resolution (<750 bp), the highest to date in mammalian cells. Similar to our results, Lemos' work revealed that rDNA contacts contain repressive domains. We have now included this information in this revised manuscript. However, we cannot provide and discuss additional data between the two HiC analyses since Lemos' work did not analyse the rDNA contacts in terms of histone modifications, gene expression, gene density, and contact distribution on chromosomes. Further, they did not compare ESCs and differentiated cells. Finally, the

coordinates of genomic contacts with rDNA were not provided. Consequently, we could not re-analyse these data and compare them with our HiC-rDNA results.

Wang et al. (PMID: 33446254) reported an integrative computational method to analyse genome-wide intranuclear chromosome positioning and nuclear compartmentalization relative to multiple nuclear structures. This work used a NAD profile generated by van Steensel lab using DamID in human K562 cells. However, to our surprise, no work has been published describing that the application of this Nucleolar-DamID type generates NADs. In other words, Wang et al. used a NAD dataset generated with a novel method that was not validated by orthogonal methods such as FISH, HiC, etc as we have done in this work. Consequentially, we are not discussing or citing this work.

Reviewer #3 (Remarks to the Author):

To the authors:

Remarks to the authors: Bersaglieri et al. in their manuscript "Genome-wide maps of nucleolus interaction reveal distinct layers of repressive chromatin domains" adapted the previously described DamID method to be able to map for the first time the chromatin domains (NADs) which interact with the nucleolus in ESC and after their differentiation into NPC. In addition, they provide new HiC-rDNA data about the frequency of chromatin interactions with the nucleolus. These findings are very interesting and address an important missing aspect of epigenetic regulation: the contribution of the nucleolus to the 3D organization of the chromatin in the eukaryotic nucleus. The data presented are of high quality and the paper is well-written. Therefore, I recommend the paper for publication in Nature Communications after addressing my minor comments

General Comments: The authors have chosen to normalize the H2B-Dam_NoLS to H2B-Dam to compensate for an eventual incorporation of the tagged H2B into chromatin regions outside of the nucleolus. They have shown many convincing controls to exclude this, however they did not consider the possibility that incorporation of H2B-Dam-NoLS due to its rather big size could interfere with rRNA expression and integrity of the nucleolar chromatin. In particular

1. in Fig. 1D they could show that localization of TTF1 in the presence of H2B-Dam_NoLS without ActD is not affected

Author: We included new IF analysis analyses (**Fig. 1c and e**) using NPM1 and Fibrillarin as nucleolar markers, which are also known to alter their cellular localization upon downregulation of rRNA transcription (caps for Fibrillarin and delocalization from the nucleolus for NPM1). The data supported the nucleolar localization of the H2B-NoLS and showed no alterations in NPM1 or Fibrillarin localization in cell expressing H2B-GFP-NoLS.

2. It would be important to show that H2B-Dam_NoLS does not affect rRNA expression by RT-PCR

Author: We measured 45S pre-rRNA in the ESC lines expressing H2B-Dam and H2B-Dam-NoLS upon Dox/Shield treatment. We did not find any significant changes in rRNA transcription. We also analysed the expression of pluripotency genes and found no alterations, indicating that the expression of the Nucleolar-DamID constructs does not affect pluripotency states. These new results have been included in the **Figure S2d**. Further, we provided images of ESCs wt and the H2B-Dam and H2B-NoLD-Dam ESC lines before and after Dox/Shield treatment showing that the morphology of the ESCs (i.e. compact and bright colonies) is not altered (new **Figure S2c**).

Minor Points:

1. I did not find anywhere (neither in the text or in the figure legend) explained what compartment A and B refer to. I suggest to shortly define them for the non `chromatin organization` experts.

Author: We included the description of A/B compartment in the Introduction.

2. In Fig S2A, panel b and c have been swapped and do not correspond to the description in the text

Author: Thank you! We corrected the text.

3. In Fig S4 the correlation coefficient like in Fig.4c is missing

Author: We included the correlation coefficient (now **Fig. S6**).

REVIEWER COMMENTS

Reviewer #1 (Remarks to the Author):

The key biological observations of the manuscript are that NAD composition is similar in ESCs and NPCs and that differentiation-dependent loss of nucleolus association can be linked to specific gene activation during neurogenesis. These results are consistent with similar observations on lamina-associated domains (Peric-Hupkes et al., 2010, Mol Cell). According to the authors, the most important advance is the novel methodology of NAD mapping, as highlighted throughout the manuscript (see e.g. Abstract: "...lack of methods for the identification of nucleolar associated domains (NADs)", "The methodologies here developed will finally make possible to include the contribution of the nucleolus in nuclear space and genome function in diverse biological systems."). Nucleolar-DamID/HiC-rDNA is proposed as the ultimate approach for future studies of genome organisation at the nucleolus. Given the published AP3-DamID (nucleolus contact data generated using Dam methylase fusion with a nucleolar targeting peptide repeat, Wang et al, 2021, Genome Biol, see also in reviewer #2's comments), the novelty of the method is severely compromised and warrants additional benchmarking, more thorough validation and improvement. Importantly, AP3-DamID, which lacks H2B but has a strong NoLS, performs at several times higher signal intensity than Nucleolar-DamID (log₂ ratios of m6A values in the range of +1/-1 for AP3-DamID-seq vs. -0.21/+0.21 for Nucleolar-DamID-seq).

Theoretical considerations that could explain the very low S/N ratio of Nucleolar-DamID were discussed in detail in the rebuttal. However, no attempt was made to experimentally improve the assay, e.g. by removing H2B and circumventing potential problems associated with it. The numbers still remain in the noisy -0.21/+0.21 log₂-range. FISH was improved (optimised protocol, 6 target regions in total) but still does not contain enough information to assess the accuracy of NAD detection and dynamics. The six selected regions are from different chromosomes. The resolving power could only be assessed if several adjacent NAD and iNAD regions (in pairs or as consecutive series), labelled with different colours, were analysed in the same cell. Sensitivity and accuracy in the detection of differentiation-dependent NADs and iNADs could only be critically evaluated when domains that dynamically change their contact during differentiation were analysed simultaneously with adjacent domains that do not, in a multiplex design, both in ESCs and NPCs. Singleplex analyses of 4 NADs (1 ESCs-specific, 1 NPC-specific, 2 general), 1 LAD and 1 iNAD/iLAD from different chromosomes do not provide sufficient validation. rDNA contacts mapped by HiC-rDNA should also be validated in combined rDNA-NAD-FISH.

The revised manuscript still gives a stronger impression of the methods than their proven analytical power actually is. Importantly, nearly nothing is known about the false positive and false negative rates of Nucleolar-DamID versus the other NAD mapping methods that use various high-throughput sequence analyses and diverse bioinformatics pipelines. It is unclear whether identifying more NADs means higher sensitivity or more false positives, or less NADs more false negatives. The dependence of HiC-rDNA on the availability of high-quality datasets also severely limits its applicability in future studies investigating NADs in different cells. These limitations of the methodology are still not clearly pointed out, which could give the reader misleading perceptions and expectations.

Despite the improvements and with appreciation of the value of the new NAD maps in mouse ESCs and NPCs, the novelty, quality and validation of the methodology are limited. The manuscript leaves open questions about potential false-positive and false-negative nucleolus contact annotations, applicability in vivo (not in vitro in cultured cells) even if the text promises a generally applicable precise methodology for the investigation of NADs in every cell type.

Reviewer #2 (Remarks to the Author):

The authors present a much-improved manuscript, and I agree that the technique highlighted is a novel and important addition to the field. There are still some concerns, as follows:

I think the reply to reviewer 1 on pg. 3 of the rebuttal "the suggested ChIPseq of the DamID constructs cannot be performed since their expression is extremely low and cannot be detected with antibodies" is insufficient. More modern approaches to genome-scale protein localization, i.e. Cut 'n' Run and Cut 'n' Tag experiments (e.g. PMID: 33846645, 33846646, 34637755) have been scaled down to the single cell level. Therefore, in population experiments, it seems quite likely that even lowly-expressed proteins should provide detectable signals, especially since the proteins in question are histones that are in very close proximity to the DNA. Therefore, such experiments should be attempted to address the fundamental question of where the Dam-tagged NoLS and control histones are localized across the genome.

Rebuttal, p. 6: "NoLS is the key to determining nucleolar association, but it does not guarantee exclusive nucleolar localisation. This is also evident from the H2B-GFP NoLS signals in Figure 1b, where a large proportion of the signal is nuclear." I think the author's reply is insufficient here. I agree that the NoLS provides a high degree of enrichment of nucleolar association, but I also agree with the reviewers' observation of some signal throughout the nucleus in Figures 1b, c, and e. I understand the authors' point that these IF experiments are performed with greater levels of expression than used in the DamID experiments. Therefore, the best approach to resolve the issue is to perform the Cut 'n' Run experiments suggested above, using the exact transgene constructs and levels of expression as used in the DamID analysis.

There has been previous description and analysis of what are called here "NAD only" regions. The authors should be commended for their very thorough analysis of the genomic attributes of these regions, but there should also be comparisons with the earlier studies that referred to such loci as "Type II" NADs (Vertii et al 2019; Bizhanova et al 2020) regarding the overlap with H3K9me3 and H3K27me3 and early replication timing.

Reviewer #3 (Remarks to the Author):

Reviewer#3

Remarks to the authors:

In general I find the paper from Bersaglieri et al. "Genome-wide maps of nucleolus interaction reveal distinct layers of repressive chromatin domains" very interesting and a valuable resource for the fields of chromatin organization and epigenetic regulation. Their findings provide solid novel information to better understand 3D chromatin organization and the underlying molecular mechanisms.

The data presented are of high quality and the conclusions have been further strengthened upon revision.

The authors have addressed all my points and concerns with the exception of my point 1, which I feel have been misunderstood. I highly recommend this article for publication in Nature Communication but I would like my point 1 to be experimentally addressed or eventually discussed if not possible to be performed.

General Comments: The authors have chosen to normalize the H2B-Dam_NoLS to H2B-Dam to compensate for an eventual incorporation of the tagged H2B into chromatin regions outside of the nucleolus.

They have shown many convincing controls to exclude this, however they did not consider the possibility

that incorporation of H2B-Dam-NoLS due to its rather big size could interfere with rRNA expression and integrity of the nucleolar chromatin. In particular

1. in Fig. 1D they could show that localization of TTF1 in the presence of H2B-Dam_NoLS without ActD is not affected.

Author: We included new IF analysis analyses (Fig. 1c and e) using NPM1 and Fibrillarin as nucleolar markers, which are also known to alter their cellular localization upon downregulation of rRNA transcription (caps for Fibrillarin and delocalization from the nucleolus for NPM1). The data supported the nucleolar localization of the H2B-NoLS and showed no alterations in NPM1 or Fibrillarin localization in cell expressing H2B-GFP-NoLS.

Reviewer: In the new Fig.1e the authors show that nucleolar localization of H2B-GFP, TTF1-GFP and H2B-NoLS-GFP is not affected by ActD treatment. According to the Figure legend description H2B-Dam-NoLS has not been cotransfected together with GFP-TTF1. This experiment does not answer my concern about the correct localization of TTF1 in the presence of H2B-Dam-NoLS without ActD treatment. This is important to make sure that the nucleolar chromatin organization is not affected by overexpression of the big fusion protein H2B-Dam-NoLS. As TTF1 antibody are available one could perform IF of endogenous TTF1 in cells transfected with H2B-Dam-NoLS.

2. It would be important to show that H2B-Dam_NoLS does not affect rRNA expression by RT-PCR

Author: We measured 45S pre-rRNA in the ESC lines expressing H2B-Dam and H2B-Dam-NoLS upon Dox/Shield treatment. We did not find any significant changes in rRNA transcription. We also analysed the

expression of pluripotency genes and found no alterations, indicating that the expression of the Nucleolar-

DamID constructs does not affect pluripotency states. These new results have been included in the Figure

S2d. Further, we provided images of ESCs wt and the H2B-Dam and H2B-NoLD-Dam ESC lines before and after Dox/Shield treatment showing that the morphology of the ESCs (i.e. compact and bright colonies)

is not altered (new Figure S2c).

Reviewer: Thank you for addressing this. It is now evident that overexpression of H2B-Dam-NoLS does not interfere with correct transcriptional regulation of rRNA expression

All my minor points have been addressed

REVIEWER COMMENTS

Response to the reviewers

In this revised version, we modified the text (highlighted in red) to include the comparison of NAD-only (our study) and NAD Type II (Bizhanova *et al.*) that was requested by Reviewer 2. Our response and clarifications to the reviewers' comments can be found below.

Reviewer #1 (Remarks to the Author):

1. The key biological observations of the manuscript are that NAD composition is similar in ESCs and NPCs and that differentiation-dependent loss of nucleolus association can be linked to specific gene activation during neurogenesis. These results are consistent with similar observations on lamina-associated domains (Peric-Hupkes *et al.*, 2010, *Mol Cell*). According to the authors, the most important advance is the novel methodology of NAD mapping, as highlighted throughout the manuscript (see e.g. Abstract: "...lack of methods for the identification of nucleolar associated domains (NADs)", "The methodologies here developed will finally make possible to include the contribution of the nucleolus in nuclear space and genome function in diverse biological systems."). Nucleolar-DamID/HiC-rDNA is proposed as the ultimate approach for future studies of genome organisation at the nucleolus. Given the published AP3-DamID (nucleolus contact data generated using Dam methylase fusion with a nucleolar targeting peptide repeat, Wang *et al.*, 2021, *Genome Biol*, see also in reviewer #2's comments), the novelty of the method is severely compromised and warrants additional benchmarking, more thorough validation and improvement.

Author. We apologize for the confusion. Although our work has established novel methods for the analyses of NADs, we think that the most important advance of our work are the novel findings showing how chromosomes are organized around the nucleolus in ESCs and derived NPCs. Accordingly, these results represent the majority of the data that were placed in 6 out of 7 main Figures (Figs 2-7) and 8 out of 10 Suppl. Figure (Figs. S3-10).

We kindly disagree with the Reviewer regarding the compromised novelty of our method. The NAD method described in Wang *et al.* is completely different from ours and it has been applied in human cells whereas our work analyzed mouse ESCs and NPCs. Further, and in contrast to Wang *et al.*, our method has been validated using orthogonal approaches such as DNA-FISH and HiC-rDNA.

2. Importantly, AP3-DamID, which lacks H2B but has a strong NoLS, performs at several times higher signal intensity than Nucleolar-DamID (log₂ ratios of m6A values in the range of +1/-1 for AP3-DamID-seq vs. -0.21/+0.21 for Nucleolar-DamID-seq).

Theoretical considerations that could explain the very low S/N ratio of Nucleolar-DamID were discussed in detail in the rebuttal. However, no attempt was made to experimentally improve the assay, e.g. by removing H2B and circumventing potential problems associated with it. The numbers still remain in the noisy -0.21/+0.21 log₂-range.

Author. The experimental setups of our Nucleolar-DamID and the AP3-DamID are very different and cannot be compared to m6A-ratio. The AP3-DamID has used as control GFP-Dam that is completely different from AP3-Dam. In contrast, H2B and H2B-NoLS used in the Nucleolar-DamID are very similar to each other and only differ in the nuclear localization. This is why the m6A ratio AP3-Dam vs GFP-Dam is higher than H2B-NoLS-Dam vs H2B-Dam. Moreover, Nucleolar-DamID and AP3-DamID use different pipelines and this can also impact on the final range of values.

We did not attempt to increase the m6A using the GFP-Dam as control since this was not requested. Further, we think that the use of GFP-Dam as control will increase the m6A ratio at the expense of specificity since GFP-Dam association with the genome is too different from H2B-NoLS-Dam also due to its propensity to be incorporated in open chromatin (Aughey *et al.*, 2018). Further, GFP-Dam cannot compensate the eventual incorporation of H2B-NoLS-Dam outside the nucleolus whereas H2B-Dam can do this. Thus, the H2B-Dam is the best control of the Nucleolar-DamID. Finally, in Dam-ID there is no definition of noise for the m6A ratio since this depends on the type of Dam-control fusion protein. We have shown that the obtained NAD profiles

have statistical significance and are specific as also supported by several orthogonal validations, including DNA-FISH, HiC-rDNA and published dataset such as SPRITE analyses.

5. FISH was improved (optimised protocol, 6 target regions in total) but still does not contain enough information to assess the accuracy of NAD detection and dynamics. The six selected regions are from different chromosomes. The resolving power could only be assessed if several adjacent NAD and iNAD regions (in pairs or as consecutive series), labelled with different colours, were analysed in the same cell. Sensitivity and accuracy in the detection of differentiation-dependent NADs and iNADs could only be critically evaluated when domains that dynamically change their contact during differentiation were analysed simultaneously with adjacent domains that do not, in a multiplex design, both in ESCs and NPCs. Singleplex analyses of 4 NADs (1 ESCs-specific, 1 NPC-specific, 2 general), 1 LAD and 1 iNAD/iLAD from different chromosomes do not provide sufficient validation. rDNA contacts mapped by HiC-rDNA should also be validated in combined rDNA-NAD-FISH.

Author. We apologize for the confusion. Our work did not deal with dynamics of NADs. We described two fixed states, ESCs and derived NPCs. Accordingly, the term “dynamics” is present only twice in our manuscript, one in the introduction and one in a title paragraph “Dynamics of genome organization around the nucleolus during ESC differentiation”. To avoid any misunderstanding, we have modified this title by removing the word “dynamics”. While the experiments concerning dynamic changes in contacts are very interesting, we believe they go beyond the scope of this work.

We performed DNA-FISH by precisely following the instructions of this Reviewer that asked us to use more probes to validate NADs in ESCs and NPCs and to include the validation of NADs identified only by the Nucleolar-DamID and not by the previous methods. The DNA-FISH analyses provided meet the gold-standard of the field in validating nuclear positioning of domains detected by genome technologies, this includes previous LAD works and previous methods based on nucleoli purification. Finally, as stated in the manuscript, the HiC-rDNA results largely overlap with the Nucleolar-DamID. Accordingly, the DNA-FISH probes showing proximity to the nucleolus where rRNA genes are located were also identified as NADs by the HiC-rDNA.

6. The revised manuscript still gives a stronger impression of the methods than their proven analytical power actually is. Importantly, nearly nothing is known about the false positive and false negative rates of Nucleolar-DamID versus the other NAD mapping methods that use various high-throughput sequence analyses and diverse bioinformatics pipelines. It is unclear whether identifying more NADs means higher sensitivity or more false positives, or less NADs more false negatives. The dependence of HiC-rDNA on the availability of high-quality datasets also severely limits its applicability in future studies investigating NADs in different cells. These limitations of the methodology are still not clearly pointed out, which could give the reader misleading perceptions and expectations.

Author. As stated in the manuscript “*We used a previously established DamIDseq pipeline (Marshall and Brand, 2015) and constructed genome-wide maps of NADs in ESCs by taking the resulting H2B-Dam-NoLS over H2B-Dam m6A ratio as a measure for the relative contact frequency of DNA sequences with the nucleolus (FDR <0.01)*”. We validated our data in many ways (HiC-rDNA, DNA-FISH, SPRITE). As already stated in the previous response to reviewers and manuscript, our DNA-FISH analyses have already validated NADs identified only by the Nucleolar-DamID but not by other previous methods. Further, as discussed in the previous Reviewers’ response, we identified with the Nucleolar-DamID NAD regions that were validated for nucleolar proximity using DNA-FISH by Bizhanova *et al.* but that were not mapped as NAD by their biochemical purification of nucleoli. Thus, we believe that the Nucleolar-DamID is a robust method that does not show any evident disadvantage compared to previous methodologies. Finally, while we are very much convinced that our method is superior to previous methods, in the manuscript this opinion is very much mitigated. Indeed, in the discussion, we have highlighted several similarities between our and previous works on NADs.

It is not very clear to us why the use of HiC-rDNA is severely limited by the dependency on the availability of high-quality datasets. This is a limitation for the use of any dataset that must be of high quality otherwise they cannot be used. We are currently applying HiC-rDNA to other high-quality HiC data, and it works very well.

7. Despite the improvements and with appreciation of the value of the new NAD maps in mouse ESCs and NPCs, the novelty, quality and validation of the methodology are limited. The manuscript leaves open questions about potential false-positive and false-negative nucleolus contact annotations, applicability *in vivo* (not *in vitro* in cultured cells) even if the text promises a generally applicable precise methodology for the investigation of NADs in every cell type.

Author. These points have already addressed in the previous sections and extensively described in the previous reviewers' response. We would like to clarify that the future applicability *in vivo* was not mentioned in our manuscript. However, we are quite positive that *in vivo* application could be possible with the Nucleolar-DamID since the DamID method has found its application in many systems, including *in vivo* systems.

Reviewer #2 (Remarks to the Author):

The authors present a much-improved manuscript, and I agree that the technique highlighted is a novel and important addition to the field. There are still some concerns, as follows:

Author. We thank the reviewer for the appreciation of our work. We acknowledged to have satisfactorily addressed all her/his requests made in the previous reviewing since the new concerns below originated from the points raised by another Reviewer.

I think the reply to reviewer 1 on pg. 3 of the rebuttal "the suggested ChIPseq of the DamID constructs cannot be performed since their expression is extremely low and cannot be detected with antibodies" is insufficient. More modern approaches to genome-scale protein localization, i.e. Cut 'n' Run and Cut 'n' Tag experiments (e.g. PMID: 33846645, 33846646, 34637755) have been scaled down to the single cell level. Therefore, in population experiments, it seems quite likely that even lowly-expressed proteins should provide detectable signals, especially since the proteins in question are histones that are in very close proximity to the DNA. Therefore, such experiments should be attempted to address the fundamental question of where the Dam-tagged NoLS and control histones are localized across the genome.

Rebuttal, p. 6: "NoLS is the key to determining nucleolar association, but it does not guarantee exclusive nucleolar localisation. This is also evident from the H2B-GFP NoLS signals in Figure 1b, where a large proportion of the signal is nuclear." I think the author's reply is insufficient here. I agree that the NoLS provides a high degree of enrichment of nucleolar association, but I also agree with the reviewers' observation of some signal throughout the nucleus in Figures 1b, c, and e. I understand the authors' point that these IF experiments are performed with greater levels of expression than used in the DamID experiments. Therefore, the best approach to resolve the issue is to perform the Cut 'n' Run experiments suggested above, using the exact transgene constructs and levels of expression as used in the DamID analysis.

Author We apologize for not sufficiently clarifying this important point. Indeed, in the previous response we forgot to include a very important information concerning the application of antibody-based methodologies to show the genomic occupancy of Dam-tagged H2B-NoLS and H2B in the corresponding ESC lines. Unfortunately, there are not antibodies against Dam or H2B-NoLS to perform ChIP or related methods such as Cut 'n' Run and Cut 'n' Tag. Accordingly, the recently developed Cut 'n' Run method coupled to DamID (pA-DamID, PMID: 32893442) is based on the fusion of protein A (pA) to Dam and the use of an antibody against a component of the nuclear lamina. Unfortunately, since we do not have antibodies against H2B-NoLS we cannot apply this methodology in our system. Thus, although it would be very nice to apply a third orthogonal approach in addition to the DNA-FISH and HiC-rDNA to further validate our Nucleolar-DamID and hence the genomic occupancy of H2B-NoLS-Dam and H2B-Dam, we cannot perform ChIP or Cut 'n' Run experiments in this system.

As stated in the manuscript, in order to compensate eventual incorporations of H2B-NoLS-Dam in genomic regions outside the nucleolus, we used as control H2B-Dam. We believe that this is a key control, since H2B and H2B-NoLS should differ only in their nuclear localization, thereby providing accurate NAD profiles that we were able to validate using HiC-rDNA and DNA-FISH and published datasets such as SPRITE.

There has been previous description and analysis of what are called here "NAD only" regions. The authors should be commended for their very thorough analysis of the genomic attributes of these regions, but there

should also be comparisons with the earlier studies that referred to such loci as “Type II” NADs (Vertii et al 2019; Bizhanova et al 2020) regarding the overlap with H3K9me3 and H3K27me3 and early replication timing.

Author. We included in the discussion information of NAD-only (our study) and Type II NADs (Bizhanova et al.) in ESCs since this is the cell system that both studies analyzed. We stated that both studies showed that NAD-only have higher gene density and gene expression than NAD-LAD (i.e Type I NAD). However, we cannot make a clear statement on the replication timing and H3K9me3 content since these measurements were not provided in Bizhanova et al. Further, the analysis of H3K27me3 in Bizhanova et al did not compare Type I and Type II NAD making impossible any direct comparison between our and their data.

Reviewer #3 (Remarks to the Author):

Remarks to the authors:

In general I find the paper from Bersaglieri et al. “Genome-wide maps of nucleolus interaction reveal distinct layers of repressive chromatin domains” very interesting and a valuable resource for the fields of chromatin organization and epigenetic regulation. Their findings provide solid novel informations to better understand 3D chromatin organization and the underlying molecular mechanisms.

The data presented are of high quality and the conclusions have been further strengthened upon revision.

The authors have addressed all my points and concerns with the exception of my point 1, which I feel have been misunderstood. I highly recommend this article for publication in Nature Communication but I would like my point 1 to be experimentally addressed or eventually discussed if not possible to be performed.

General Comments: The authors have chosen to normalize the H2B-Dam_NoLS to H2B-Dam to compensate for an eventual incorporation of the tagged H2B into chromatin regions outside of the nucleolus.

They have shown many convincing controls to exclude this, however they did not consider the possibility that incorporation of H2B-Dam-NoLS due to its rather big size could interfere with rRNA expression and integrity of the nucleolar chromatin. In particular

1. in Fig. 1D they could show that localization of TTF1 in the presence of H2B-Dam_NoLS without ActD is not affected.

Author: We included new IF analysis analyses (Fig. 1c and e) using NPM1 and Fibrillarin as nucleolar markers, which are also known to alter their cellular localization upon downregulation of rRNA transcription (caps for Fibrillarin and delocalization from the nucleolus for NPM1). The data supported the nucleolar localization of the H2B-NoLS and showed no alterations in NPM1 or Fibrillarin localization in cell expressing H2B-GFP-NoLS.

Reviewer: In the new Fig.1e the authors show that nucleolar localization of H2B-GFP, TTF1-GFP and H2B-NoLS-GFP is not affected by ActD treatment. According to the Figure legend description H2B-Dam-NoLS has not been cotransfected together with GFP-TTF1. This experiment does not answer my concern about the correct localization of TTF1 in the presence of H2B-Dam-NoLS without ActD treatment. This is important to make sure that the nucleolar chromatin organization is not affected by overexpression of the big fusion protein H2B-Dam-NoLS. As TTF1 antibody are available one could perform IF of endogenous TTF1 in cells transfected with H2B-Dam-NoLS.

Authors. We apologize for not sufficiently clarifying our experimental set up to determine whether H2B-Dam-NoLS affects the integrity of nucleolar chromatin. To answer this question, instead to use TTF1 (as suggested by the reviewer), we used staining of Fibrillarin that associates with rRNA genes (as TTF1) and is a marker of nucleolar chromatin integrity (as TTF1) since it localized at caps upon nucleolar stress (as TTF1 does). The similar features of TTF1 and Fibrillarin are also evident in **Fig. 1e** in cells treated with ActD, where TTF1 and Fibrillarin co-localize at the nucleolar caps. Thus, nucleolar chromatin alterations in cells can be detected with both Fibrillarin and TTF1. However, the field preferentially uses Fibrillarin since Fibrillarin antibodies are highly validated whereas, as described below, the commercially available TTF1 antibodies are less and that is why the few studies on TTF1 usually use the transfection of GFP-TTF1 constructs, as we have also done (**Fig. 1e**).

We have chosen to use Fibrillarin for four main reasons.

1. We could not perform co-transfection experiments as suggested by the reviewers since both TTF1 and H2B-NoLS are GFP-tagged.

2. The use of Fibrillarin antibodies allowed to show cases of nuclear chromatin alterations (i.e. caps) to readers not familiar with the nucleolus using the co-staining of Fibrillarin and the transfected GFP-TTF1.

3. Finally, to our knowledge, there are no good validated and commercially available TTF1 antibodies. There are many commercial antibodies that are mistakenly referred as Pol I factor TTF1. However, these antibodies are for NKX2-1 (that has as alias name TTF1) that is much smaller than TTF1 (42 kDa whereas TTF1 is ca. 110 kDa) and does not localize in the nucleolus. In the past, we had some few amounts of an in-house made TTF1 antibodies. Unfortunately, this good antibody is not anymore available.

Thus, having already in our possess validated tools for the detection of endogenous proteins such as Fibrillarin as marker of nucleolar chromatin alterations in cells, we decided to use this experimental strategy to analyze whether H2B-Dam-NoLS could affect the integrity of nucleolar chromatin in cells. These data are also supported by the staining of NPM1, another nucleolar marker that is affected by nucleolar stress (**Fig. 1c**). Further, we would like to clarify that alterations in nucleolar chromatin are always associated with downregulation of rRNA transcription. The data of **Figure S2d** supported the IF analyses by showing no effect on rRNA transcription.

2. It would be important to show that H2B-Dam_NoLS does not affect rRNA expression by RT-PCR

Author: We measured 45S pre-rRNA in the ESC lines expressing H2B-Dam and H2B-Dam-NoLS upon Dox/Shield treatment. We did not find any significant changes in rRNA transcription. We also analysed the expression of pluripotency genes and found no alterations, indicating that the expression of the Nucleolar-DamID constructs does not affect pluripotency states. These new results have been included in the Figure S2d. Further, we provided images of ESCs wt and the H2B-Dam and H2B-NoLD-Dam ESC lines before and after Dox/Shield treatment showing that the morphology of the ESCs (i.e. compact and bright colonies) is not altered (new Figure S2c).

Reviewer: Thank you for addressing this. It is now evident that overexpression of H2B-Dam-NoLS does not interfere with correct transcriptional regulation of rRNA expression

All my minor points have been addressed

Author. We thank this reviewer for the constructive comments.

REVIEWERS' COMMENTS

Reviewer #1 (Remarks to the Author):

While the work is a valuable attempt to establish and apply a novel approach for NAD analysis, unfortunately the most critical questions remained still open. The authors have made a variety of arguments, many of which sound good at first, as to why they cannot or need not further improve and validate various aspects of the work. However, these are only arguments, and reading through the moderately modified text, the manuscript still reads as if all points have been strongly validated, when in fact they have only been argued in the rebuttal letters. I am concerned that the manuscript in its current form still gives a stronger impression of the results than they actually are and has not adequately qualified these results for the reader in the same way that the authors qualified the results for the reviewers.

The analytical performance of nucleolar DamID is clearly limited due to the low signal-to-noise ratio. This is undoubtedly far below that of other DamID techniques, including AP3-DamID. The conclusiveness of the study is further compromised by the non-comprehensive FISH. In comparison, the first studies more than a decade ago included more comprehensive single-cell analyses (PMID:20361057, PMID:20826608). PMID:20361057 shows FISH analyses of 11 chromosomal regions in two different cell lines. PMID:20826608 includes duplex FISH analyses of 6 chromosomal regions (three pairs, pairwise from the same chromosome) and additional live-cell imaging experiments complement the single cell validation of the results obtained by genomic analyses. Importantly, in these old studies in which NAD mapping was introduced based on high-throughput genomics using isolated nucleoli, only a single cell state was analysed. Comparative NAD analysis of two fixed cell states using a novel approach requires much more comprehensive single cell validation, as suggested during the review process.

Since the first report of a new technique serves as a reference for all subsequent studies, these critical points are particularly important. Therefore, as it stands, the study can only be considered as an interesting first attempt that needs further optimisation. The critical presentation of the flaws along with suggestions for improvement would definitely help the community to recognise the risks when aiming to explore and understand the genome organisation around the nucleolus using this NAD mapping approach without improving it further. I find this all very pity, because I think the datasets presented contain a lot of interesting and potentially valuable information, and the novel NAD mapping approach could be very useful after some optimisation.

Reviewer #2 (Remarks to the Author):

The authors have responded thoroughly to the reviewers, and I now recommend publication.

Reviewer #3 (Remarks to the Author):

The authors have now clarified my previous concern. The new experiments clearly add to the contribution of this paper. I recommend the manuscript from Bersaglieri et al. for publication in Nature Communications.

Response to reviewers' comments

Reviewer #1 (Remarks to the Author):

While the work is a valuable attempt to establish and apply a novel approach for NAD analysis, unfortunately the most critical questions remained still open. The authors have made a variety of arguments, many of which sound good at first, as to why they cannot or need not further improve and validate various aspects of the work. However, these are only arguments, and reading through the moderately modified text, the manuscript still reads as if all points have been strongly validated, when in fact they have only been argued in the rebuttal letters. I am concerned that the manuscript in its current form still gives a stronger impression of the results than they actually are and has not adequately qualified these results for the reader in the same way that the authors qualified the results for the reviewers.

Authors: Our manuscript is based on solid data as also stated by the two other reviewers. The text of the manuscript has been modified according to the requests of the reviewers. However, it is unfeasible to include all the topics discussed during the revision process with this Reviewer since many of them were either already described and discussed in the manuscript or were not relevant to this work (i.e. nucleolar localization of Dam, GATC frequency in the mouse genome, etc). In any case, these reviewer's responses will also be published, and the readers will have a complete view of the many topics discussed during the revisions.

The analytical performance of nucleolar DamID is clearly limited due to the low signal-to-noise ratio. This is undoubtedly far below that of other DamID techniques, including AP3-DamID.

Authors: This point has been extensively discussed in the previous response. The reasons to choose H2B-Dam as control of the Nucleolar-DamID have been described in the text and in the previous rebuttal letters that clarify why it is not correct to compare our Nucleolar-DamID to other DamID methods that use other controls such as GFP-Dam. Thus, we do not see any reasonable argument that limits the analytical performance of the Nucleolar-DamID. This point that was not criticized by the two other reviewers, which are experts in genomics and DamID and defined our work as “*novel and an important addition to the field*”, “*valuable resource*”, “*valuable study*”, and “*of high quality*”.

The conclusiveness of the study is further compromised by the non-comprehensive FISH. In comparison, the first studies more than a decade ago included more comprehensive single-cell analyses (PMID:20361057, PMID:20826608). PMID:20361057 shows FISH analyses of 11 chromosomal regions in two different cell lines. PMID:20826608 includes duplex FISH analyses of 6 chromosomal regions (three pairs, pairwise from the same chromosome) and additional live-cell imaging experiments complement the single cell validation of the results obtained by genomic analyses. Importantly, in these old studies in which NAD mapping was introduced based on high-throughput genomics using isolated nucleoli, only a single cell state was analysed. Comparative NAD analysis of two fixed cell states using a novel approach requires much more comprehensive single cell validation, as suggested during the review process.

Authors: We do not think that the conclusiveness of our study is compromised by the non-comprehensive FISH. We validated our data in many ways that are not only limited to the DNA-FISH (i.e., HiC-rDNA and SPRITE). In the case of DNA-FISH we used 6 DNA-FISH probes and all of them validated our NAD profiles in both ESCs and NPCs. Further, as stated in the manuscript, previously published DNA-FISH for the analysis of NADs in ESCs confirmed the validity of the Nucleolar-DamID (Bizhanova et al., 2020).

The reviewer refers to the DNA-FISH validation performed in Nemeth et al (PMID:20361057) and van Koningsbruggen et al. (PMID:20826608), which were also cited in our work. However, the reviewer did not mention important details that should be considered for a correct comparison between our and these works. Nemeth et al (PMID:20361057) used 11 DNA-FISH probes. However, only 5 of them were for NADs while the rest served for positive or negative controls. The work of van Koningsbruggen et al. (PMID:20826608) used 6 DNA-FISH probes as we did. The fact that they used pairs of probes did not add any relevant

information and, accordingly, it was not discussed. This strategy only reduced the numbers of cells to be imaged and analyzed. The live-cell imaging experiments, on the other hand, were performed to analyse NADs during cell cycle, a point that we did not aim to address in our work.

Thus, the DNA-FISH analyses and the numbers of DNA-FISH probes used in our study meet the gold-standard of the field in validating nuclear positioning of domains detected by genome technologies. This is supported by the other two reviewers that did not request any additional validation.

Since the first report of a new technique serves as a reference for all subsequent studies, these critical points are particularly important. Therefore, as it stands, the study can only be considered as an interesting first attempt that needs further optimisation. The critical presentation of the flaws along with suggestions for improvement would definitely help the community to recognise the risks when aiming to explore and understand the genome organisation around the nucleolus using this NAD mapping approach without improving it further. I find this all very pity, because I think the datasets presented contain a lot of interesting and potentially valuable information, and the novel NAD mapping approach could be very useful after some optimisation.

Authors: We described all the details of the Nucleolar-DamID in results and method section and provided solid data that validate the application of it, including DNA-FISH, HiC-rDNA, and SPRITE. As already stated in the previous rebuttal letter, we did not encounter any technical issue for the Nucleolar-DamID and, consequently, we did not list them in the manuscript. If researchers intend to apply Nucleolar-DamID to their cell system, we will always be able to give advice. This is already happening. Further, although our work has established novel methods for the analyses of NADs, we think that the most important advance of our work are the novel findings showing how chromosomes are organized around the nucleolus in ESCs and derived NPCs. Accordingly, these results represent the majority of the data that were placed in 6 out of 7 main Figures (Figs 2-7) and 8 out of 10 Suppl. Figure (Figs. S3-10).

Reviewer #2 (Remarks to the Author):

The authors have responded thoroughly to the reviewers, and I now recommend publication.

Authors: Thank you!

Reviewer #3 (Remarks to the Author):

The authors have now clarified my previous concern. The new experiments clearly add to the contribution of this paper. I recommend the manuscript from Bersaglieri et al. for publication in Nature Communications.

Authors: Thank you!